# Climatology and long-term evolution of ozone and carbon monoxide in the UTLS at northern mid-latitudes, as seen by IAGOS from 1995 to 2013

Yann Cohen[1,2], Hervé Petetin[1], Valérie Thouret[1], Virginie Marécal[2], Béatrice Josse[2], Hannah Clark[1], Bastien Sauvage[1], Alain Fontaine[1], Gilles Athier[1], Romain Blot[1], Damien Boulanger[3], Jean-Marc Cousin[1], and Philippe Nédélec[1]

[1]Laboratoire d'Aérologie, Université de Toulouse, CNRS, UPS, France
[2]Centre National de Recherches Météorologiques, CNRS-Météo-France, UMR 3589, Toulouse, France
[3]Observatoire Midi-Pyrénées, Toulouse, France

*Correspondence to:* yann.cohen@aero.obs-mip.fr

**Abstract.** In situ measurements in the upper [..[1] ]troposphere–lower stratosphere (UTLS) are performed in the framework of the European research infrastructure IAGOS (In-service Aircraft for a Global Observing System) for ozone since 1994 and for carbon monoxide (CO) since 2002. The flight tracks cover a wide range of longitudes in the northern extratropics, extending from the North American western coast (125°W) to the eastern Asian coast (135°E), and more recently over the northern Pacific [..[2] ]Ocean. Several tropical regions are also sampled frequently, such as the Brazilian coast, central and southern Africa, southeastern Asia and the [..[3] ]western half of Maritime Continent. As a result, a new set of climatologies for O$_3$ [..[4] ](Aug. 1994–Dec. 2013) and CO (Dec. 2001–Dec. 2013) in the upper troposphere (UT), tropopause layer and lower stratosphere (LS) are made available, including [..[5] ]gridded horizontal distributions on a semi-global scale, and seasonal cycles over eight well sampled regions of interest in the northern extratropics. The seasonal cycles generally show a summertime maximum in O$_3$ and a springtime maximum in CO in the UT, in contrast [..[6] ]to the systematic springtime maximum in O$_3$ and the [..[7] ]quasi-absence of seasonal cycle of CO in the LS. This study highlights some regional variabilities in the UT notably (i) a [..[8] ]west–east difference of O$_3$ in boreal summer with up to 15 ppb more O$_3$ over central Russia compared with northeast America, (ii) a systematic [..[9] ]west–east gradient of CO from 60°E to 140°E[..[10] ], especially noticeable in spring and summer with about 5 ppb by 10 degrees longitude[..[11] ], (iii) a broad spring/summer maximum of CO over Northeast Asia, and (iv) a spring maximum of O$_3$ over Western North America. Thanks to almost 20 years of O$_3$ and 12 years of CO

---

[1]removed: troposphere - lower
[2]removed: ocean. Different
[3]removed: western
[4]removed: (Aug. 1994 – Dec. 2013) and CO (Dec. 2001 – Dec. 2013)
[5]removed: quasi-global
[6]removed: with
[7]removed: quasi- absence
[8]removed: west-east
[9]removed: west-east
[10]removed: (
[11]removed: )

measurements, the IAGOS database is a unique data set to derive trends in the UTLS at northern mid-latitudes. Trends in $O_3$ in the UT are positive and statistically significant in most regions, ranging from +0.25 to +0.45 ppb.yr$^{-1}$, characterized by the significant increase of the lowest values of the distribution. No significant trends of $O_3$ are detected in the LS. Trends of CO in the UT, tropopause and LS are all negative and statistically significant. The estimated slopes range from -1.37 to -0.59

5 ppb.yr$^{-1}$, with a nearly homogeneous decrease of the lowest values of the monthly distribution (fifth percentile) contrasting with the high inter-regional variability of the decrease in the highest values ([..[12] ]95[th] percentile).

# 1 Introduction

Ozone plays an important role in the thermal structure of the stratosphere, and in the oxidizing capacity of the troposphere as a major source of hydroxyl radicals (OH). Tropospheric ozone is also a strong greenhouse gas (IPCC, 2013, AR5). In contrast

with carbon dioxide ($CO_2$) and methane ($CH_4$), the scientific understanding concerning its radiative forcing [..[13] ]is still at a medium level. Nevertheless, [..[14] ]its impact on surface temperature [..[15] ]has been shown to reach its maximum when changes [..[16] ]in $O_3$ mixing ratios occur in the upper [..[17] ]troposphere–lower stratosphere (UTLS) region (e.g. Riese et al., 2012). Carbon monoxide (CO) is a precursor for $CO_2$ and tropospheric $O_3$. It is also a major sink for OH radicals in unpolluted atmosphere (Logan et al., 1981; Lelieveld et al., 2016) thus increasing the lifetime of $CH_4$, such that CO emissions are con-

sidered as "virtually certain" to cause a positive [..[18] ]radiative forcing (IPCC, 2013, AR5). Because of its ∼2-month lifetime (Edwards et al., 2004), it can be used as a tracer of combustion processes, mainly anthropogenic emissions and biomass burning (Granier et al., 2011), although it is also a product of the oxidation of $CH_4$ and isoprene ($C_6H_8$). Identifying the contributions from different factors on $O_3$ concentrations in the UTLS is more difficult than for CO. First, the $O_3$ mixing ratio is influenced by transport from both the stratosphere (e.g. Hsu and Prather, 2009; Stevenson et al., 2013) and the troposphere (e.g. Barret

et al., 2016; Zhang et al., 2016). Second, tropospheric $O_3$ is produced from a wide variety of precursors ([..[19] ]non-methane volatile organic compounds, CO, $HO_2$) emitted from numerous surface sources, both natural and anthropogenic. Third, nitrogen oxides ([..[20] ]$NO_x$) implicated in the production of $O_3$ are [..[21] ]not only emitted at the surface by combustion processes, but also produced in the free troposphere by lightning. In parallel, [..[22] ]nitrogen oxides can be released by the decomposition of reservoir species such as peroxyacetyl nitrate (PAN), a long-lived species that can be transported at intercontinental scales

in the free troposphere (e.g. Hudman et al., 2004).

Establishing trends in $O_3$ and CO with observations is important for evaluating the impact of the reduction of anthropogenic

---

[12]removed: 95th

[13]removed: (RF)

[14]removed: it has been shown that

[15]removed: reaches

[16]removed: to

[17]removed: troposphere - lower

[18]removed: RF

[19]removed: NMVOCs

[20]removed: NOx

[21]removed: emitted not only

[22]removed: NOx

emissions in the northern [..<sup>23</sup> ]mid-latitudes on the atmospheric chemical composition. [..<sup>24</sup> ]Trends in both gases are also important for assessing changes in atmospheric radiative forcing. Furthermore, both observed trends and climatologies (i.e. time-averaged data) are used as reference values in order to evaluate the ability of models to reproduce the past atmospheric composition, and thus to forecast future changes. Consequently, assessing the mean distribution and the long-term evolution of

[..<sup>25</sup> ]$O_3$ and CO is within the focus of several research programs dealing with the understanding of atmospheric composition, regarding air quality and climate issues.

The present study is based on the measurements of these two trace gases, available in one single data set, the IAGOS database (http://www.iagos.org). In the framework of the IAGOS program (In-service Aircraft for a Global Observing System; Petzold et al., 2015), the measurements on board commercial aircraft at [..<sup>26</sup> ]cruise levels (9–12 km above sea-level) provide an accu-

rate sampling with a high frequency in the UTLS, particularly over northern mid-latitudes. With a monitoring period reaching [..<sup>27</sup> ]19 years for $O_3$ and 12 years for CO in this area, the [..<sup>28</sup> ]time series based on the IAGOS data set can provide helpful information for shorter aircraft campaigns, allowing the quantification of background values as an interannually variable signal. Also, with a dense geographical coverage spreading over the mid-latitudinal zonal band (except the Pacific [..<sup>29</sup> ]Ocean so far as regular measurements only started in 2012), spatial distributions seen by IAGOS are a source of complementary informa-

tion for local observations, quantifying the spatial variability at synoptic and intercontinental scales. The monitoring period is now sufficient to derive representative climatologies and long-term evolution of both species over different regions of interest.

Since the 1990s, several observation systems have been monitoring these two species, aiming at a better understanding of the processes controlling their spatial distribution and temporal evolution, (multi-)decadal trends and interannual variability [..<sup>30</sup>

]in the UTLS. Satellite-based instruments provide a global coverage but with limited vertical resolution while sondes, LiDAR (Light Detection And Ranging) and aircraft in situ measurements during ascent and descent phases offer a precise view of the vertical profile over a limited area. Neu et al. (2014b) have recently presented a comprehensive intercomparison of currently available satellite [..<sup>31</sup> ]$O_3$ climatologies (gathering limb and nadir viewing)[..<sup>32</sup> ], in the 300–70 hPa range. Although suffering from a coarse vertical resolution, such an analysis offers a basis for comparison of the large-scale spatio-temporal

characteristics of the $O_3$ distribution (e.g. zonal mean cross sections, seasonal variability, [..<sup>33</sup> ]interannual variability). The authors point out the large differences in these satellite-based climatologies in the UTLS. [..<sup>34</sup> ]Observing this layer often

---

<sup>23</sup>removed: midlatitudes

<sup>24</sup>removed: It is also important for enhancing the knowledge about the role of $O_3$ in the increasing atmospheric radiative forcing.

<sup>25</sup>removed: these two trace gases

<sup>26</sup>removed: 9-12 km a.s.l. cruise levels

<sup>27</sup>removed: approximately 20

<sup>28</sup>removed: time-series

<sup>29</sup>removed: ocean

<sup>30</sup>removed: (IAV)

<sup>31</sup>removed: (

<sup>32</sup>removed: $O_3$ climatologies in the UTLS (300-70 hPa )

<sup>33</sup>removed: IAV)within the UTLS

<sup>34</sup>removed: A

requires a more detailed picture[..[35] ], with a higher [..[36] ]spatio-temporal resolution and multiple species[..[37] ], that can all be addressed with aircraft platforms. Tilmes et al. (2010) proposed an integrated picture of $O_3$, CO and $H_2O$ climatologies over [..[38] ]the Northern Hemisphere, mostly representative of North America and Europe[..[39] ], based on research aircraft campaigns. They provide a valuable data set representing the vertical gradients of $H_2O$, $O_3$ and CO in three regimes: tropics,

[..[40] ]subtropics, and the polar region in the Northern Hemisphere. Compared to commercial aircraft, research aircraft can fly at higher altitudes. Such a study has [..[41] ]taken the most of this advantage, allowing tropopause-referenced tracer profiles and [..[42] ]tracer–tracer correlations, and highlighting further the transport and mixing processes on a wider vertical range[..[43] ]. However, the IAGOS program using commercial aircraft as a multi-species measurement platform [..[44] ]enables a better characterization of inter-regional differences, given its higher spatio-temporal resolution. One of the objectives of the

present study is to emphasize the use of IAGOS as a complementary database to the above-mentioned ones, allowing further investigations on composition and trends in $O_3$ and CO in the UTLS ([..[45] ]8–12 km) in the [..[46] ]Northern Hemisphere.
[..[47] ]Concerning carbon monoxide, previous studies using satellite measurements have shown a significant decadal decrease in the CO column at the global scale, with a higher sensitivity in the mid-troposphere. Worden et al. (2013) found a negative trend over several northern extratropical regions with [..[48] ]Measurements of Pollution in the Troposphere [..[49] ](MOPITT) and

Atmospheric InfraRed Sounder (AIRS), respectively from 2000 to 2011 and from 2003 to 2011. Laken and Shahbaz (2014) derived a globally negative trend in the CO column with MOPITT from March 2000 to April 2012, despite significant increases in some tropical and subtropical regions. In the upper troposphere[..[50] ], aircraft measurements have also highlighted regional decreases in CO [..[51] ]mixing ratios. A significant negative trend was derived with MOZAIC flights (Measurements of water vapor and Ozone by Airbus In-service airCraft), over western Europe from 2003 to 2009 (Gaudel et al., 2015b). More

specifically above Frankfurt airport, Petetin et al. (2016) showed a decrease in CO using IAGOS data recorded during vertical ascents and descents from [..[52] ]2002 to 2012. Recently, Jiang et al. (2017) suggested that the trends in CO were likely to be

---

[35]removed: of the UTLS

[36]removed: spatial and temporal

[37]removed: is often required and can

[38]removed: northern hemisphere (

[39]removed: )

[40]removed: sub-tropics, and polar region of northern hemisphere. Research aircrafthave a major advantage that they may fly at high

[41]removed: allowed

[42]removed: tracer-tracer correlationsto further highlight

[43]removed: than allowed by commercial aircraft

[44]removed: presents the advantage of higher spatial and temporal resolutionto further characterize regional differences

[45]removed: 8-12

[46]removed: northern hemisphere

[47]removed: Previous

[48]removed: MOPITT (

[49]removed: ) and AIRS (

[50]removed: (UT)

[51]removed: concentrations

[52]removed: 2003

caused by the decrease [..53 ]in emissions from United-States, eastern China and Europe from 2001 to 2015, [..54 ]though CO emissions in the European region [..55 ]have leveled off since 2008.

Recent studies concerning $O_3$ trends show more varied results. Based on SCIAMACHY ([..56 ]SCanning Imaging Absorption spectroMeter for Atmospheric CHartographY) tropospheric column measurements during [..57 ]2003–2011, Ebojie et al.

(2016) derived increases over Alaska and decreases over the North American outflow, southern Europe and Siberia. Tropospheric Ozone Mapping System (TOMS) showed an increasing tropospheric $O_3$ column in both [..58 ]the North and South Pacific Ocean at mid-latitudes from 1979 to 2003. Neu et al. (2014a) highlighted an increase in mid-tropospheric $O_3$ (510 hPa) in the 30°[..59 ]–50° zonal band from mid-2005 to 2010. Granados-Muñoz and Leblanc (2016) derived a $1\sigma$-significant increase between 7 and 10 km above sea-level (a.s.l.) over Table Mountain Facility (TMF, 35°N, 119°W, Jet Propulsion Lab-

oratory, California) from 2000 to 2015, using LiDAR measurements. Gaudel et al. (2015a) combined LiDAR and ozonesondes measurements from 1991 until 2010 throughout the troposphere over Observatoire de Haute-Provence (OHP, France). From 6 up to 8 km a.s.l., they found a $1\sigma$-significant [..60 ]positive trend. The increase is also observed at the lower part of the free troposphere at some Global Atmospheric Watch stations (GAW). Most of the [..61 ]Northern Hemisphere GAW stations above 2 km a.s.l. show a significant positive trend (Oltmans et al., 2013), indicating that the increase probably takes place

throughout the free troposphere. Free and [..62 ]upper-tropospheric $O_3$ trends derived from various instruments are gathered in the reviews of Cooper et al. (2014) and Monks et al. (2015). Amongst the in situ observations, significant positive trends since 1971 are reported from ozonesonde data above [..63 ]western Europe and Japan, and also from aircraft data in the [..64 ]upper troposphere above northeastern USA, North Atlantic Ocean, Europe, [..65 ]Middle East, southern China and Japan. More precisely during the [..66 ]1990–2010 decades, ozonesonde measurements above North America and the North Pacific

Ocean showed positive trends. Thouret et al. (2006) [..67 ]highlighted significant positive trends for $O_3$ mixing ratios above [..68 ]East United-States, North Atlantic and Europe from 1994 to 2003 with the IAGOS data set, [..69 ]both in the upper troposphere and in the lower stratosphere. The IAGOS time series is now 10 years longer and the present study brings crucial information to the interpretation of the previous trends.

---

[53] removed: of emissions from the US

[54] removed: despite the unchanged CO emissions since 2008

[55] removed: .

[56] removed: Scanning Imaging Absorption spectrometer for Atmospheric Chartography

[57] removed: 2003-2011

[58] removed: northern and southern mid-latitudal Pacific Ocean

[59] removed: -50

[60] removed: increase

[61] removed: northern hemisphere

[62] removed: upper tropospheric

[63] removed: Western

[64] removed: UT

[65] removed: Middle-East

[66] removed: 1990-2010

[67] removed: showed

[68] removed: eastern United States, northern

[69] removed: in the UT and the LS

Whereas the bulk of these studies show an increase in $O_3$ and a decrease in CO in the free troposphere, there is no available study based on in situ measurements only, in the UTLS on a decadal timescale, with a dense temporal and geographical sampling at the hemispheric scale. The objective of the present study is to examine climatologies for $O_3$ and CO derived from the whole set of IAGOS measurements at cruise levels, as well as the trends in these two trace gases at a regional scale in the northern mid-latitudes. It is both an update and an extension of Thouret et al. (2006), with a broader climatology and longer time series [..70 ]to derive trends, and a more complete data set with the addition of CO [..71 ]measurements. Today, the IAGOS database allows such an analysis using regular and high resolution measurements in the UTLS, spanning 19 years for $O_3$ and 12 years [..72 ]for CO. This paper aims at providing the most complete picture of the [..73 ]$O_3$ and CO [..74 ]distributions at a hemispheric scale based on in situ measurements. We investigate regional differences in terms of seasonal cycles and trends over the northern mid-latitudes. In Sect. 2 we describe the IAGOS program, the selected UTLS definition and the methodology used to [..75 ]analyze the data. In Sect. 3 we present the climatologies, time series and analysis of the subsequent trends.

## 2 Data and methodology

### 2.1 IAGOS data set

The data used in this study [..76 ]are provided by the IAGOS European research infrastructure (www.iagos.org; Petzold et al., 2015). IAGOS consists in automatic measurements of several trace gases and aerosols on board commercial aircraft, aiming at a better understanding of the chemical composition of the atmosphere at the global scale.

IAGOS has partly taken over the MOZAIC program (Measurement of [..77 ]OZone and water vapour on Airbus In-service [..78 ]airCraft; Marenco et al., 1998). The measurements of ozone and water vapour started in August 1994 (Thouret et al., 1998; Helten et al., 1998). The measurements of CO started in December 2001 (Nédélec et al., 2003). Routine IAGOS-Core [..79 ]measurements started in July 2011, with a new concept of aircraft system (e.g. near real time data transmission) and instruments (e.g. combined ozone and CO in one single analyzer, as detailed in Nédélec et al., 2015). [..80 ]Respectively, ozone and carbon monoxide have been monitored [..81 ]with dual-beam UV and IR absorption photometers. Measurements are affected by an uncertainty range of [± 2 % ± 2 ppb] for ozone (Thouret et al., 1998) and [± 5 % ± 5 ppb] for CO (Nédélec et al., 2003), with a temporal resolution of 4 s and 30 s[..82 ].

---

[70]removed: from which

[71]removed: . Today

[72]removed: of

[73]removed: distributions of

[74]removed: at a quasi-global

[75]removed: analyse

[76]removed: is

[77]removed: Ozone

[78]removed: Aircraft

[79]removed: (used in this study, called IAGOS hereafter)

[80]removed: Ozone

[81]removed: since Aug. 1994 and Dec. 2001 respectively,

[82]removed: respectively

Although the technology remained the same, the consistency between MOZAIC and [..[83] ]IAGOS-Core data sets raised questions and the overlapping period ([..[84] ]2011–2014) has been used to give the users the assurance that [..[85] ]MOZAIC–IAGOS is a unique data set (named IAGOS database from now on). Indeed, it is of particular importance to make sure that the long-term evolutions of $O_3$ and CO distributions would not be biased by this technical change. A deep and thorough comparison between

5 MOZAIC and IAGOS, based on several hundred vertical profiles over Frankfurt, has been summarized in Nédélec et al. (2015). The consistency between the two data sets has been demonstrated and hence, the results in the present study can be considered as independent of the transition from MOZAIC to IAGOS-Core. Thus, the recent study by Petetin et al. (2016), as well as the [..[86] ]Tropospheric Ozone Assessment Report (Gaudel and Cooper, 2017, in preparation) analysis commonly named TOAR, makes use of the [..[87] ]MOZAIC–IAGOS database as a seamless one. The period of the present study spreads from the

10 start of MOZAIC program [..[88] ]until December 2013, since the data from the following years are not [..[89] ]fully calibrated yet.

## 2.2 Methodology

### 2.2.1 Definition [..[90] ]for the upper troposphere and [..[91] ]the lower stratosphere

During each flight in extratropical latitudes, both tropospheric and stratospheric air masses are encountered. Their different properties make it necessary to distinguish between upper troposphere and lower stratosphere. Thus, we discriminate the air

masses with three varying pressure intervals, i.e. the upper troposphere (UT), the lower stratosphere (LS) and the tropopause layer (TPL) as a mixing region between.

In extratropical latitudes, the discrimination between the [..[92] ]upper-tropospheric and the lower-stratospheric measurement points is based on the dynamical definition of the tropopause, adapted from Thouret et al. (2006): it is centered on the isosurface of potential vorticity (PV) defined as [..[93] ][..[94] ]PV = 2 pvu (1 potential vorticity unit $\equiv 10^{-6}$ K.m$^2$.kg$^{-1}$.s$^{-1}$). The

20 tropopause pressure is systematically defined as the highest level of the 2 pvu isosurface. As a consequence, the double tropopause events encountered at mid-latitudes, more frequently in winter (Randel et al., 2007), are not filtered out. It implies that air masses attributed to the upper troposphere are not purely tropospheric since they may include some stratospheric intrusions. As indicated in Fig. 1, the TPL is considered as a 30 [..[95] ]hPa thick layer, centered on this 2 [..[96] ]pvu

---

[83] removed: IAGOS

[84] removed: 2011-2014

[85] removed: MOZAIC-IAGOS

[86] removed: TOAR Climate (Gaudel and Cooper, 2017, in preparation) , analysis

[87] removed: MOZAIC-IAGOS

[88] removed: up to

[89] removed: yet fully calibrated

[90] removed: of UT

[91] removed: LS

[92] removed: upper tropospheric and the lower stratospheric

[93] removed: PV = 2 pvu (1 potential vorticity unit $\equiv 10^{-6} \cdot$ K $\cdot$ m$^2 \cdot$ kg$^{-1} \cdot$ s$^{-1}$). The PV values are part of the IAGOS ancillary data (

[94] removed: ) and are derived from the ECMWF operational analysis with a $1° \times 1°$ resolution, and interpolated along the aircraft trajectories.

[95] removed: hPa-thick

[96] removed: p.v.u.

isosurface. The UT and the LS are defined as constant pressure intervals relative to the tropopause, respectively [..[97] ][..[98] ][..[99] ][$P_{2\,pvu}$ + 75 hPa, $P_{2\,pvu}$ + 15 hPa] and [$P_{2\,pvu}$ - 45 hPa, $P_{2\,pvu}$ - 150 hPa]. The large pressure interval of the LS gathers all the stratospheric data points. The choice of a transition layer rather than a surface for the tropopause definition is motivated by the need to lower the impact of the vertical uncertainties on the PV calculations. With a thickness as great as 30 hPa, the TPL provides a clear partitioning between the characteristics of the UT and the LS. Thus, the analysis focuses on the composition and trends in these two distinct layers. Note also that a comparison between the dynamical tropopause estimate based on the PV fields, and the chemical tropopause estimate based on $O_3$ vertical profiles considering all IAGOS measurements near Frankfurt (where IAGOS data are the most numerous), have shown errors but a negligible systematic bias (Petetin et al., 2016).

[..[100] ]

The PV values have been derived using [..[101] ]the European Centre for Medium-Range Weather Forecasts (ECMWF) operational analysis since 2000, with a 1° × 1° resolution, and interpolated along the aircraft trajectories. As for other ancillary data developed for the IAGOS [..[102] ]database (www.iagos.org), the up-to-date better vertical resolution of the operational analysis since 2000 was clearly more relevant [..[103] ]e.g. for the representativeness of vertical transport processes[..[104] ], thus for the precision of the ancillary data itself. This has been [..[105] ]demonstrated particularly for the coupling system (SOFT-IO) combining the FLEXPART Lagrangian particle dispersion model and emissions inventories to provide systematic [..[106] ]origins of CO mixing ratio anomalies observed by IAGOS throughout the troposphere (see Sauvage et al., 2017, for further details).

Although this study focuses on the northern mid-latitudes, we also present briefly some results from measurements in the tropics. In the 25°[..[107] ]S–25°N band, the tropopause is higher than any commercial aircraft flight level, since they are bounded by the 196 hPa pressure level (below 12 km). Consequently, all the IAGOS measurement points recorded above 8 km a.s.l. are considered as belonging to the tropical UT.

### 2.2.2 IAGOS data analysis

This study gathers climatological analyses in each layer, focusing on global horizontal distributions, latitudinal averages in the northern mid-latitudes, and seasonal cycles at subcontinental scales. We also present monthly time series, anomalies and subsequent long-term trends. The following paragraph summarizes the processing applied to the data for each of these analyses.

---

[97]removed: $P_{2\,pvu}$ + 75 hPa, $P_{2\,pvu}$ + 15 hPa

[98]removed: and

[99]removed: $P_{2\,pvu}$ - 45 hPa, $P_{2\,pvu}$ - 150 hPa

[100]removed: As mentioned above,

[101]removed: ECMWF operational analysis

[102]removed: data-set (

[103]removed: for

[104]removed: representativeness and then

[105]removed: in particular demonstrated

[106]removed: origin

[107]removed: S – 25

Horizontal distributions [..[108] ]of $O_3$ and CO are computed by gathering all [..[109] ]IAGOS measurements recorded into 5[..[110] ]$^\circ \times 5^\circ$ cells, with respect to the season and the layer. [..[111] ]Averages are calculated only in cells where the amount of [..[112] ]observations exceeds 2,000 (1,000) for $O_3$ [..[113] ](CO) over the period 1995–2013 (2002–2013). The horizontal distribution of $O_3$/CO ratios are also computed by averaging the instantaneous $O_3$/CO ratios in each cell. The selected size of the cells and thresholds in the amount of data ensure a good [..[114] ]representativeness of the time period, without filtering out a too high proportion of data.

We perform a [..[115] ]subcontinental-scaled analysis in order to take the zonal variability into account. Indeed, averages made in continental-sized regions or in zonal bands are not fitted for [..[116] ]IAGOS data because of the high geographical variability of the measurements. [..[117] ]In eight regions of interest (see Fig. 1), monthly time series are calculated by averaging all corresponding instantaneous observations (not by averaging the $5^\circ \times 5^\circ$ binned data). These monthly averages are calculated only when (i) the amount of available data points in a month exceeds 300, and (ii) the first and last measurements are separated by a 7-day period at least. These selected thresholds limit the influence of short-term events which are not representative of the whole month. The boundaries of each region are chosen as a compromise between a high level of sampling and a good representativeness of the expected impact of local surface emissions and meteorological conditions. Based on these monthly time series, mean seasonal cycles are computed over the period 1995–2013. In order to avoid inter-seasonal biases due to the interannual variability, we retained only the years with data available during 7 months distributed over 3 seasons at least. A test has been done with requiring available data for all 4 seasons. It slightly reduced the amount of complete years (2 years or less removed for each region), yielding quasi-identical results. The seasonal cycles have also been found poorly sensitive to a 2-month change in the amount of required months per year. These mean seasonal cycles in all regions are used to extract deseasonalized monthly anomalies from the original monthly time series. Based on the same population of data points as used for calculating the mean values, other useful metrics like the fifth and 95[th] percentiles of the $O_3$ and CO mixing ratios are also calculated, as well as the $O_3$/CO ratio.

---

[108] removed: from IAGOS data set

[109] removed: the measurement points

[110] removed: $^\circ \times$

[111] removed: For each cell, the average is taken into account only if

[112] removed: measurement points exceeds 2000

[113] removed: and 1000 for COover the whole period , i. e. 1995 – 2013 and 2002 – 2013 respectively.

[114] removed: representativity

[115] removed: subcontinental-scale

[116] removed: the

[117] removed: The eight regions of this study are illustrated in Fig. 1.

[..[118] ][..[119] ][..[120] ]Based on these monthly anomalies, the trends are computed using non-parametric [..[121] ]Mann–Kendall analysis combined with [..[122] ]Theil–Sen slope estimate (Sen, 1968). Calculations are performed with the OpenAir R package dedicated to atmospheric sciences (Carslaw and Ropkins, 2012). [..[123] ]Theil–Sen slope estimate corresponds to the median of the slopes defined by every couple of points. [..[124] ]In this paper, all trend uncertainties are given at a 95% confidence interval (i.e. with a statistical significance above 2 $\sigma$). The linear regression is based on the hypothesis that every point is independent of its neighbours. As a consequence, a positive (negative) autocorrelation can lead to an overestimation (underestimation) of the regression slope. In the aim of reducing such kinds of bias, the method implemented in the OpenAir package applies a correcting factor to one half of the confidence interval depending on the autocorrelation. More details are available in the Openair Manual (https://cran.r-project.org/web/packages/openair/openair.pdf).

---

[118]removed: The monthly means are computed by averaging the measurement points sampled in each region for each layer. The averages are taken into account only if the amount of available data points in a month exceeds 300, and if the first and last measurements are separated by a 7 days-period at least. These selected thresholds limit the influence of short-lived events which are not representative of the whole month. The boundaries of each region are chosen as a compromise between a high level of sampling and a good representativity of the expected impact of local surface emissions and meteorological conditions.

[119]removed: The mean seasonal cycles are computed from 1995 to 2013, without taking into account the incomplete years in order to avoid biases between two seasons. In this study, the so-called incomplete years are defined as years with less than 7 available months and less than 3 sampled seasons. A threshold at 4 sampled seasons would be too restrictive, whereas a threshold at 2 would not ensure a good representativity of the whole years, especially if the sampled seasons are changing in time. With at least 7 available months a year, we make sure that most seasons are represented by more than one month. The absolute anomaliesare derived from the deseasonalised monthly means, i.e. the monthly means after substraction of the seasonal cycle. The relative anomalies correspond to the absolute ones, divided by the mean value of the seasonal cycle.

[120]removed: The trends are then computed from these anomalies, by using

[121]removed: Mann-Kendall

[122]removed: the Theil-Sen

[123]removed: The Theil-Sen

[124]removed: The statistical significance is validated if the null hypothesis is rejected at a 95% confidence interval, i.e. if there is less than 5% probability that the observed long-term variability is only due to statistical fluctuations. The

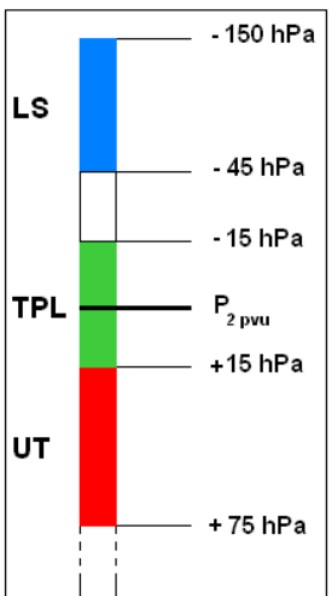
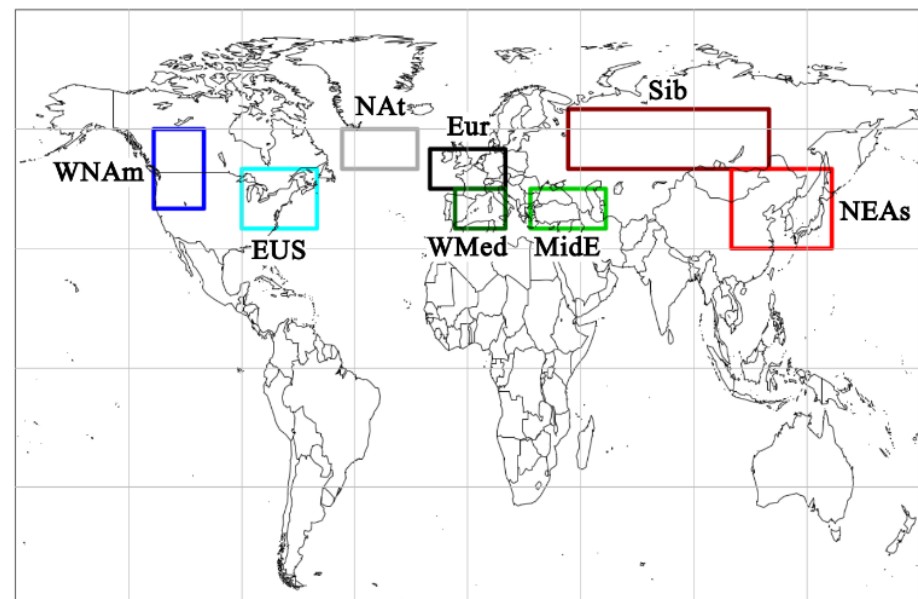

**Figure 1.** Left panel: Schematic representation of the definition of the upper troposphere (UT), tropopause layer (TPL) and lower stratosphere (LS) in red, green and blue respectively. The pressure levels are indicated on the right, relatively to the PV isosurface ($P_{2\,pvu}$). Right panel: Map of the regions compared in this study. From west to east: Western North America (WNAm), Eastern [..[125] ]United-States (EUS), North Atlantic (NAt), Europe (Eur), the Western Mediterranean basin (WMed), [..[126] ]Middle East (MidE), Siberia (Sib), and Northeastern Asia (NEAs).

The statistics of IAGOS data are [..[127] ]summarized in Tab. 1 for each region and each layer. [..[128] ]Northwest America is the least sampled region ($\simeq$ 60% of available months and $\simeq$ 1,400 flights in each layer) with a shorter time coverage per month ($\simeq$ 18 days per month): consequently, a mean value computed in this region is less likely to be a representative estimation of the monthly mean volume mixing ratio (VMR). [..[129] ]Aside from this region, the next least sampled regions are [..[130] ]the

5  Western Mediterranean basin, Siberia and Northeast Asia ($\simeq$ 75% of available months with $\simeq$ 3,000 flights in each layer), with a temporal range of 22 days per month on average. Finally, [..[131] ]East United-States, North Atlantic and northwestern Europe – named hereafter Europe – are the most sampled regions ($\simeq$ 96% of available months with $\simeq$ 9,000 flights in each layer in [..[132] ]East United-States and North Atlantic, and more than 10,000 in [..[133] ]Europe) with a mean temporal range of 27 days per month.

10  In order to ensure a good representativeness of the whole measurement period for the trends analyses, we consider the best

---

[127]removed: summarised

[128]removed: WNAm

[129]removed: Except this one, the

[130]removed: WMed, Sib and NEAs

[131]removed: EUS, NAt and Eur

[132]removed: EUS and NAt

[133]removed: Eur

estimate slope as realistic if the ratio of available months is higher or equal than 60%. Indeed, for each species and each layer, every region filling that condition is sampled almost from the start to the end of the measurement period (as shown later). Consequently, our [..<sup>134</sup> ]analysis includes neither the trends in $O_3$ in the whole UTLS above [..<sup>135</sup> ]Northwest America nor the trends in CO in the LS above [..<sup>136</sup> ]the Western Mediterranean basin.

---

[134]removed: analyse
[135]removed: WNAm
[136]removed: WMed

**Table 1.** Regional characteristics and data statistics in each layer. The third and fourth columns refer to the proportion of available monthly data relatively to the whole period: Aug. [..[137] ]1994–Dec. 2013 for $O_3$ and Dec. [..[138] ]2001–Dec. 2013 for CO. The fifth and sixth columns indicate the total number of selected flights. The seventh and eighth columns show the time interval $\Delta t$ separating the first and the last days of measurements, averaged over the available data months.

| Region | Layer | Av. months (%) | | Nb. of flights | | $\Delta t$ (days/month) | |
|---|---|---|---|---|---|---|---|
| | | $O_3$ | CO | $O_3$ | CO | $O_3$ | CO |
| Western NAm (WNAm) | LS | 53 | 65 | 1 215 | 1 160 | 21 | 22 |
| $\left(125-105°W\right.$ | TPL | 53 | 66 | 1 402 | 1 338 | 22 | 23 |
| $\left. 40-60°N\right)$ | UT | 50 | 66 | 1 471 | 1 438 | 22 | 23 |
| [..[139] ]East US (EUS) | LS | 94 | 92 | 8 536 | 3 701 | 27 | 26 |
| $\left(90-60°W\right.$ | TPL | 94 | 92 | 10 503 | 4 418 | 27 | 27 |
| $\left.35-50°N\right)$ | UT | 94 | 92 | 11 607 | 4 998 | 27 | 27 |
| North Atlantic (NAt) | LS | 94 | 92 | 9 125 | 4 149 | 27 | 27 |
| $\left(50-20°W\right.$ | TPL | 93 | 91 | 7 706 | 3 488 | 27 | 26 |
| $\left.50-60°N\right)$ | UT | 91 | 89 | 5 299 | 2 550 | 25 | 25 |
| Europe (Eur) | LS | 96 | 94 | 13 939 | 7 217 | 29 | 28 |
| $\left(15°W-15°E\right.$ | TPL | 97 | 94 | 21 162 | 10 812 | 29 | 29 |
| $\left.45-55°N\right)$ | UT | 97 | 94 | 25 021 | 13 129 | 29 | 29 |
| Western Med. (WMed) | LS | 64 | 55 | 1 806 | 1 126 | 23 | 25 |
| $\left(5°W-15°E\right.$ | TPL | 69 | 60 | 2 074 | 1 305 | 23 | 25 |
| $\left.35-45°N\right)$ | UT | 70 | 60 | 2 025 | 1 182 | 24 | 24 |
| Middle East (MidE) | LS | 89 | 88 | 3 516 | 2 260 | 24 | 23 |
| $\left(25-55°E\right.$ | TPL | 88 | 85 | 3 362 | 2 107 | 24 | 23 |
| $\left.35-45°N\right)$ | UT | 82 | 76 | 2 846 | 1 713 | 23 | 23 |
| Siberia (Sib) | LS | 76 | 68 | 4 615 | 2 126 | 26 | 26 |
| $\left(40-120°E\right.$ | TPL | 76 | 65 | 3 808 | 1 779 | 26 | 25 |
| $\left.50-65°N\right)$ | UT | 72 | 61 | 2 274 | 1 159 | 24 | 24 |
| Northeastern Asia (NEAs) | LS | 74 | 65 | 3 502 | 1 676 | 26 | 25 |
| $\left(105-145°E\right.$ | TPL | 73 | 63 | 4 326 | 2 021 | 26 | 26 |
| $\left.30-50°N\right)$ | UT | 76 | 64 | 4 834 | 2 323 | 27 | 26 |

## 3 Results

We first present the analysis of the new set of climatologies as an update of previous studies made with a shorter data set, and based on the same tropopause definition (Thouret et al., 2006; Petzold et al., 2015) as described in section 2.2.1. The present section starts with the [..140 ]characterization of the seasonal distributions at the global scale. It is followed by the analysis of
the seasonal cycles over the eight regions defined in Fig. 1. Finally, we present the analysis of the regional long-term trends (12 years for CO, 19 years for $O_3$).

### 3.1   Horizontal distributions of $O_3$ and CO

Figures 2 and 3 show the [..141 ]seasonally averaged distribution of CO and $O_3$ respectively in the UT, in 5° × 5° bins. We use the same colour scale for the four maps to highlight the seasonal and regional differences. [..142 ]By averaging over the latitude
range 25–75°N all the cells for each 5° interval in longitude, we obtain a zonal distribution that is projected below each map. It allows us to examine the zonal gradients in $O_3$ or CO, and to highlight intercontinental differences. The averaging interval has been chosen in order to include all upper-tropospheric IAGOS measurements in the area considered here as the extratropics. We sometimes refer to Fig. A1 as a support to our analysis. It represents the seasonally averaged distribution of the $O_3$/CO ratio in the UT, following the same organization as in Figs. 2 and 3.

In Fig. 2, the horizontal distributions of CO mixing ratios reveal an overall spring maximum at northern mid-latitudes, extending into summer over eastern Asia and [..143 ]the Bering Sea. There are also systematic maxima in the sampled tropical regions (i.e. Brazilian coast, central and southern Africa, southeast Asia), depending on their respective dry seasons. These features characterize the combination of the two most important sources, namely the anthropogenic emissions and the boreal/tropical biomass burning. At northern mid-latitudes, in contrast [..144 ]to near-surface observations (e.g. Zbinden et al.,
2013), the averaged CO mixing ratio is higher during spring in the UT. During winter, the weaker convection and the longer lifetime of CO allow the accumulation of this trace gas in the lower troposphere despite the stronger wintertime frequency of warm conveyor belts (e.g. Madonna et al., 2014), thus leading to a climatological maximum spread from late winter until early spring [..145 ](e.g. in western Europe: Petetin et al., 2016). As the convection activity increases, the CO confined in the lower troposphere starts being uplifted to the UT. During summer, CO is less abundant in the lower layers, and [..146 ]photochemical
activity reaches its maximum, acting as a major sink for CO by OH attack (e.g. Lelieveld et al., 2016). In the tropics, the local maxima take place during SON over the Brazilian coast, during the whole year over tropical Africa and during [..147

---

$^{140}$removed: characterisation

$^{141}$removed: 12 and 19-year

$^{142}$removed: The longitudinal distribution averaged over the latitude range 25 to 75°N is projected below each map to examine the longitudinal gradients of $O_3$ or CO, and hypothetically highlight an intercontinental difference. The averaging interval has been chosen in order to take all upper tropospheric IAGOS measurements in the area considered here as the extratropics.

$^{143}$removed: northern Pacific

$^{144}$removed: with

$^{145}$removed: (Petetin et al., 2016)

$^{146}$removed: the photochemistry

$^{147}$removed: JJA-SON

]JJA–SON over the Bay of [..[148] ]Bengal–Southeast Asia. The maxima over tropical Africa are shifted northward during MAM and southward during SON. [..[149] ]All these local maxima observed in the UT are not necessarily correlated with the dry seasons when biomass burning emissions are enhanced, because surface emissions are uplifted more efficiently during the wet seasons within deep convection (Liu et al., 2010). Further details on the distribution of CO and $O_3$ near these locations, as

seen by IAGOS, may be found in Sauvage et al. (2005, 2007a, 2007c) and Yamasoe et al. (2015). Globally, the highest values are recorded over eastern Asia (up to 170 ppb), [..[150] ]the Bering Sea (165 ppb), North America (150 ppb), subtropical Africa (145 ppb) and the Brazilian coast (145 ppb).[..[151] ]

The horizontal distributions of the $O_3$/CO ratio in Fig. A1 are anticorrelated with most CO maxima in Fig. 2. In the

10 tropics, low values between 0.4 and 0.5 are observed during the whole year over equatorial Africa and during JJA–SON over the Bay of Bengal–Southeast Asia. In the northern mid-latitudes, the low ratios correlated with high CO values are found during the whole year over eastern Asia, during summer over the northernmost Pacific area, over the west coast of Canada and, in the same country, over Nunavut and the northern Labrador peninsula. These CO maxima are characterized by a higher anticorrelation between $O_3$ and CO, showing a stronger impact of lower-tropospheric air

masses.

Further vertical information is made available with the intercomparison between the IAGOS data set and several satellite-based instruments, which have different vertical sensitivity profiles. Laat et al. (2010) present the horizontal distribution of yearly averaged CO column from SCIAMACHY during 2004–2005 (see their Fig. 7, top left panel), with an almost vertically uniform sensitivity. The maxima observed by IAGOS over central Africa and over Southeast Asia,

the latter during summer, are well represented. Both reach more than $2.75\,10^{18}$ molecules.cm$^{-2}$. On the contrary, the fall maximum over the Brazilian coast observed by IAGOS is not visible in the SCIAMACHY columns. All those three (sub)tropical maxima seen by IAGOS are visible in the seasonal climatologies of MOPITT total columns from 2001 to 2012 (Osman et al., 2016) and Aura Microwave Limb Sounder (Aura MLS) mixing ratios at 215 hPa (Huang et al., 2016). In the northern extratropics, the main difference with the MLS climatology by Huang et al. (2016) lies in the springtime

and summertime maxima over eastern Siberia and Manchuria observed by IAGOS only. One possible reason is that the altitude level at 215 hPa in the extratropics is generally included in the LS according to our tropopause definition, despite the associated 5 km vertical resolution of MLS in the UTLS which allows upper-tropospheric CO to impact the measurements at 215 hPa.

Seasonal maxima highlighted in these decadal climatologies are consistent with extreme CO events already recorded in the

---

[148]removed: Bengal-southeast

[149]removed: These

[150]removed: northern Pacific

[151]removed: These regional maxima are also shown with the climatology of MOPITT total column from 2001 to 2012 (Osman et al., 2016) and MLS mixing ratio at 215 hPa (Huang et al., 2016). The main difference with the MLS climatology by Huang et al. (2016) lies in the springtime and summertime maxima over eastern Siberia and Manchuria observed by IAGOS only. The possible reasons are the 5 km vertical resolution of MLS regarding CO mixing ratio in the UTLS, or the fact that the altitude level at 215 hPa in the extratropics generally belongs to our definition of the lower stratosphere.

UT over eastern Asia (Nédélec et al., 2005), [..[152] ]the northern Pacific Ocean (Clark et al., 2015) and downwind of Alaskan forest fires (Elguindi et al., 2010), and in the LS downwind of Alaska and Yukon right to the eastern coast of the Atlantic [..[153] ]Ocean (Cammas et al., 2009). All of them were originating from intense biomass burning.

In the northern extratropical UT, the common characteristic of the four seasons lies in the eastward increase of climatological
5  CO mixing ratio, from 60°E to 135°E. The difference in CO mixing ratios reaches $\simeq$ 20 ppb during fall and winter, and $\simeq$ 40 ppb during spring and summer. There is also a low decrease from 120°W to 60°E, $\simeq$ 20 ppb in spring and within 10 ppb during the rest of the year. It is likely the resulting combination of stronger emissions from both anthropogenic and biomass burning over the Asian continent (e.g. Jiang et al., 2017), and an efficient vertical transport over there (Madonna et al., 2014; Huang et al., 2016).

---

[152]removed: northern Pacific
[153]removed: ocean

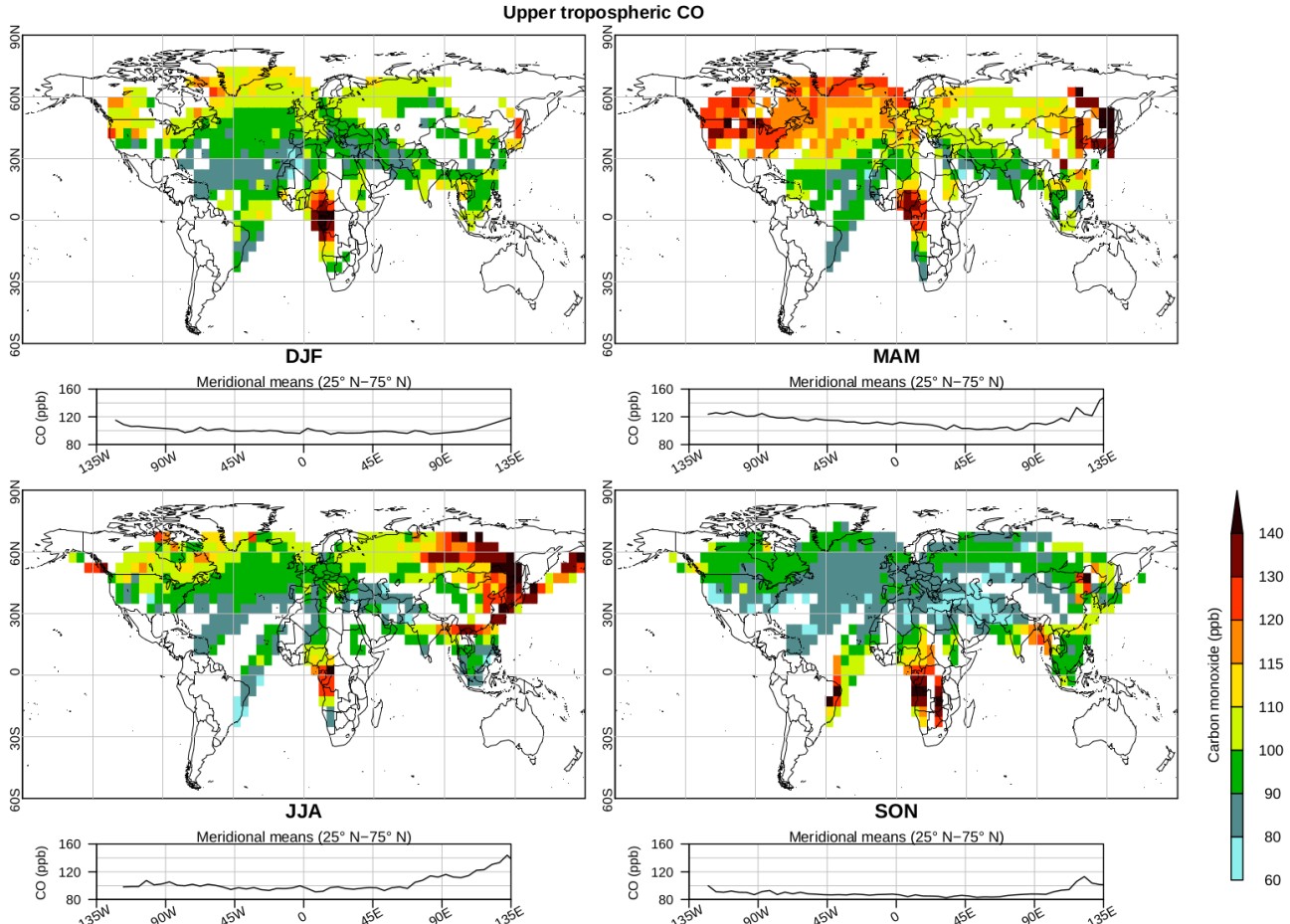

**Figure 2.** Horizontal distributions of CO volume mixing ratios in the UT, averaged from December 2001 to November 2013 [..[154] ]for each season. Each 2D distribution is projected on the zonal axis below, with a meridional average of the northern extratropical zonal band (from 25°N to 75°N). The 2D (respectively 1D) distributions have a 5° × 5° (respectively 5°) resolution. [..[155] ]The abscissa axes are the same in the maps and in the projections below. Note the larger [..[156] ]colour interval from 60 to 80 ppb, and the smaller intervals from 110 to 120 ppb.

The 20-year seasonal distribution of $O_3$ in the UT is shown in Fig. 3. The longitudinal variability is more detailed in the projections below the 2D distributions, where the 5° per 5° seasonal means are meridionally averaged from 25 to 75°N. At northern mid-latitudes, ozone clearly exhibits a minimum during [..[157] ]winter (47 ppb on average in the 25°[..[158] ]N–65°N zonal band) and a maximum during [..[159] ]summer (79 ppb), characterizing the seasonal peak of photochemical activity.

5  The highest values exceed 90 ppb over Siberia and northeastern China, while minima [..[160] ]below 30 ppb are located on both

---

[157]removed: Winter

[158]removed: N – 65

[159]removed: Summer

[160]removed: lower than

sides of [..[161] ]the equatorial Pacific Ocean, at least during the wet seasons (i.e. from December until February on the western South American coast, and from December to March in [..[162] ]Maritime Continent). In the [..[163] ]Northern Hemisphere, all over the year, the $O_3$ mixing ratio is higher on the eastern half of the extratropical zonal band: the annual mean in the Asian continent ([..[164] ]30–140°E) is 11% higher than in the western part (125°[..[165] ]W–15°E). The collocated maxima in the $O_3$/CO ratio (Fig. A1) indicate a higher occurrence of correlated rich-$O_3$ and low-CO air masses, suggesting a stronger influence from the lower stratosphere.

[..[166] ]

In the Southern Hemisphere, the SON maximum in $O_3$ over the Brazilian coast mostly reflects the influence of lightning in South America (Bremer et al., 2004; Sauvage et al., 2007b; Yamasoe et al., 2015). The minima observed at both sides of the equatorial Pacific [..[167] ]Ocean during the respective wet seasons can highlight the strong convection of low-ozone air masses from the tropical marine boundary layer, typically ranging from 10 to 30 ppb (e.g. Thompson et al., 2003).

The satellite-based instruments [..[168] ]SCIAMACHY and Ozone Monitoring [..[169] ]Instrument–Microwave Limb Sounder ([..[170] ]OMI–MLS) provided climatologies of tropospheric $O_3$ column, averaged respectively from 2003 to 2011 and from October 2004 to December 2010. They both showed a springtime maximum, and a geographical maximum over the Atlantic [..[171] ]Ocean between 30° and 45°N, which is due to [..[172] ]stratosphere–troposphere exchange events and surface transport of pollution from eastern USA during spring and summer (Fig. 1 in Ebojie et al., 2016; Figs. [..[173] ]5a–c in Ziemke et al., 2011). The discrepancies with the IAGOS climatologies can be due to uncertainties involving the stratospheric signal, i.e. the ozone stratospheric column, and/or the height of the tropopause [..[174] ](Ebojie et al., 2014). The summertime climatology of $O_3$ mixing ratio at 464 hPa provided by Tropospheric Emission Spectrometer (TES) from 2005 to 2010 in Verstraeten et al. (2013, Fig. 1) shows similar features with IAGOS climatology. The $O_3$ mixing ratios are higher above the Asian continent than west of 15°E, and there is a local maximum above Manchuria (northeastern China). In the tropics, the maximum over equatorial Africa and the minimum above and east of [..[175] ]Maritime Continent are also seen [..[176] ]with TES. At least in the extratropics,

---

[161] removed: equatorial Pacific
[162] removed: the
[163] removed: northern hemisphere
[164] removed: 30 – 140
[165] removed: W – 15
[166] removed: In the southern hemisphere
[167] removed: ocean
[168] removed: Scanning Imaging Absorption Spectrometer for Atmospheric Chartography (SCIAMACHY )
[169] removed: Instrument-Microwave
[170] removed: OMI-MLS
[171] removed: ocean
[172] removed: STE
[173] removed: 5a-c
[174] removed: , and the total ozone column (Ebojie et al., 2014).
[175] removed: the Maritime continent
[176] removed: by

the altitude range observed with TES corresponds to [..[177] ]100–150 hPa below the lower limit of our definition of the UT. Thus, [..[178] ]in summer, the properties we observe in the UT may extend downward, representing a wider upper part of the free troposphere.

In the tropics, Livesey et al. (2013) used [..[179] ]Aura MLS observations at 215 hPa since 2004 until 2011. They highlighted CO maxima over India and southeast Asia during [..[180] ]July–August, northern equatorial Africa in [..[181] ]February–April, southern equatorial Africa in [..[182] ]September–November, and equatorial Brazil in [..[183] ]October–November. According to their study and references therein, the two maxima over Asian regions are linked with anthropogenic emissions uplifted to the tropical UT by strong convection (Jiang et al., 2007), whereas the other maxima [..[184] ]originate from biomass burning. [..[185] ]Most of these maxima are consistent between the MLS and IAGOS data sets. One exception may be northern India (25–30°N cells) where IAGOS does not clearly highlight a summertime maximum. However, available IAGOS data in this region and season are too scarce to conclude, likely because IAGOS aircraft often fly below the UT lower boundary ($P_{2\ pvu} + 75$ hPa) in the 25–30°N zonal band that becomes subtropical in summer, with a tropopause typically above the 150 hPa altitude level. $O_3$ maxima are also shown by MLS above subtropical Brazil and southern subtropical Africa during [..[186] ]September–November. Last, the climatological means they derived reached their geographical minima all along the equatorial Pacific [..[187] ]Ocean (15°[..[188] ]S–15°N), which is consistent with the overall minima observed by IAGOS at both sides of the equatorial Pacific [..[189] ]Ocean.

The results shown in Fig. A1 motivated our choice in several regions of interest presented in Sect. 2.2.2. Lower values characterize Northwest America and Northeast Asia, whereas higher values are found over the whole Mediterranean basin and especially over Middle East.

---

[177] removed: 100-150

[178] removed: during

[179] removed: Aura-MLS

[180] removed: July-August

[181] removed: February-April

[182] removed: September-November

[183] removed: October-November

[184] removed: originated

[185] removed: Except for India where few summertime IAGOS data do not allow the comparison, all these maxima are consistent between MLS and IAGOS data sets. Besides,

[186] removed: September-November

[187] removed: ocean

[188] removed: S-15

[189] removed: ocean.

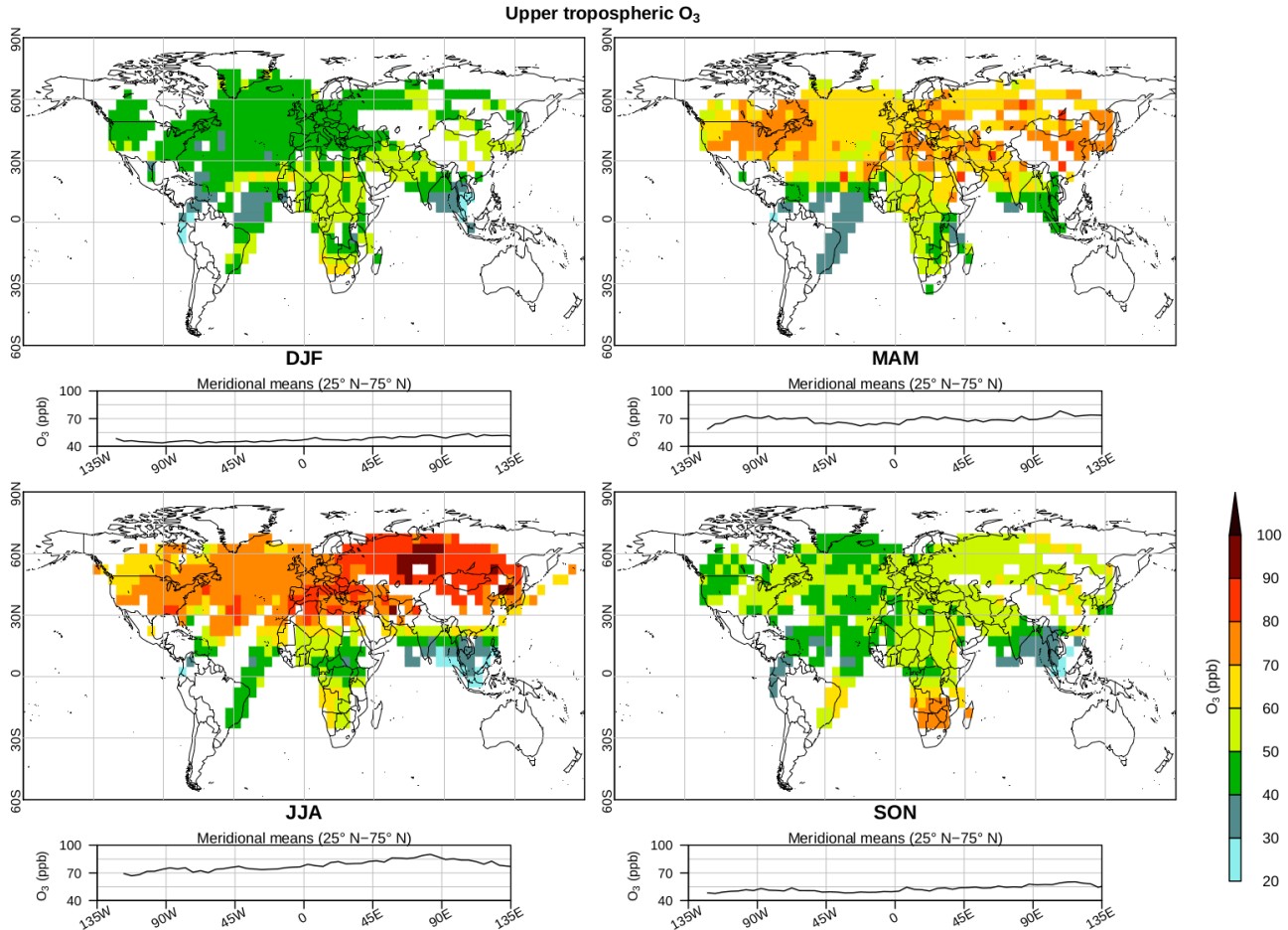

**Figure 3.** Horizontal distributions of O$_3$ volume mixing ratios in the UT averaged from December 1994 to November 2013, in every season. Each 2D distribution is projected on the zonal axis below, with a meridional average of the northern extratropical zonal band (from 25°N to 75°N). The 2D (respectively 1D) distributions have a 5° × 5° (respectively 5°) resolution. The abscissa axes are the same in the maps and in the projections below.

The horizontal distribution of the O$_3$ mixing ratios in the LS is presented in Fig. 4. It contrasts with the [..[190] ]upper-tropospheric distribution, thus confirming the present definition of tropopause as a realistic transition layer between the troposphere and the stratosphere. The [..[191] ]lower-stratospheric climatology is characterized by a strong maximum during [..[192] ]spring (approximately 33% more than the annual concentrations). It originates from the [..[193] ]January–February maximum of the downward O$_3$ flux at $\theta$ = 380 K both driven by the [..[194] ]Brewer–Dobson circulation, and by the stratospheric photo-

---

[190]removed: upper tropospheric

[191]removed: lower stratospheric

[192]removed: Spring

[193]removed: January-February

[194]removed: Brewer-Dobson circulation(BDC)

chemistry which activates at the end of winter. This feature is followed by approximately 100 days of transport until reaching the tropopause (Olsen et al., 2004). In the low latitudes of the northern extratropics, the distribution shows a poleward gradient, consistent with the adiabatic transport between the tropical UT and the extratropical lowermost stratosphere. [..[195] ]For all seasons but autumn, the highest values are observed over [..[196] ]Canada–Greenland and northern Siberia. During spring, these

5    results are coherent with the climatology of the downward [..[197] ]$O_3$ flux shown in Olsen et al. (2013, Fig. 5), where [..[198] ]the $O_3$ mixing [..[199] ]ratios derived from MLS observations in the stratosphere ($\theta = 380$ K) [..[200] ]are combined with the MERRA reanalysis meteorological fields.

---

[195] removed: During

[196] removed: Canada-Greenland

[197] removed: 380 K

[198] removed: observations of

[199] removed: ratio

[200] removed: from Microwave Limb Sounder (MLS)

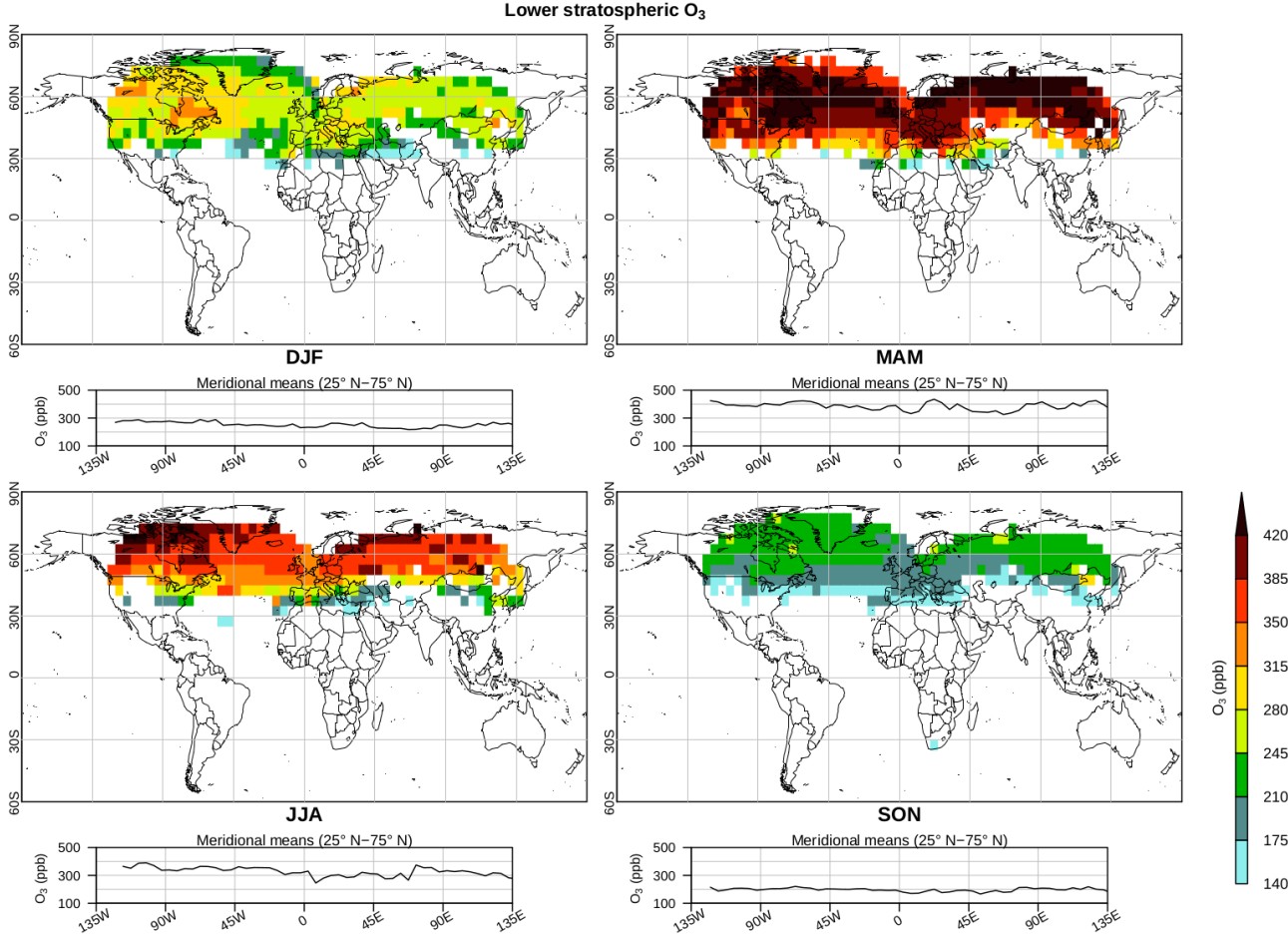

**Figure 4.** [..[201] ]Horizontal distributions of $O_3$ volume mixing ratios in the LS averaged from December 1994 to November 2013, in every season. Each 2D distribution is projected on the zonal axis below, with a meridional average of the northern extratropical zonal band (from 25°N to 75°N). The 2D (respectively 1D) distributions have a $5° \times 5°$ (respectively $5°$) resolution. The abscissa axes are the same in the maps and in the projections below. As explained in section 2.2.1, there is no data in the subtropical stratosphere.

## 3.2 Regional seasonal cycles

In order to further assess the regional variability of $O_3$ and CO mixing ratios, we analyzed the [..[202] ]time series of the eight regions displayed in Fig. 1 and defined in Tab. 1. We first compare the mean seasonal cycles, before characterizing and [..[203] ]analyzing the anomalies and then, derive the trends. In order to make a first estimate of the [..[204] ]inter-regional variability of

---

[202] removed: time-series

[203] removed: analysing

[204] removed: interregional

the two trace gases, the [..205 ]mean seasonal cycles are displayed in Fig. [..206 ]6. Similarly, we show the seasonal cycles of the O$_3$-to-CO ratio in order to provide a synthesis between the two data sets. First of all, as a support to our analysis, we present the seasonal cycles for the mean pressure at the 2 pvu altitude in Fig. 5.

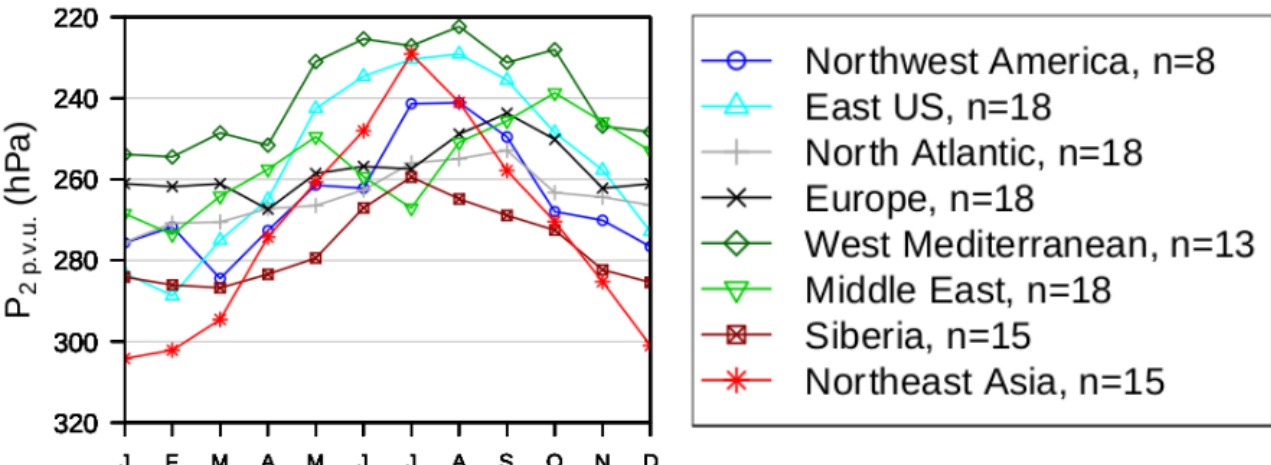

**Figure 5.** [..207 ] Mean seasonal cycles of P$_{2\,pvu}$ (hPa) in each region. The average has been done by selecting the same years than for the O$_3$ mixing ratio, which amount is indicated in the legend as the number n.

5 [..208 ]All regions exhibit a higher tropopause in summer. The northernmost region (Siberia) is characterized by the lowest tropopause altitude, whereas the higher tropopause altitudes occur in one of the southernmost regions (the Western Mediterranean basin).

---

205 removed: distributions of monthly means of mixing ratios during the whole monitoring period are shown

206 removed: **??**. Completary information is shown in Figs.7 and 8, representing the seasonal cycles of O$_3$ and CO respectively, for the monthly fifth percentile (P5), mean value and 95th percentile (P95).

208 removed: On the left panels, O$_3$ shows a similar seasonal cycle for all regions.

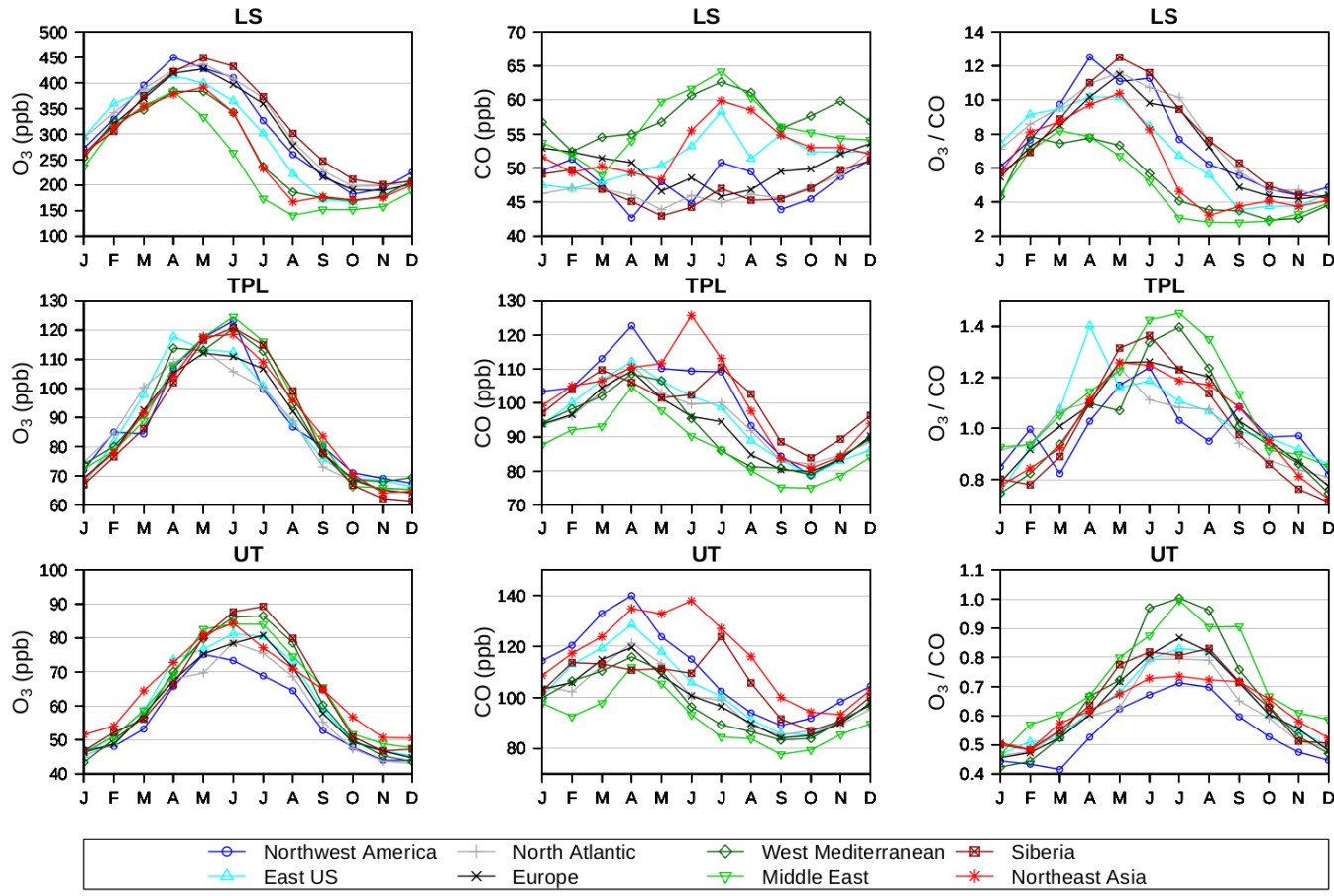

**Figure 6.** Mean seasonal cycles of $O_3$, CO and $O_3$/CO (from left to right) in the upper troposphere, tropopause layer and lower stratosphere (from bottom to top).

Complementary information is shown in Figs. 7, 8 and 9 representing the seasonal cycles of $O_3$, CO and $O_3$/CO respectively, for the monthly fifth percentile (P5), mean value and 95[th] percentile (P95). The interannual variability (IAV) is illustrated by the error bars, and defined as an interannual standard deviation of the monthly mean value. We first present the main characteristics in each of these three figures and their corresponding columns in Fig. 6, before analyzing and discussing the regional behaviours.

In Fig. 6, the mean $O_3$ mixing ratio (left column) shows a similar seasonal cycle for all regions. On average, the upper-tropospheric $O_3$ mixing ratio ranges from 46 ± 4 ppb in December–January up to 81 ± 7 ppb in June–July, while the lower-stratospheric $O_3$ mixing ratio ranges from 180 ± 20 ppb in October–November up to 410 ± 43 ppb in April–May, according to the following notation: mean ± IAV. The IAV indicated here (not shown in this figure) is averaged over the regions. The seasonal maximum generally takes place in [..[209] ]June–July in the UT, [..[210] ]May–June in the TPL and

---

[209]removed: June-July

[210]removed: May-June

[..211 ]April–May in the LS. This feature highlights a seasonal cycle in the TPL halfway between [..212 ]upper-tropospheric and lower-stratospheric cycles, and confirms again the tropopause definition used here as a realistic transition between the troposphere and the stratosphere. [..213 ]

The seasonal cycles of the mean CO mixing ratio (central column) show a broader range of variability [..214 ]between the regions. The five western regions ([..215 ]Northwest America, East United-States, North Atlantic, Europe and the Western Mediterranean basin) exhibit a spring maximum and a [..216 ]late summer–fall minimum in the UT and in the TPL. Asian regions ([..217 ]Siberia, Northeast Asia) present a different [..218 ]behaviour in the UT with a broad [..219 ]spring–summer maximum. Northwest America is noticeable [..220 ]for the highest mixing ratios recorded from November to April. Siberia shows a significant maximum in July. [..221 ]The relative seasonal amplitude – here defined as the $\frac{max - min}{mean}$ ratio – ranges from 33% in the Western Mediterranean basin up to 46% in Northwest America. In comparison, the seasonal variations of CO in the LS are substantially lower in most regions. In this layer, the relative seasonal amplitude reaches its maximum at 27% in Middle East, followed by substantially lower amplitudes at 22% in Northeast Asia and East United-States, and within the 16–18% range in the other regions.

[..222 ]

The $O_3$/CO mean value (right column) ranges between 0.4 and 1.0 in the UT with a clear maximum in summer for all the regions. It ranges between 0.7 and 1.5 in the TPL and shows discrepancies amongst the regions, mostly due to the inter-regional variability of the CO seasonal cycle. In the Western Mediterranean basin and Middle East, the maxima take place during June–July, similarly to the upper-tropospheric cycles. In Northwest America, Europe, Siberia and Northeast Asia, the maxima are shifted in May–June. In the North Atlantic Ocean and in East US, the maxima are steeper and occur during May and April respectively. This $O_3$/CO ratio varies from 3 up to 13 in the LS, with early maxima from February until May in East US, the Western Mediterranean basin and Middle East, whereas the maxima generally take place from April until June in the other regions. These differences in the LS arise from the higher summertime stratospheric CO mean values in East US, the Western Mediterranean basin and Middle East, and lower summertime stratospheric ozone mean values in the last two regions. The different ranges of the ratio between the three layers also confirm that our so-called

---

[211] removed: April-May

[212] removed: upper tropospheric and lower stratospheric

[213] removed: On the middle panels, seasonal cycles for CO

[214] removed: within

[215] removed: WNAm, EUS, NAt, Eur and WMed

[216] removed: fall

[217] removed: Sib, NEAs

[218] removed: behavior

[219] removed: spring-summer

[220] removed: with

[221] removed: In most regions, there is no noticeable seasonal variation in the LS. The right panel shows the pressure altitude of the 2 pvu isosurface characterizing the altitude of the tropopause. As expected, all regions present a higher tropopause in summer. Besides, it is interesting to note the north-south gradient of the tropopause height. Sib (WMed), the northernmost (one of the southernmost) region(s) is presenting the lowest (highest) tropopause altitude.

[222] removed: On average, the upper tropospheric $O_3$ VMR varies from 46 ppb in December-January to 81 ppb in June-July, while the upper tropospheric CO VMR varies from 87 ppb in September-October to 123 ppb in April. The lower stratospheric

tropopause layer is clearly a transition zone between the UT and the LS.

The $O_3$ [..[223] ]summertime maxima observed above most [..[224] ]regions in the UT are [..[225] ]driven by strong [..[226] ]photochemical activity, consistent with the $O_3$/CO seasonal cycles (same figure, bottom right panel). As shown in Fig. 7 (bottom panel), these $O_3$ maxima are significant for the whole monthly distribution. Stratmann et al. (2016) used the IAGOS-CARIBIC data set to compute a climatology for nitrogen oxides [..[227] ]and reactive nitrogen species ($NO_y$) from 2005 until 2013, with the same definition for the UT. The regions they defined as Europe and North Atlantic are the most sampled ones in the CARIBIC data set, which allows us to compare the seasonal cycles they derived in these regions with ours. Their study is based on larger regions than the ones we defined. Still, they derived [..[228] ]upper-tropospheric NO and $NO_y$ maxima well correlated with the $O_3$ maxima from our study. It is consistent with the leading role of photochemistry in the summertime $O_3$ maximum, at least in these two regions. Furthermore, Gressent et al. (2014) highlighted the influence of springtime and summertime lightning activity, [..[229] ]warm conveyor belts and convection over North America on the $O_3$ enhancement in the UT over North Atlantic and Europe from 2001 to 2005.

Beside these general features, the following subsections highlight [..[230] ]significantly different seasonal characteristics.

### 3.2.1 Northeast Asia, [..[231] ]Northwest America and Siberia

The CO seasonal cycle in the UT over [..[232] ]Northeast Asia is different from the others [..[233] ](Fig. 8). Among all the regions, [..[234] ]Northeast Asia shows the highest mixing [..[235] ]ratio during May–September. Its CO maximum ranges up to 140 ppb [..[236] ]and lasts from April to June, in contrast to the April climatological mean from most regions around 120 ppb. [..[237] ]It is characterized by the seasonal maximum in the 95[th] percentile during the same months. The high values in April are likely due to a strong wintertime CO accumulation in the lower troposphere (Zbinden et al., 2013), strong springtime agricultural fire emissions (Tereszchuk et al., 2013) and boreal biomass burning emissions [..[238] ](Andela et al., 2013). The high values in June can be associated with anthropogenic and biomass burning emissions coupled with geographically more frequent [..[239]

---

[223]removed: VMR varies from 180 ppb in October-November to 410 ppb in April-May, whereas the averaged lower stratospheric CO VMR does not exhibit any seasonal cycle. The

[224]removed: of the

[225]removed: likely to be

[226]removed: photochemistry. Indeed, Stratmann et al. (2016) used

[227]removed: (NOx) and their reservoir species (NOy

[228]removed: upper tropospheric NO and NOy

[229]removed: WCBs

[230]removed: some particular and local

[231]removed: western North

[232]removed: NEAs is very

[233]removed: .

[234]removed: NEAs

[235]removed: ratios during May-September. The CO maximum is

[236]removed: , compared

[237]removed: Its maximum lasts from April to June. It is driven by two peaks in the percentile 95 (see the Fig. 8), one in April and one in June. The peak in April

[238]removed: (?). The peak

[239]removed: WCBs

]warm conveyor belts (e.g. Madonna et al., 2014; Nédélec et al., 2005) and summertime Asian convection (Huang et al., 2016). A [..[240] ]peak is found in June in the TPL too, and in the LS to a lesser extent. High summertime CO [..[241] ]mixing ratios are also observed over eastern Asia by MLS observations at 215 hPa (Huang et al., 2016), slightly above the TPL.

The mean [..[242] ]upper-tropospheric CO in Siberia shows a leveling off from February [..[243] ]until June and peaks in July, like the [..[244] ]95th percentile. This last feature is consistent with [..[245] ]the Global Fire Emissions Database (GFED3: van der Werf et al., 2010) and the Global Fire Assimilation System (GFASv1.0: Kaiser et al., 2012) inventories for CO emissions from Asian boreal forest fires [..[246] ](Andela et al., 2013). Since the peak is noticeable in the TPL too, it [..[247] ]suggests a non-negligible impact of pyroconvection (Tereszchuk et al., 2013).

In contrast [..[248] ]to the other regions, the [..[249] ]upper-tropospheric $O_3$ seasonal cycle in [..[250] ]Northwest America does not exhibit a summertime maximum [..[251] ](Fig. 7). On the contrary, the seasonal pattern common with the other regions breaks at June and July, with mean values approximately 10 ppb lower on average. [..[252] ]Northeast Asia is also affected by an early decrease in July. These two Pacific-coast regions show an early decrease in the seasonal cycles of the [..[253] ]fifth percentile too, down to very low values in July: 32 ppb and 42 ppb respectively[..[254] ]. Meanwhile, the [..[255] ]fifth percentile averaged among the other regions reaches 53 ppb. This is consistent with the study of Zbinden et al. (2013), which highlights that the free troposphere above Los Angeles (118.17°W, 34.00°N) and visited Japanese airports (Osaka, Nagoya, Tokyo: 35°N, 138°E approximately) is influenced by a seasonal change in wind directions. During summer, the Asian monsoon and [..[256] ]the North American monsoon contribute to poor-$O_3$ air masses from [..[257] ]the subtropical Pacific Ocean. LiDAR measurements over TMF during summer in [..[258] ]2013–2014 also showed the influence of these air masses, notably between 9 and 12 km a.s.l. (Granados-Muñoz et al., 2017). It is worth noting that despite the common characteristics with TMF, the [..[259] ]Northwest America region is located above 40°N. Thus, in contrast [..[260] ]to TMF, Northwest America may not be impacted by the [..[261]

---

[240]removed: similar
[241]removed: VMRs
[242]removed: upper tropospheric CO in Sib.
[243]removed: to
[244]removed: P95 (see the Fig. 8). This
[245]removed: GFED and GFAS inventories that show a peak during July
[246]removed: (?)
[247]removed: may suggest a non negligible
[248]removed: with
[249]removed: upper tropospheric
[250]removed: WNAm
[251]removed: .
[252]removed: NEAs
[253]removed: P5
[254]removed: (Fig. 7).
[255]removed: P5
[256]removed: northern
[257]removed: subtropical Pacific ocean
[258]removed: 2013-2014
[259]removed: WNAm
[260]removed: with TMF, WNAm
[261]removed: rich-NOx

]rich-NO$_x$ air masses originating from Central America (Cooper et al., 2009, Fig. 7), where a strong summertime lightning activity and the North American monsoon anticyclone allow the buildup of a recurrent maximum in O$_3$ in the UT (e.g. Cooper et al., 2007). Cooper et al. (2007 and 2009) also showed the impact of mid-latitudinal Eurasian emissions on [..[262] ]free-tropospheric O$_3$ above several American sites, potentially representative of the northern part of [..[263] ]the Northwest America region. This could explain the fact that [..[264] ]the summertime 95$^{th}$ percentile in O$_3$ (Fig. 7) is similar to [..[265] ]the other regions, despite the absence of nearby sources upwind and the more frequent clean tropical air masses. It is consistent with the strong [..[266] ]95$^{th}$ percentile for CO in [..[267] ]Northwest America (Fig. 8), correlated with the [..[268] ]one from Northeast Asia, suggesting that these two regions have common upwind strong emissions. It is weaker in [..[269] ]Northwest America than in Northeast Asia, highlighting the decrease of CO mixing ratio during the long-range transport, probably due to dilution and photochemistry. In Fig. 9, the springtime and summertime O$_3$/CO ratios in the UT exhibit lower values (0.6–0.7) in these two Pacific-coast regions compared to the other locations (at least 0.1 higher). It is mainly characterized by a lower fifth percentile (0.4), consistent with a higher frequency of high-CO and poor-O$_3$ air masses. However, in contrast to ozone, the seasonal cycles of the O$_3$/CO ratio do have a summer maximum. The interruption observed in the O$_3$ seasonal cycles thus remains characterized by the summer maximum in photochemical activity, despite the fast change in the monthly air composition.

Interestingly, in the UT, [..[270] ]Northwest America shows one of the lowest mean springtime O$_3$ mixing ratios too, rather linked with the upper values (see the low P95 during [..[271] ]March–April in Fig. 7), whereas its mean springtime CO is the highest (Fig. 8). This last feature is characterized by a distribution shifted toward the upper values, compared to the other regions, as for the O$_3$/CO ratio. The lower P95 in O$_3$ associated with the stronger P5 in CO during [..[272] ]February–April likely indicates that [..[273] ]Northwest America is less frequently impacted by rich-O$_3$ air masses, and less frequently impacted by poor-CO air masses. [..[274] ]Lower-stratospheric O$_3$ also exhibits the strongest P5 in [..[275] ]Northwest America, amongst all the regions. Since this feature only takes place during [..[276] ]March–April, it is unlikely driven only by the aircraft flight level, relative to the tropopause. Furthermore, the P$_{2\,pvu}$ averages shown in Fig. 5 are not at lower altitudes during April. Con-

---

[262]removed: free tropospheric

[263]removed: WNAm

[264]removed: summertime P95

[265]removed: polluted

[266]removed: maxima of P95

[267]removed: WNAm

[268]removed: ones from NEAs. These correlated maxima in April and June suggest

[269]removed: WNAm than in NEAs

[270]removed: WNAm

[271]removed: February-April

[272]removed: February-April

[273]removed: WNAm

[274]removed: Lower stratospheric

[275]removed: WNAm

[276]removed: March-April

sequently, the stronger P5 in [..277 ]Northwest America suggests less frequent tropospheric air masses in the LS during [..278 ]March–April. All these features [..279 ]indicate less efficient springtime stratosphere–troposphere exchange in Northwest America.

### 3.2.2  The **Western Mediterranean basin and [..280 ]Middle East**

As expected in Fig. 7 (Fig. 8), the mean $O_3$ (CO) concentrations in the LS are lower (higher) in [..281 ]the Western Mediterranean basin, Middle East and Northeast Asia, which are the southernmost regions of this study. Indeed, as the dynamical tropopause is generally higher there (Fig. [..282 ]5), although the flights are classified LS, they are likely to sample air masses closer to the tropopause[..283 ]: the substantial difference in $O_3$ mixing ratios is thus explained by its strong vertical gradient
in the stratosphere. [..284 ]During July–August, three categories of regions can be established depending on the width of the monthly distribution of the $O_3$/CO ratio. Referring to Fig. 9, the northernmost regions (Northwest America, North Atlantic, Europe, Siberia) exhibit a higher $O_3$/CO ratio for all three metrics (mean value, fifth and 95th percentiles). On the opposite, the southernmost regions (the Western Mediterranean basin, Middle East and Northeast Asia) show a lower ratio, for all three metrics again. The remaining region (East United-States) shows a strong intra-monthly variability with
a low fifth percentile, an intermediate mean value and a high 95th percentile. These categories thus correspond respectively to regions mostly impacted by extratropical air masses, to regions strongly impacted by subtropical air masses and to the region influenced by both extratropical and subtropical air masses. In June, Northeast Asia belongs to this intermediate category with its low fifth and high 95th percentiles, before reaching the subtropically influenced regions in July.

In summer, CO mixing ratios are similar over [..285 ]the Western Mediterranean basin and Middle East, but Middle East $O_3$ is significantly lower. This feature is consistent with the mixing ratios derived from [..286 ]OMI–MLS observations at 150 hPa presented in Park et al. (2007) and at 100 hPa in Park et al. (2009). In southern Asia, polluted surface air masses are uplifted by deep convection during the Asian Monsoon, up to the tropical UT. The western part of the anticyclonic circulation then transports poor-$O_3$ air masses northward and horizontally (Barret et al., 2016). This may impact the LS in [..287 ]Middle
East (Park et al., 2007), consistent with CH[..288 ]4 measurements from IASI and AIRS[..289 ], coupled with modelling us-

---

277removed: WNAm

278removed: March-April

279removed: seem to indicate less frequent springtime STE events in WNAm

280removed: Middle-East

281removed: WMed, MidE and NEAs

282removed: **??**, top right panel),

283removed: , classified LS but closer to the lower boundary

284removed: However, it does not explain the difference between MidE and the two other regions during summer. At this time of the year

285removed: WMed and MidE, but MidE

286removed: OMI-MLS

287removed: MidE

288removed: 4

289removed: and modelling with the CTM MOCAGE and the CCMs

ing the CNRM-AOCCM[..²⁹⁰ ], LMDz-OR-INCA chemistry climate models and the MOCAGE chemistry transport model (Ricaud et al., 2014). The summertime [..²⁹¹ ]lower-stratospheric CO mixing ratio is comparable between [..²⁹² ]the Western Mediterranean basin, Middle East and Northeast Asia. The last one is impacted by frequent [..²⁹³ ]warm conveyor belts coupled with a strong fire activity (Madonna et al., 2014; Jiang et al., 2017).

5   In the UT, there is more $O_3$ and less CO in [..²⁹⁴ ]the Western Mediterranean basin and Middle East. Figure 9 better illustrates the distinction of these two regions from the others, with a high $O_3$/CO ratio. Its mean value reaches 0.95 in July in the two regions in the UT, and is both characterized by higher fifth and 95$^{th}$ percentiles. Note that $O_3$/CO is also higher in the TPL ($\simeq 1.5$). All these features show the impact of the stronger summertime subsidence at these latitudes on the whole monthly distribution in the UT. In Middle East particularly, Etesian winds interact with the Asian Monsoon Anticyclone

10  (AMA), enhancing the subsidence of high-level air masses (e.g. Tyrlis et al., 2013) thus allowing a recurrent summertime $O_3$ pool down to the mid-troposphere (Zanis et al., 2014).

---

²⁹⁰removed: and
²⁹¹removed: lower stratospheric CO VMR
²⁹²removed: WMed, MidE and NEAs
²⁹³removed: WCB
²⁹⁴removed: WMed and MidE, thus characterizing the stronger summertime subsidence at these latitudes. In MidE

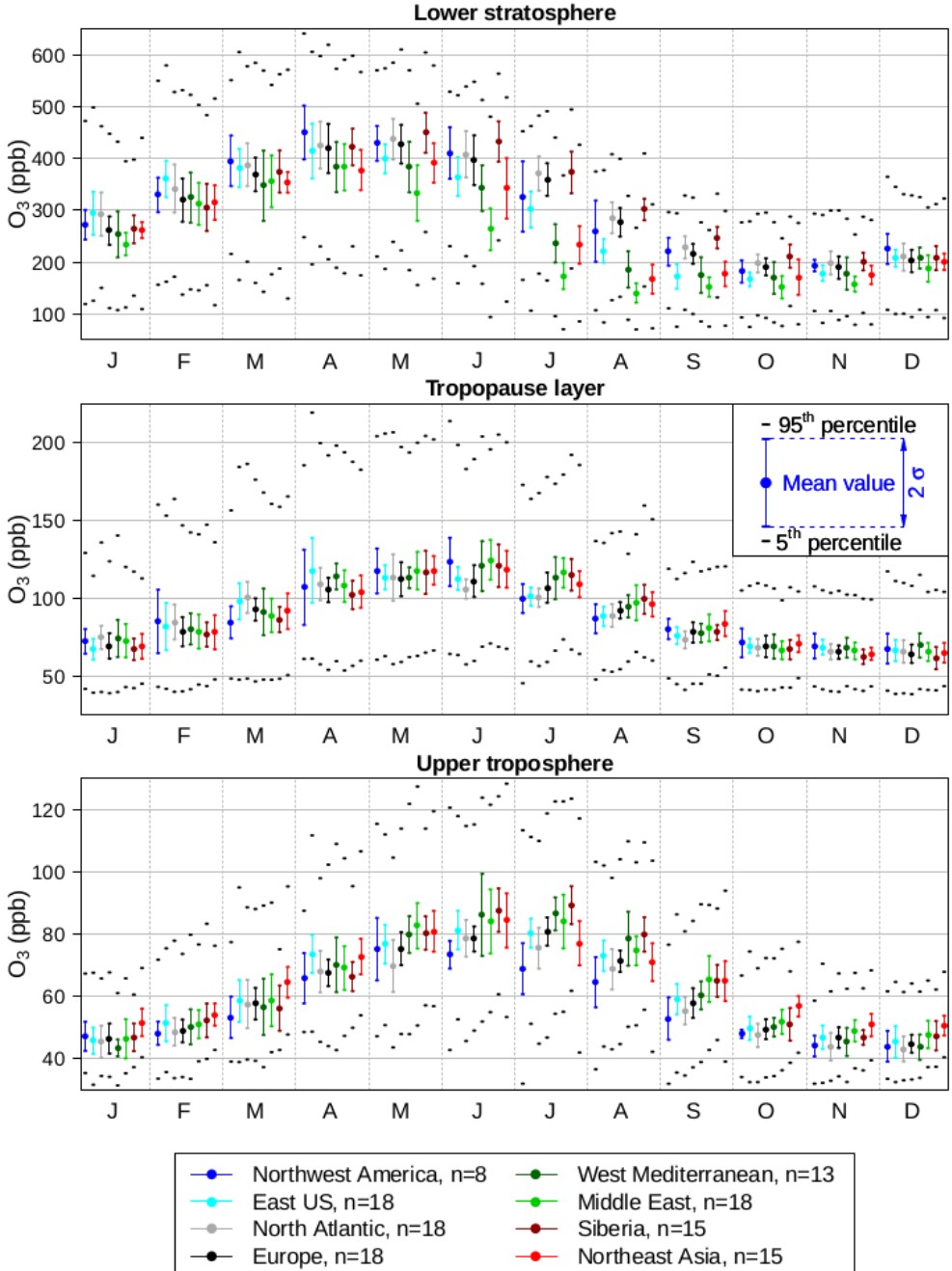

**Figure 7.** [..295 ]Seasonal cycles of $O_3$ for the monthly mean value (coloured points), the fifth and 95th percentiles (lower and upper black ticks respectively). The interannual variability (coloured error bars) corresponds to the interannual standard deviation of the monthly mean value $\sigma$. From bottom to top, the graphics represent the cycles in the upper troposphere, tropopause layer and lower stratosphere. The amount of years taken into account in the calculation of the upper-tropospheric $O_3$ cycles is indicated in the bottom legend as the number n.

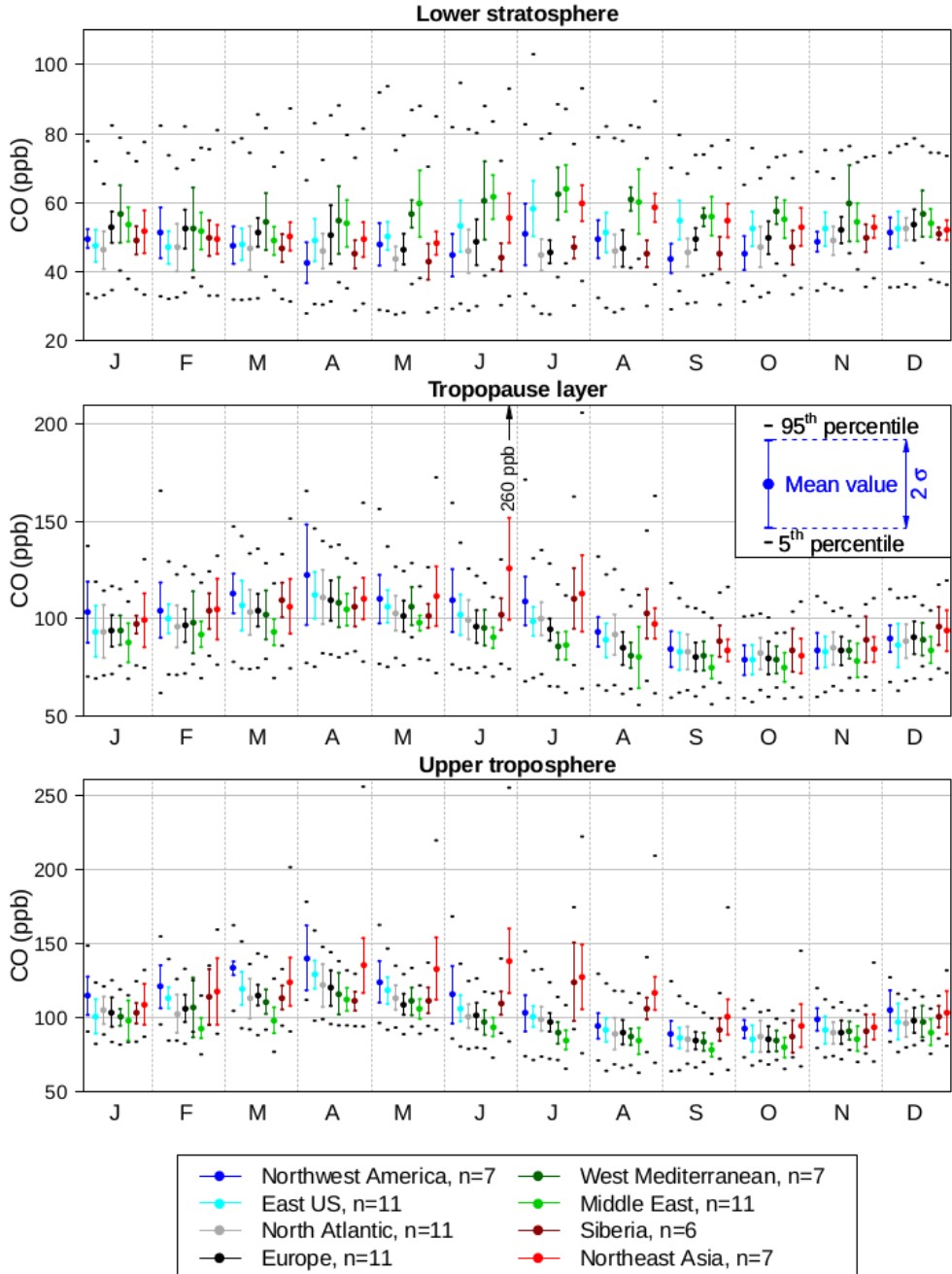

**Figure 8.** [..296]Seasonal cycles of CO for the monthly mean value (coloured points), the fifth and 95th percentiles (lower and upper black ticks respectively). The interannual variability (coloured error bars) corresponds to the interannual standard deviation of the monthly mean value $\sigma$. From bottom to top, the graphics represent the cycles in the upper troposphere, tropopause layer and lower stratosphere. The amount of years taken into account in the calculation of the upper-tropospheric CO cycles is indicated in the bottom legend as the number n.

**Figure 9.** Seasonal cycles of the $O_3$/CO ratio for the monthly mean value (coloured points), the fifth and $95^{th}$ percentiles (lower and upper black ticks respectively). The interannual variability (coloured error bars) corresponds to the interannual standard deviation of the monthly mean value $\sigma$. From bottom to top, the graphics represent the cycles in the upper troposphere, tropopause layer and lower stratosphere. The amount of years taken into account in the calculation of the upper-tropospheric $O_3$/CO cycles is indicated in the bottom legend as the number n.

## 3.3 Trends: comparison, [..[297] ]characterization

In this section, annual and seasonal trends of $O_3$ and CO are investigated in the different regions (see Sect. 2.2.2 for details on the calculations of trends), from August 1994 until December 2013 for $O_3$ and from December 2001 until December 2013 for CO. [..[298] ]For a more complete description of the data set, the corresponding time series for both species are given for the three layers in Figs. B1–B6 in Appendix.

[..[299] ]

[..[300] ]

[..[301] ]

[..[302] ]

[..[303] ]

[..[304] ]

[..[305] ]As shown in these figures, no monthly data is available during most of 2010, due to an interruption of measurements. Most regions have a good sampling efficiency, with about [..[306] ]70–97% of available data (see Tab. 1). Specifically, the lower sampling frequency in [..[307] ]Northwest America (top-left panel) allows [..[308] ]relatively few available monthly data. For $O_3$, the sampling focuses on the period [..[309] ]2003–2009. Since less than 60% of monthly data are available, and since the available months are especially representative of the second half of the monitoring period, we cannot take it into account in the study of the 20-year trends. After 2009, the sampling is efficient only during winter, such that we can only compute the trend of wintertime CO in this region. The sampling is also too low for CO in [..[310] ]the Western Mediterranean basin, in

---

[297] removed: characterisation

[298] removed: First, we present the corresponding time series for both species in the three layers in Figs. B1 – B6.

[299] removed: $O_3$ monthly mean values in the UT (thin curve). The interval from fifth and 95th percentiles is filled in red. The yearly mean values are represented by the bold curve and the black points.

[300] removed: $O_3$ monthly mean values in the TPL (thin curve). The interval from fifth and 95th percentiles is filled in green. The yearly mean values are represented by the bold curve and the black points.

[301] removed: $O_3$ monthly mean values in the LS (thin curve). The interval from fifth and 95th percentiles is filled in blue. The yearly mean values are represented by the bold curve and the black points.

[302] removed: CO monthly mean values in the UT (thin curve). The interval from fifth and 95th percentiles is filled in red. The yearly mean values are represented by the bold curve and the black points.

[303] removed: CO monthly mean values in the TPL (thin curve). The interval from fifth and 95th percentiles is filled in green. The yearly mean values are represented by the bold curve and the black points.

[304] removed: CO monthly mean values in the LS (thin curve). The interval from fifth and 95th percentiles is filled in blue. The yearly mean values are represented by the bold curve and the black points.

[305] removed: As it can be seen

[306] removed: 70-97

[307] removed: WNAm

[308] removed: relative

[309] removed: 2003-2009

[310] removed: WMed

the LS (55%). In Siberia, Northeast Asia and [..³¹¹ ]the Western Mediterranean basin (excluding the LS for this region), the sampling is moderate for CO (between 60 and 70%) but data are available both at the beginning and at the end of the period, which makes [..³¹² ]trends calculations still relevant.

For additional information, the monthly relative anomalies we used for the [..³¹³ ]trends calculations are shown in Figs. C1–

5 C4.

In the UT, the anomalies have several common features: a negative anomaly until 1997, followed by a leveling off [..³¹⁴ ]during 2000–2009 and then by a positive anomaly since 2011. The first period ([..³¹⁵ ]1994–1997) has already been associated with the multi-year ozone recovery since the eruption of Mt Pinatubo in 1991: this event has been an important source of particulate matter in the lower stratosphere at global scale, contributing to the ozone depletion in the UT, TPL and LS (e.g. Tang et al.,

2013). The positive anomaly in 1998 has been referenced as a consequence of the extreme El Niño event in 1997, as explained in Thouret et al. (2006) and Koumoutsaris et al. (2008).

The LS does not show such a positive anomaly at the end of the period, nor the same [..³¹⁶ ]interannual variability. We can note the positive anomalies in 2006 in [..³¹⁷ ]Northwest America, Europe and Siberia. It seems consistent with the downward 380 K ozone flux anomalies shown in Olsen et al. (2013, Fig. 5).

**3.3.1 Trends in O$_3$**

O$_3$ trends are summarized in Figs. 10 and 11. Figure 10 shows the trends of monthly fifth, mean values and [..³¹⁸ ]95$^{th}$ percentiles for O$_3$, on the left, central and right panels respectively. Figure 11 shows the trends of the O$_3$ mean value in the UT, with respect to the [..³¹⁹ ]season. In parallel, the absolute trends are reported in Tabs. D1 and D2 with their corresponding [..³²⁰ ]confidence intervals, in Appendix D. Note that all confidence intervals in this paper are given at the 95% level. For a better

precision, we use the values reported in this table to comment the following figures. The relative trends (in %.yr$^{-1}$) are also reported in Tabs. D3 and D4 in order to facilitate the comparison with other [..³²¹ ]data sets. For clarity, only the significant trends at [..³²² ]the 95% confidence level are reported.

At the annual scale, mean [..³²³ ]upper-tropospheric O$_3$ is significantly increasing in the seven regions except [..³²⁴ ]North Atlantic, with trend estimates ranging from +0.25 up to +0.45 ppb.yr$^{-1}$. The strongest and most significant trend is found in

---

[311] removed: WMed

[312] removed: trend

[313] removed: calculation of the trends

[314] removed: (2000 – 2009) and

[315] removed: 1994-1997

[316] removed: IAV

[317] removed: WNAm, Eur and Sib

[318] removed: 95th

[319] removed: seasons

[320] removed: 95%-confidence

[321] removed: datasets

[322] removed: a

[323] removed: upper tropospheric

[324] removed: NAt

[..325 ]Northeast Asia, with +0.45 [0.23; 0.68] ppb.yr$^{-1}$. The least significant trend concerns [..326 ]Middle East (+0.25 [0.01; 0.45] ppb.yr$^{-1}$). The trends in the TPL are similar to the [..327 ]upper-tropospheric trends, albeit stronger and generally with a better significance. On the other hand, all mean O$_3$ trends remain insignificant in the LS.

[..328 ]On one hand, quite similar results are obtained for the O$_3$ fifth percentile in the UT and TPL. Again, the strongest increase in the UT is observed in [..329 ]Northeast Asia (+0.57 [0.35; 0.80] ppb.yr$^{-1}$), closely followed by [..330 ]Siberia (+0.50 [0.31; 0.67] ppb.yr$^{-1}$). The Atlantic regions ([..331 ]East United-States, North Atlantic and Europe) exhibit an intermediate significance, with a stronger increase at approximately +0.45 ppb.yr$^{-1}$ [0.20; 0.65] in [..332 ]East United-States and North Atlantic. The small southern regions ([..333 ]the Western Mediterranean basin and Middle East) show the lowest and least significant increase at +0.34 [0.08; 0.52] and +0.30 [0.12; 0.48] ppb.yr$^{-1}$ respectively. The main difference with the [..334 ]O$_3$ [..335 ]mean value is found in the LS where a significant positive trend [..336 ]occurs in a few regions (North Atlantic, Siberia, Northeast Asia). [..337 ]On the other hand, the upper-tropospheric P95 increases significantly in [..338 ]Europe and the Western Mediterranean basin only, at +0.29 [0.06; 0.47] ppb.yr$^{-1}$ and +0.64 [0.06; 1.16] ppb.yr$^{-1}$ respectively, with p-values higher than 0.02 indicating a low significance level. In the TPL, the increase is observed in [..339 ]East United-States, Europe and Siberia, while no significant trend is found in the LS. Therefore, the general positive trend of mean O$_3$ mixing ratios in both the UT and TPL is associated with an increase of the background values (represented by the fifth percentile). [..340 ]The Western Mediterranean basin may be an exception with its strong increase of P95, contributing as well to the strong positive trend [..341 ]in the mean value.

The O$_3$ seasonal trends in the UT are shown in Fig. 11. As a main result, we note that each region shows a significant positive trend only during specific seasons. A significant increase of $\sim$0.5 ppb.yr$^{-1}$ is observed in the Atlantic regions ([..342 ]East United-States, North Atlantic and Europe) in winter, with a non-negligible significance in [..343 ]North Atlantic and Europe. Only Siberia shows a significant increase during spring (0.50 ppb.yr$^{-1}$), as [..344 ]Northeast Asia during summer

---

325removed: NEAs

326removed: the MidE region

327removed: upper tropospheric

328removed: Quite

329removed: NEAs

330removed: Sib

331removed: EUS, NAt and Eur

332removed: EUS and NAt

333removed: WMed and MidE

334removed: mean

335removed: results

336removed: is found

337removed: On the other hand, the upper tropospheric

338removed: Eur and WMed

339removed: EUS, Eur and Sib

340removed: WMed

341removed: of

342removed: EUS, NAt and Eur

343removed: NAt and Eur. Only Sib

344removed: NEAs

(0.84 ppb.yr$^{-1}$). Most of the regions show a significant increase during fall. The fifth percentile of O$_3$ mixing ratios is increasing in most regions during winter and spring. The strongest P05 trend is observed during summertime in [..[345] ]Northeast Asia (+1.07 [0.58; 1.42] ppb.yr$^{-1}$). Concerning the [..[346] ]95$^{th}$ percentile of O$_3$, all trends remain insignificant, except in Europe during winter (+0.43 [0.10; 0.88] ppb.yr$^{-1}$).

The upper-tropospheric O$_3$ trends are also computed over several periods, in order to test the sensitivity of our results to the start and the end of the monitoring period. They are shown in Fig. 12 for the three metrics (mean value, fifth and 95$^{th}$ percentiles). The trends over the whole period are compared to the ones computed over 1994–2008, thus excluding the last three years positive anomalies, and to the ones computed over 2002–2013, i.e. the monitoring period for CO.

Removing the last years positive anomaly leads to the loss of the 2-$\sigma$ statistical significance for most significant trends, in three regions for the fifth percentile (North Atlantic, the Western Mediterranean basin, Middle East), four regions for the mean value (East United-States, the Western Mediterranean basin, Middle East, Siberia), and the two only regions for the 95$^{th}$ percentile (Europe, the Western Mediterranean basin). Consequently, the significant increase of the mean O$_3$ mixing ratio shown in Fig. 10 is robust for Europe and Northeast Asia, and the significant increase in the fifth percentile

is robust also in East US and Siberia. In the other cases, the 2-$\sigma$ confidence interval is strongly sensitive to the higher ozone values during 2001–2013.

On the 2001–2013 period (bottom central panel in Fig. 12) the statistical significance for the mean values is stronger in Middle East and Siberia, and weaker elsewhere compared to the full period. The regions where it decreases also show a weakening in the significance of the fifth percentile. The trends in 95$^{th}$ percentiles (right panels) appear to be limited

by the first years. Thus, it shows a transition between the first part and the last part of the monitoring period, highlighted by a change in the trends in the monthly distribution. The upper-tropospheric O$_3$ trends are higher during 2002–2013 because of the positive anomaly at the end of this period. One possible explanation for this anomaly lies in an enhanced transport across the tropopause caused by the 2009–2010 strong El Niño event, as shown in Lin et al. (2015). Responses in tropospheric ozone at the midlatitudes to the El Niño Southern Oscillation (ENSO) have been identified in Wespes et al.

(2017) with a 4-month or 6-month time lag. Further modeling studies are needed to assess the link between the chemical composition of the upper troposphere and the ENSO.

It is worth noting that besides the IAGOS database, in situ observations are very sparse in the UTLS and [..[347] ]their time coverage is usually too short for determining long-term trends. This considerably limits our ability to compare these results to the literature. [..[348] ]As mentioned in Sect. 1, the [..[349] ]IAGOS measurements during 1994–2003 showed significant positive

trends for ozone mixing ratios above [..[350] ]East United-States, North Atlantic and Europe in the UT (respectively: +0.99 $\pm$

---

[345]removed: NEAs

[346]removed: 95th

[347]removed: usually have a time coverage which is

[348]removed: As mentionned

[349]removed: first 9 years of IAGOS measurements

[350]removed: eastern United States, northern

0.82 %.yr$^{-1}$, +1.12 ± 0.86 %.yr$^{-1}$ and +1.00 ± 0.90 %.yr$^{-1}$) and the LS (+1.54 ± 1.37 %.yr$^{-1}$ above [..$^{351}$ ]North Atlantic and +1.99 ± 1.21 %.yr$^{-1}$ above Europe) (Thouret et al., 2006). With a new decade of observations, our results suggest that the previous trends in the LS were influenced by the low values since 1994 until 1997, following [..$^{352}$ ]the Pinatubo eruption. More recently, Petetin et al. (2016) used IAGOS measurements to study vertical profiles of O$_3$ and CO above Frankfurt and

5  Munich airports, during similar periods. As in Thouret et al. (2006), they defined the UT with the same method. In this layer, they found a barely significant increase of O$_3$ during winter. At the annual scale, they derived a significant positive trend of P5, consistent with our results. However, they did not observe the [..$^{353}$ ]99%-significant positive trend of P95 that we derived [..$^{354}$ ]($+0.4$ ppb.yr$^{-1}$, p $= 8.10^{-3}$). Since we compute the trends with the same methodology, the discrepancies are due to the greater size of our Europe region, which allows a higher amount of data and thus a more significant statistical analysis.

In terms of O$_3$ trends in the free troposphere (rather than the UT), many studies highlighted increasing mixing ratios over time. Based on a combined data set (MOZAIC, research aircraft, ozonesondes, LiDAR) over western North America ([..$^{355}$ ]25–55°N, [..$^{356}$ ]130–90°W), Cooper et al. (2010) derived an increase of springtime O$_3$ ([..$^{357}$ ]April–May) in the UT over the period [..$^{358}$ ]1995–2008 (+0.58±0.52 ppb.yr$^{-1}$ for the median O$_3$). Over the period [..$^{359}$ ]2000–2015, Granados-Muñoz and

15  Leblanc (2016) derived a positive trend from LiDAR measurements above [..$^{360}$ ]TMF at 7–10 km, with +0.31±0.30 ppb.yr$^{-1}$ for the median (the trend of the [..$^{361}$ ]95$^{th}$ percentile being +0.55±0.60 ppb.yr$^{-1}$, thus [..$^{362}$ ]significant at the 90% confidence level [..$^{363}$ ]only). Based on ozonesondes over the period [..$^{364}$ ]1995–2008, Logan et al. (2012) did not observe any significant increase of O$_3$ mixing ratios between 400 and 300 hPa in Europe. The difference with our results [..$^{365}$ ]can be partly explained by the fact that the period studied by Logan et al. (2012) does not extent to 2011[..$^{366}$ ], the beginning of the strong positive

20  anomaly in upper-tropospheric O[..$^{367}$ ]$_3$, according to the IAGOS data set (see Fig. B1).

---

$^{351}$removed: Northern

$^{352}$removed: Pinatubo's eruption.

$^{353}$removed: significant

$^{354}$removed: . Since we compute the trends with the same methodology, the discrepancies are due to the greater size of our region Eur, which allows us a more frequent sampling, thus smoothing the temporal variability. We assume that in the UT, the spatial variability is too weak to be responsible for the discrepancies between the two studies.

$^{355}$removed: 25-55

$^{356}$removed: 130-90

$^{357}$removed: April-May

$^{358}$removed: 1995-2008 (with trends of

$^{359}$removed: 2000-2015

$^{360}$removed: Table Mountain Facility (35°N, 119°W, Jet Propulsion Laboratory, California) at 7-10

$^{361}$removed: 95th

$^{362}$removed: insignificant at a 95

$^{363}$removed: ). Although no trend calculation was performed in WNAm, it may confirm the upper tropospheric O$_3$ increase as effective in the whole northern mid-latitudes, except the Pacific ocean.

$^{364}$removed: 1995-2008

$^{365}$removed: may

$^{366}$removed: when a strong positive

$^{367}$removed: $_3$ anomaly in the UT is observed in the

Free and [..[368] ]upper-tropospheric ozone trends derived from various instruments were reviewed in Cooper et al. (2014). Significant positive trends over the period [..[369] ]1971–2010 are reported with ozonesonde data in the free troposphere above Western Europe and Japan, ranging at 0.1[..[370] ]–0.3 ppb.yr$^{-1}$. As [..[371] ]mentioned in Sect. 1, the significant positive trends measured at the highest [..[372] ]Northern Hemispheric GAW stations (above 2 km a.s.l.: Zugspitze, Rocky Mountain National

Park, Jungfraujoch, Mt-Bachelor Observatory, Mt-Waliguan) rise between $0.05 \pm 0.04$ ppb.yr$^{-1}$ and $0.33 \pm 0.05$ ppb.yr$^{-1}$. Only the most increasing trends (Rocky Mountain and Jungfraujoch) are within the range of the ones derived in the present study. Among the GAW stations [..[373] ]mentioned in Sect. 1, the other ones show a smaller increase. In the free troposphere at 510 hPa, although on a shorter timescale (2005–2010), Neu et al. (2014a) derived a positive trend of $\simeq +$[..[374] ]1 %.yr$^{-1}$ in the [..[375] ]30–50°N zonal band[..[376] ], based on TES measurements. Thus, the increase of the O$_3$ mixing ratio is [..[377]

]probably more representative of the middle and upper altitude ranges of the free troposphere than its lower part. As said previously, the trends in the TPL are similar to the [..[378] ]upper-tropospheric ones. Since no [..[379] ]lower-stratospheric trend is significant at the annual scale, we cannot make any conclusion about the evolution of [..[380] ]lower-stratospheric ozone. Our results alone do not allow us to explain these trends in the TPL. However, several studies reported a global significant increase of the tropopause height during the last decades (-0.05 hPa.yr$^{-1}$, $p < 2.10^{-16}$ over the period [..[381] ]1979–2011 with

ERA-Interim reanalyses in Škerlak et al. (2014); see also Gettelman et al., 2011 and references therein). Above the regions of this study, the typical vertical gradient of O$_3$ observed in the [..[382] ]four upper flight levels ([..[383] ]where the gradient is the strongest amongst our data) is about 1 ppb.hPa$^{-1}$ (not shown). According to this yearly mean value, the elevation of the TPL would induce an ozone increase by an order of magnitude below the observed trends. [..[384] ]Such an increase of the TPL [..[385] ]height is thus unlikely to explain a major part of the positive trends in the TPL.

Gettelman et al. (2010) combined a multimodel analysis to derive O$_3$ positive trends near 0.4%.yr$^{-1}$ over the century in the extratropical UTLS (Ex-UTLS), relative to the tropopause. It is similar with the trends we derive in the UT, from 0.39%.yr$^{-1}$

---

[368] removed: upper tropospheric

[369] removed: 1971 – 2010

[370] removed: -0.3

[371] removed: mentionned in section

[372] removed: northern hemispheric

[373] removed: mentionned

[374] removed: 1.1 %

[375] removed: 30-50

[376] removed: based on the Tropospheric Emission Spectrometer (TES ) satellite measurementsperformed over the period 2005-2010, although on a shorter timescale

[377] removed: likely to be

[378] removed: upper tropospheric

[379] removed: lower stratospheric

[380] removed: lower stratospheric ozone. Thus, our

[381] removed: 1979-2011

[382] removed: 4

[383] removed: thus the stronger part of the gradient

[384] removed: Thus, such

[385] removed: is not likely

in [..386 ]Middle East to 0.69%.yr$^{-1}$ in [..387 ]Northeast Asia (see Tab. D3). Their simulations did not show any dependence on tropospheric hydrocarbon chemistry. They concluded to an enhanced [..388 ]Brewer–Dobson circulation, which [..389 ]can contribute to the O$_3$ trends we observed in the UT and the TPL. This might explain the springtime positive trends we found in the TPL of [..390 ]North Atlantic, Europe, Siberia and Northeast Asia, but these trends do not occur specifically during spring.

5  Furthermore, only [..391 ]Siberia shows a springtime significant increase in the UT.

The impact of the sampling density on the observed long-term evolution has been tested by computing the trends of the [..392 ]three most sampled regions with the time coverage of the [..393 ]less sampled regions. With the sampling periods of [..394 ]the Western Mediterranean basin and Siberia, the changes for [..395 ]East United-States, North Atlantic and Europe were

10  small compared to the confidence intervals (0.1%.yr$^{-1}$ or less). With the sampling periods of [..396 ]Northeast Asia, these trends tend to be higher, but the difference remains insignificant ($\simeq 0.2\%.\text{yr}^{-1}$).

---

[386] removed: MidE

[387] removed: NEAs

[388] removed: Brewer-Dobson

[389] removed: may subsequently

[390] removed: NAt, Eur, Sib and NEAs

[391] removed: Sib

[392] removed: most 3

[393] removed: lowest

[394] removed: WMed and Sib

[395] removed: EUS, NAt and Eur

[396] removed: NEAs

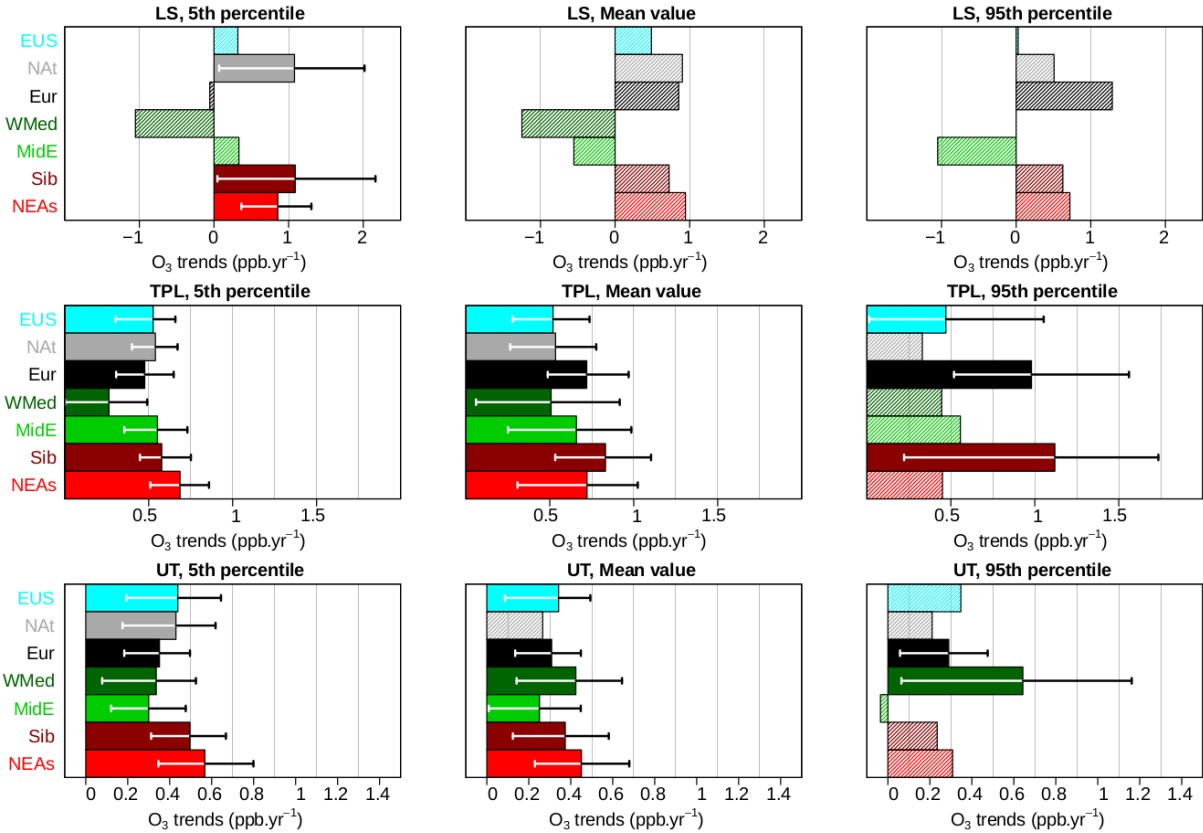

**Figure 10.** O$_3$ trends (ppb.yr$^{-1}$) in the UT, TPL and LS (from bottom to top), for the monthly fifth percentile (left [..[397] ]column), mean value (central [..[398] ]column) and [..[399] ]95[th] percentile (right [..[400] ]column). For each significant trend, the error bars represent the 95% confidence interval. The insignificant trends are represented by hatched areas. The coloured labels correspond to the regions defined in Fig. 1.

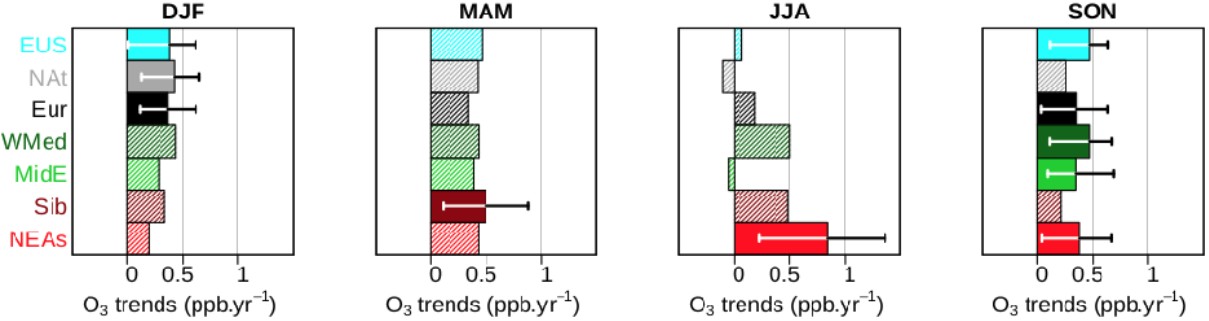

**Figure 11.** O$_3$ trends (ppb.yr$^{-1}$) in the UT, in each season. For each significant trend, the error bars represent the 95% confidence interval. The insignificant trends are represented by hatched areas. The coloured labels correspond to the regions defined in Fig. 1.

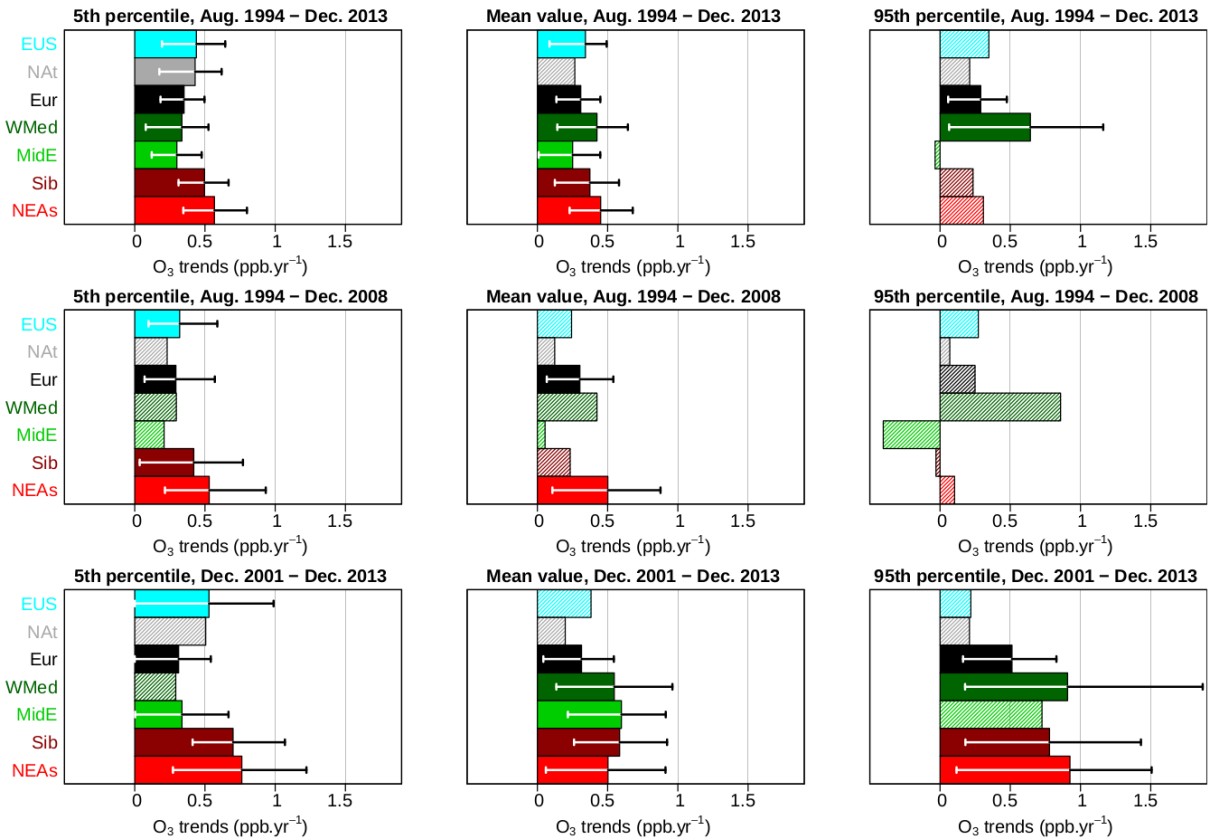

**Figure 12.** $O_3$ trends (ppb.yr$^{-1}$) in the UT, over the whole monitoring period (1994–2013: top panels), over 1994–2008 (central panels) and over the monitoring period for CO (2002–2013: bottom panels), for the monthly fifth percentile (left column), mean value (central column) and 95$^{th}$ percentile (right column). For each significant trend, the error bars represent the 95% confidence interval. The insignificant trends are represented by hatched areas. The coloured labels correspond to the regions defined in Fig. 1.

### 3.3.2 Trends in CO

CO trends are summarized in Figs. 13 and 14. As for $O_3$, more precise values are given in Tabs. D5 and D6 with their corresponding 95%-confidence intervals, followed by the equivalent relative trends in Tabs. D7 and D8, in the Appendix D. Figure 13 shows the trends of the fifth, mean values and [..[401] ]95$^{th}$ percentile for monthly CO data on the left, central and right panels respectively. Figure 14 shows the trends of the CO mean value in the UT, with respect to the [..[402] ]season. As discussed at the beginning of Sect. 3.3, we do not take the trend in the LS of [..[403] ]the Western Mediterranean basin into account. Similarly,

---

[401]removed: 95th

[402]removed: seasons

[403]removed: WMed region

only wintertime trends are taken into account in [..[404] ]Northwest America.

In all layers and almost all regions, the mean CO mixing ratios are significantly decreasing, with trends ranging between -0.80 and -2.19 ppb.yr$^{-1}$ in the UT/TPL and between -0.5 and -0.8 ppb.yr$^{-1}$ in the LS. The lowest and least significant trends are observed in [..[405] ]Middle East (significant only in the TPL, with -0.8 ppb.yr$^{-1}$). In the UT and TPL, the highest trends of mean CO are observed in [..[406] ]Northeast Asia (-2.19 [-3.33;-0.82] ppb.yr$^{-1}$). Results show a similar picture for the fifth percentile, although with usually slightly lower trends. Apart from [..[407] ]Middle East, these trends are well homogeneous among the different regions. The results appear more contrasted for the [..[408] ]95$^{th}$ percentile. In the UT, a strong negative trend is observed in [..[409] ]Northeast Asia (-3.81 [-6.75; -1.37] ppb.yr$^{-1}$), about a factor of 2 higher than the trends observed in [..[410] ]East United-States, North Atlantic, Europe and Siberia (about -1.9 ppb.yr$^{-1}$), while no significant trend is found in both [..[411] ]the Western Mediterranean basin and Middle East.

[..[412] ]

The particularity of these last two regions lies in a [..[413] ]lower magnitude and the [..[414] ]absence of significance in the P95 trends[..[415] ], compared to the five other regions (excluding [..[416] ]Northwest America, as mentioned previously). [..[417] ]On one hand, as discussed in Sect. 3.2, they are expected to be less impacted by surface emissions, especially [..[418] ]Middle East because of a stronger subsidence. On the other hand, the CO decrease in the [..[419] ]five other regions is influenced by less frequent and/or weaker high CO events, as suggested by the decrease of the P95. The strong interannual variability of the P95 in [..[420] ]Northeast Asia is probably due to boreal forest fires.

Figure 14 shows the seasonal trends of [..[421] ]upper-tropospheric CO. The mean values decrease significantly in most regions during winter, from -1.15 [-3.67; -0.06] ppb.yr$^{-1}$ in [..[422] ]Middle East to -2.97 [-5.00; -0.75] ppb.yr$^{-1}$ in [..[423] ]North-

---

[404]removed: WNAm

[405]removed: the MidE region

[406]removed: the NEAs region

[407]removed: the MidE region

[408]removed: 95th

[409]removed: NEAs

[410]removed: EUS, NAt, Eur and Sib

[411]removed: WMed and MidE

[412]removed: The main difference between 'northern and southern regions '

[413]removed: higher

[414]removed: significance of

[415]removed: in the 5 'northern'

[416]removed: WNAm, as mentionned

[417]removed: As discussed in the section 3.2, the two southern regions

[418]removed: MidE

[419]removed: 5 'northern' regions is also influenced by the decrease of highest CO values (P95), i.e.

[420]removed: NEAs

[421]removed: upper tropospheric

[422]removed: MidE

[423]removed: NEAs

east Asia. During springtime, the negative trends persist in all regions but [..[424] ]Siberia and Middle East. The summertime decrease is significant in [..[425] ]East United-States and Europe only, and the trends during fall are barely significant in [..[426] ]Europe and the Western Mediterranean basin. Europe appears as the only region where negative mean CO trends are observed during all four seasons (although with a very low significance level during fall). The general decrease is also observed

for both the fifth and [..[427] ]95th percentiles during all seasons except fall. The highest negative trends are usually observed in [..[428] ]Northeast Asia. However, due to a strong [..[429] ]interannual variability, they are not always significant. For instance, albeit insignificant, a very strong negative trend of the [..[430] ]95th percentile of CO is observed in this region during springtime (-7.5 ppb.yr$^{-1}$). This decrease with its associated large uncertainties are likely linked to the decadal decrease of Asian boreal forest fires [..[431] ]mentioned in Jiang et al. (2017).

The more homogeneous results obtained in winter compared to summer may be explained by a more efficient [..[432] ]inter-continental transport of pollution. During winter, [..[433] ]warm conveyor belt events are more frequent and rather uplift the air masses with extratropical origins (Madonna et al., 2014). Consequently, this transport pathway may enhance the response of [..[434] ]upper-tropospheric CO mixing ratios to the reduction of surface emissions by a homogenized wintertime negative trend.

Most of our results in the UT over Europe are in agreement with the study of Petetin et al. (2016) based on IAGOS data above Frankfurt and Munich airports, with similar CO negative trends. The main differences [..[435] ]lie on an expected better significance for our trends, because of the size of [..[436] ]Europe region allowing the use of a larger amount of data. It suggests the trends they derived in the UT above these airports are representative of the UT in [..[437] ]Europe, and some of their insignificant trends are due to their limited amount of data, as they found insignificant trends in summer and autumn. They also found

no significant trends for P5 during these seasons probably for the same reasons.

Over the whole troposphere, Worden et al. (2013) derived a global decrease of CO columns from MOPITT satellite observations over the period [..[438] ]2000–2011, with relative trends of -1.42±0.40 %.yr$^{-1}$ in eastern US, -1.44±0.44 %.yr$^{-1}$ in Europe and -1.60±0.96 %.yr$^{-1}$ in eastern China. At the global scale, the decrease of CO columns observed by MOPITT over

---

[424] removed: Sib and MidE
[425] removed: EUS and Eur
[426] removed: Eur and WMed
[427] removed: 95th
[428] removed: NEAs
[429] removed: IAV
[430] removed: 95th
[431] removed: mentionned
[432] removed: inter-continental
[433] removed: WCB
[434] removed: upper tropospheric
[435] removed: rely
[436] removed: Eur
[437] removed: Eur
[438] removed: 2000-2011

the period [..[439] ]2000–2012 is about -0.56 %.yr$^{-1}$ (Laken and Shahbaz, 2014). In the [..[440] ]Northern Hemisphere, the decrease of CO mixing [..[441] ]ratios at 500 hPa observed by AIRS from 2003 [..[442] ]until 2012 ranges between -1.28 ppb.yr$^{-1}$ over lands and -1.01 ppb.yr$^{-1}$ over oceans (Worden et al., 2013). Using ground-based solar [..[443] ]Fourier Transform InfraRed (FTIR) measurements at six European stations over the period [..[444] ]1998–2006, Angelbratt et al. (2011) observed similar negative trends of CO partial columns ([..[445] ]0–15 km) in central/western Europe (around -1.2±0.2 %.yr$^{-1}$ at Jungfraujoch and Zugspitze) and lower ones in northern Europe (about -0.6±0.2 %.yr$^{-1}$ at Harestua, Norway and Kiruna, Sweden). All these results remain consistent with the trends we found in the UT, although the mean estimate of our trend in [..[446] ]Northeast Asia is substantially higher (-1.89 [-2.87; -0.71] %.yr$^{-1}$, see Tab. D7 in the Supplement). Gratz et al. (2015) [..[447] ]analyzed springtime surface observations at [..[448] ]Mt-Bachelor Observatory (Oregon, [..[449] ]2,743 m a.s.l.) during [..[450] ]2004–2013 and showed a decrease of -3.1±2.4 ppb.yr$^{-1}$. They associated it with the decrease of CO anthropogenic emissions in Europe, North America and China. The decrease is substantially higher than in our study, probably because it concerns the lower part of the free troposphere, which is [..[451] ]more sensitive to the reduction of surface emissions.

The influence of the sampling frequency on the trends has been tested for each layer, and each statistic by applying the sampling frequency to the monthly time series of the [..[452] ]three most sampled regions ([..[453] ]East United-States, North Atlantic, Europe). The higher bias was obtained in the UT, for the mean values and the P5, with the sampling frequency of [..[454] ]the Western Mediterranean basin and Middle East. Applying the sampling frequency of these regions weakened the absolute CO trends [..[455] ]by 0.2 ppb.yr$^{-1}$, which remains negligible.

---

[439] removed: 2000-2012
[440] removed: northern hemisphere
[441] removed: ratio
[442] removed: to
[443] removed: FTIR
[444] removed: 1998-2006
[445] removed: 0-15
[446] removed: NEAs
[447] removed: analysed
[448] removed: Mount-Bachelor
[449] removed: 2743
[450] removed: 2004-2013
[451] removed: moire
[452] removed: most 3
[453] removed: EUS, NAt, Eur
[454] removed: WMed and MidE
[455] removed: of

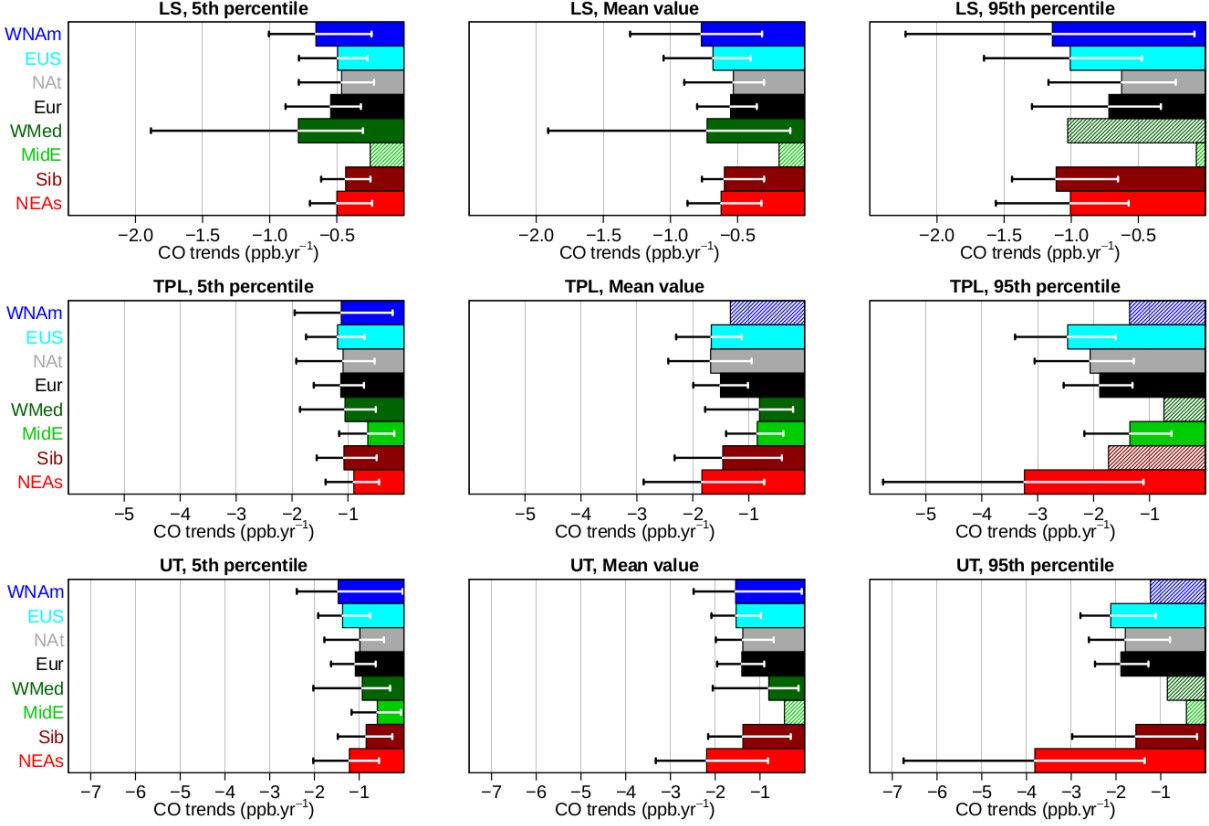

**Figure 13.** CO trends (ppb.yr$^{-1}$) in the UT, TPL and LS (from bottom to top), for the monthly fifth percentile (left [..[456] ]column), mean value (central [..[457] ]column) and [..[458] ]95$^{th}$ percentile (right [..[459] ]column). For each significant trend, the error bars represent the 95% confidence interval. The insignificant trends are represented by hatched areas. The coloured labels correspond to the regions defined in Fig. 1.

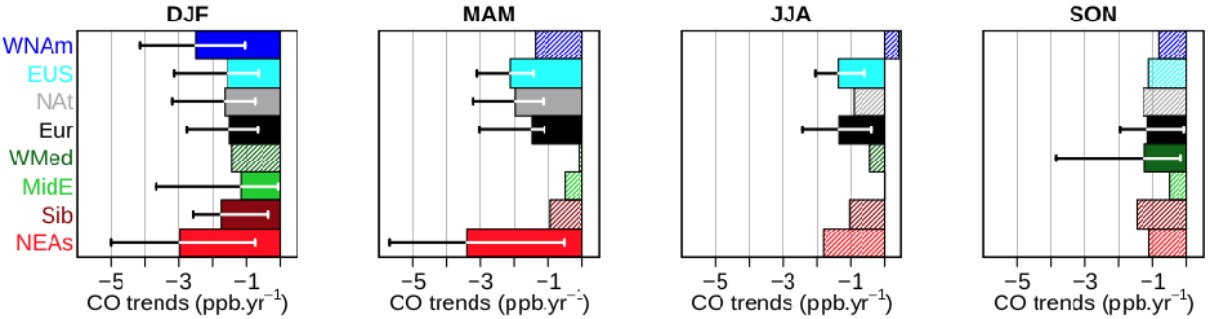

**Figure 14.** CO trends (ppb.yr$^{-1}$) in the UT, in each season. For each significant trend, the error bars represent the 95% confidence interval. The insignificant trends are represented by hatched areas. The coloured labels correspond to the regions defined in Fig. 1.

## 4   Summary and conclusions

In the framework of the European research infrastructure IAGOS, in situ measurements of $O_3$ and CO are performed in the UTLS since 1994 and 2002, respectively. Thanks to its high frequency of measurements over such long periods, IAGOS provides the most representative in situ data set in the UTLS. In the present study, we used the IAGOS data available until 2013 to establish a new semi-global climatology and to investigate the seasonal variability and trends in both $O_3$ (19 years of data) and CO (12 years of data). Results are presented in three distinct layers, namely the upper troposphere (UT), tropopause layer (TPL) and lower stratosphere (LS). We also conducted a detailed inter-regional comparison of $O_3$, CO and the $O_3$/CO ratio in eight regions of interest at northern mid-latitudes (Northwest America, East United-States, North Atlantic, Europe, the Western Mediterranean basin, Middle East, Siberia, Northeast Asia). These regions cover a wide range of longitudes excluding so far the Pacific Ocean. Note however that this Pacific region is now regularly visited by new IAGOS aircraft since mid-2012 (for a first analysis of these data, see Clark et al., 2015).

One of the objectives of the present study was to highlight the regional variability in the UTLS chemical composition in terms of horizontal distributions and trends. It is complementary to other analyses usually dealing with zonal means, and/or focusing on the vertical gradients of chemical species. Air masses were attributed to UT, TPL and LS based on the potential vorticity, following the methodology of Thouret et al. (2006) that first analyzed the IAGOS data set in the UTLS. Added value to this first study lies in a 10-year longer monitoring period above the major part of the extratropical Northern Hemisphere, completed with 12 years of CO measurements. As a tracer for transport from the surface and as one of the major $O_3$ precursors at the global scale, CO provides complementary information on the factors that control the $O_3$ seasonal cycles and trends.

The analysis of the semi-global horizontal distributions of $O_3$ and CO highlighted the following characteristics:

- At northern mid-latitudes, the more efficient photochemical activity is associated with the common summertime $O_3$ maximum in the UT, and the Brewer–Dobson circulation with the common springtime $O_3$ maximum in the LS. The general springtime CO maximum in the UT is seen as a consequence of the wintertime accumulation of emissions in the lower layers followed by an efficient vertical transport, before the summertime photochemical activity acts as a major sink for CO. However, the CO maxima extend into summer over (and downwind) regions where intense biomass burning occurs regularly, especially in Siberia. Another main feature of the northern extratropical UT relies on a zonal difference of $O_3$ in summer (with up to 15 ppb more over central Russia than in eastern North America) and an eastward gradient of CO from $60°$E to $140°$E, maximizing in spring and summer (approximately 5 ppb by $10°$ longitude).

- In the tropics, the CO mixing ratio maximizes in the regions of intense biomass burning. The $O_3$ mixing ratios reach their maximum during fall over southern Africa, while the lowest values are found on both sides of the equatorial Pacific Ocean during most seasons.

O$_3$ and CO seasonal variations were analyzed in the eight extratropical regions of interest. The inter-regional comparison can be summarized as follows:

In the upper troposphere:

- In most regions, the O$_3$ mixing ratios continuously increase from 40–50 ppbv in winter to 70–90 ppbv in summer. One regional specificity is found in Northwest America and Northeast Asia where O$_3$ mixing ratios start to decrease after May and June respectively (against July in other regions), likely due to the North American and Asian monsoons that bring poor-O$_3$ air masses from the subtropical Pacific Ocean in summer.

- The CO mixing ratios range from 80–90 ppbv in fall to 110–140 ppbv in spring in all regions except (i) Northeast Asia where this spring maximum extends to summer, and (ii) Siberia where the CO mixing ratios show a leveling-off in spring–summer followed by a distinct peak in July.

- In all regions, the O$_3$/CO ratio increases from 0.4–0.5 in winter to 0.7–1.0 in summer, the lowest values occurring in Northwest America and the highest in the Western Mediterranean basin and Middle East.

In the lower stratosphere :

- The O$_3$ mixing ratios increase from 150–200 ppbv in fall to 350–450 ppbv in late spring–early summer in most regions, one noticeable exception relying on Middle East where O$_3$ quickly decreases after April (thus earlier and more strongly than in the other regions).

- Low CO mixing ratios ranging from 45 to 65 ppbv are observed in the LS, with substantially lower and noisier seasonal variations in most regions compared to the UT, the main exceptions relying on Middle East and Northeast Asia where slightly higher CO mixing ratios are observed in summer.

- The O$_3$/CO ratio increases from 3–5 in late fall–early winter to (i) 8 in spring in the Western Mediterranean basin and Middle East, and (ii) 10–12 in late spring–early summer in the other regions. This lower O$_3$/CO ratio observed in the two former regions can result from the subtropical air masses (low O$_3$, high CO) uplifted in the Asian monsoon then transported northward by the western branch of the anticyclonic circulation.

Both in terms of mixing ratios and seasonal variations, the tropopause layer appears as a transition between these two previous layers.

The O$_3$ and CO trends were also investigated in these different regions. The two-decadal trends in O$_3$ are positive and statistically significant at the 95% level in most regions (excluding North Atlantic), with best estimates of trend ranging from +0.25 to +0.45 ppb.yr$^{-1}$. This evolution appears to be linked to the increase of the lowest values, and not necessarily with the highest values. Seasonal O$_3$ mixing ratios show a significant increase during fall and winter in the three Atlantic sector regions (East United-States, North Atlantic and Europe), and a significant increase during spring or summer in Siberia and Northeast Asia. Such an inter-regional difference in the seasonality of the O$_3$ trends suggests that the increase in the Atlantic sector regions is linked to a long-term increase of the background mixing ratios, while emissions of precursors

during favorable photochemical conditions still drive part of the increase over Asia. The $O_3$ trends in the TPL are similar to the trends in the UT, albeit stronger and generally with a better significance. In contrast, all mean $O_3$ trends remain insignificant in the LS. This is different from the results presented in Thouret et al. (2006), where the first 9 years of IAGOS data (1994–2003) showed significant positive trends in $O_3$ in the LS, with the same magnitude as those in the UT.

With an additional decade of observations, our present results suggest that the trends previously reported in the LS were influenced by the relatively low values over the period 1994–1997 (following the $O_3$ reduction subsequent to the Pinatubo eruption in 1991) and the 1998–1999 positive anomaly (induced by the extreme El Niño event in 1997: Koumoutsaris et al., 2008).

In the UT, the mean CO mixing ratios are significantly decreasing in most regions (excluding Middle East), with best

estimates of trend ranging from -1.5 to -2.2 ppb.yr$^{-1}$. Both fifth and 95$^{th}$ percentiles of the CO distribution also depict a negative trend, the decrease of the lowest values being more homogeneous compared to the highest values. At the seasonal scale, while summer and fall trends are either low or not significant, the CO mixing ratios are found to decrease significantly during winter in most regions. The strongest decrease is observed in Northeast Asia during both winter and spring (approximately -3 %.yr$^{-1}$). The decrease of CO in the northern extratropics is usually related to the reduction of

anthropogenic emissions.

Identifying unambiguously the processes responsible for these trends would require a properly validated model and is thus beyond the scope of this paper, but one can still provide some insights. The decrease of CO is related to a reduction of the anthropogenic emissions. However, our study does not highlight a direct link between the long-term evolution of $O_3$ and CO in the UT since both chemical species depict opposite trends. Although CO is only one precursor of $O_3$ among many

others, one may have expected that the reduction of its emissions would go hand in hand with the decrease of other $O_3$ precursors. However, rapid changes in the technology used in the various combustion processes may change the speciation of the emissions. In addition, many other chemical species not co-emitted with CO (e.g. biogenic compounds) play a role in the $O_3$ budget.

The lowest trends in both $O_3$ and CO in the Middle East region, where the UT is less influenced by the middle and

lower troposphere, may reflect a significant global tropospheric contribution driving the higher trends in $O_3$ and CO over other regions. This would be in line with recent findings by Zhang et al. (2016), attributing the major part of the increase in the Northern Hemisphere tropospheric $O_3$ burden to the growing (sub)tropical emissions. However, the increase of $O_3$ mixing ratios in the UT takes place in almost every extratropical region, and the inter-regional differences are not statistically significant enough to exclude other leading processes.

We compared our results to other studies analyzing trends in lower levels in the free troposphere. The $O_3$ increase is usually weaker at free-tropospheric GAW stations than in the middle and upper troposphere. For instance, based on aircraft, sondes and surface measurements in Japan from 1991 to 2015, **?**, submitted recently highlighted a dependence of the $O_3$ trends on altitude, with an $O_3$ leveling-off in the 2000s in the lower part of the free troposphere, but a persistent increase in the middle and upper troposphere. This suggests that the $O_3$ trends in the UT may be linked to an increase of

in situ local emissions of precursors in the UT (lightning, aircraft) and/or stratosphere-to-troposphere $O_3$ flux as reported

in other studies (e.g. Neu et al., 2014a). This would be also consistent with the increasing vertical cross-tropopause mass flux in the extratropics during 1996–2011 modeled with ERA-Interim and JRA-55 reanalyses (Boothe and Homeyer, 2017), linked to the acceleration of the Brewer–Dobson circulation, in agreement with observations and chemistry-climate models (Butchart, 2014 and references therein; more recently, Garfinkel et al., 2017). An increase of the stratospheric influence on the $O_3$ mixing ratios in the UT can result from (i) an increasing number of stratospheric intrusions and/or (ii) the stratospheric $O_3$ recovery (both leading to an enhancement of the stratosphere-to-troposphere $O_3$ fluxes). Our study does not support this second option since all mean $O_3$ trends in the LS were found to be statistically insignificant over the studied period, although the insignificance of our trends may be due to the interannual variability. Finally, attributing the trends of $O_3$ in the UT to one leading process remains difficult and requires further investigations thanks to global models.

The IAGOS data set has highlighted significant inter-regional and interannual variabilities, which can provide helpful information for shorter and/or localized measurement campaigns. Its good vertical accuracy in the UTLS makes it a useful dataset for evaluating the ability of current chemistry-transport and chemistry-climate models to reproduce the characteristics of the chemical composition in this layer, including climatologies and trends.

[..[460] ][..[461] ]

-   [..[462] ]

-   [..[463] ]

---

[460] removed: In situ measurements in the UTLS are performed in the framework of the European research infrastructure IAGOS for ozone since 1994 and carbon monoxide since 2002. From IAGOS data, we have established a new global climatology for ozone and carbon monoxide in three different layers, the upper troposphere, tropopause layer and lower stratosphere over the periods August 1994 – December 2013 and December 2001 – December 2013, respectively. Thanks to its high frequency of measurements over long periods, the IAGOS database provides representative sets of data in the UTLS. In the present study, we examine more precisely the seasonal cycles and the (bi-) decadal trends of these trace gases, above eight subcontinental-scale regions of the northern extratropics. These regions cover a wide range of longitudes excluding so far the Pacific Ocean. For information, this latter region has been visited by IAGOS equipped aircraft since mid-2012. A first analysis of these data has been presented by Clark et al. (2015). One of the objectives of the present study was to highlight the regional variability in the UTLS in terms of horizontal composition distributions and trends. It is complementary with other analyses usually dealing with zonal means, and/or focusing on the vertical gradients of species. The methodology attributing air masses to UT, TPL and LS is the same as in Thouret et al. (2006). Added value to these previous results lies in a 10-year longer monitoring period above the major part of the extratropical northern hemisphere, completed with 12 years of CO measurements. As one of the major $O_3$ precursors at global scale, CO provides complementary information on the factors that control the $O_3$ seasonal cycles and trends.

[461] removed: The analysis of the horizontal mean seasonal distributions highlights the following general and regional characteristics:

[462] removed: In the northern hemisphere, the more efficient photochemical activity is associated with the common summertime $O_3$ maximum in the UT, and the Brewer-Dobson circulation with the common springtime $O_3$ maximum in the LS. The general springtime CO maximum in the UT is seen as a consequence of the wintertime accumulation of emissions in the lower layers followed by an efficient vertical transport, before the summertime photochemical activity acts as a major sink for CO. However, CO maxima extend into summer over regions (and downwind regions) where large and intense boreal biomass burning occurs regularly, like Siberia and Canada. Another main feature of the northern hemisphere, in the UT, resides in a west-east difference of $O_3$ in summer with up to 15 ppb more over central Russia compared with eastern north America. There is also a systematic eastward gradient of CO from $60°$E to $140°$E of about 5 ppb by $10°$ longitude, especially noticeable in spring and summer.

[463] removed: In the tropics, the CO mixing ratio maximizes in regions of biomass burning. Interestingly, the $O_3$ distribution presents the lowest mixing ratios for the four seasons on both sides of the Pacific Ocean. $O_3$ maximizes in SON over Southern Africa.

[464]removed: The mean $O_3$ seasonal cycles based on up to 19 years vary from 45 ppb in January to 80 ppb in June-July in the UT, and from 170 ppb in October-November to 400 ppb in April-May in the LS. The CO seasonal cycles based on 12 years data vary from 80 ppb in September-October to 120 in April in the UT, with a 140 ppb maximum in the two "Pacific regions" (Western North America and Northeast Asia). Most regions do not show any seasonal cycle in lower stratospheric CO. Comparing the mean seasonal cycles over these eight areas in the northern mid-latitudes highlights other regional characteristics like (i) a broad spring-summer CO maximum over Northeast Asia (NEAs), and (ii) a springtime $O_3$ maximum over Western North America (WNAm) and Northeast Asia to a lesser extend. Such summertime irregularities in the upper tropospheric cycles of $O_3$ mixing ratios above the two "Pacific regions" likely highlight the impact of the Asian and North American monsoons, bringing poor-$O_3$ air masses from the subtropical Pacific Ocean. Indeed, the springtime and summertime CO maxima in the UT suggest that WNAm and NEAs are located downwind of a common area with strong emissions (anthropogenic and biomass burning emissions). In the LS, the Middle-East region (MidE) shows the lowest $O_3$ and highest CO mixing ratios. This can be related to the impact of subtropical air masses uplifted in the Asian monsoon, then transported northward by the western branch of the anticyclonic circulation.

[465]removed: The bi-decadal trends in $O_3$ are positive and statistically significant at a $2\sigma$ level in most regions (excluding North Atlantic), with the best estimate of the trend ranging from +0.25 to +0.45 ppb.yr$^{-1}$ on average. This evolution appears to be linked to the increase of the lowest values, and not necessarily with the highest values. Seasonal $O_3$ mixing ratios show a significant increase during fall and winter in the three Atlantic regions (Eastern US, North Atlantic and Europe), and a significant increase during spring or summer in the Asian regions, Siberia and Northeast Asia, respectively. Such a regional difference in the seasonality of $O_3$ trends suggests that the increase in the Atlantic regions is linked to a long-term elevation of the background mixing ratios, while emissions of precursors during favorable photochemical conditions still drive part of the increase over Asia. The trends in $O_3$ in the TPL are similar to the trends in the UT, albeit stronger and generally with a better significance. Nevertheless, all mean $O_3$ trends remain insignificant in the LS. This is different from the results presented in Thouret et al. (2006), where the first nine years of IAGOS data showed positive trends in $O_3$ in the LS, with the same magnitude as those in the UT. With an additional decade of observations, our present results suggest that the previous reported trends in the LS were influenced by the low values from 1994 until 1997 and the 1998-1999 positive anomaly, i.e. during the $O_3$ recovery after Pinatubo's eruption in 1991, followed by the global anomaly subsequent to the extreme El-Niño event of 1997 (Koumoutsaris et al., 2008).

[466]removed: The decadal trends in CO are negative and significant in most regions (excluding MidE), with the best estimate ranging from -1.5 to -2.2 ppb.yr$^{-1}$ on average in the UT. Both fifth and 95th percentiles from the CO distribution are decreasing. The decrease in the lowest values is more homogeneous than in the highest values, which can be related to the strong interannual variability of biomass burning emissions. However, the wintertime decrease is significant in the UT over all regions, while the summertime trends are either very low or not significant. The strongest decreases are observed in winter and spring over NEAs. As the overall decrease is associated with the reduction of surface emissions in the northern extratropics, the locally insignificant trends of P95 above MidE suggests that the UT over there is less impacted by surface emissions than over other regions. This was indeed the rationale for defining this region as the "large scale subsidence" region.

[467]removed: Negative trends in CO in the UT definitely show the impact of global reduction of surface emissions. Positive trends in $O_3$ in the UT and TPL definitely show no direct link between the long-term evolution of $O_3$ and surface CO emissions. It is important to note that CO does not represent all the $O_3$ precursors. Trends in the LS are not significant for $O_3$ over this period. Therefore, it is still difficult to attribute the trends of $O_3$ in the UT to one leading process. The lowest trends in both $O_3$ and CO in the MidE region, where the UT is less influenced by the middle and lower troposphere, may reflect a significant global tropospheric contribution driving the higher trends of $O_3$ and CO over other regions. This is in line with recent findings by Zhang et al. (2016), attributing the major part of the increase in the northern hemisphere tropospheric $O_3$ burden to the increase in the emissions located between the tropics. However, the increase in the $O_3$ mixing ratio takes place in almost every region, and the regional differences are not statistically significant to exclude other leading processes. We compared our results to other studies analyzing trends in lower tropospheric levels. It is clear that the $O_3$ increases observed in most free tropospheric GAW stations are weaker than the ones in the middle and upper troposphere. This difference between the upper sublayers and the lower one in the free troposphere is found notably in Tanimoto et al. (2017, in preparation) using aircraft, sondes and surface measurements in Japan from 1991 to 2015. On one hand, the $O_3$ mixing ratios were shown to increase monotonically in the middle and upper parts of the free troposphere. On the other hand, the lower part of the free troposphere was characterized by an increase until 2000, then by a flattening. They concluded that it was linked to the influence of the decreasing baseline $O_3$ in the boundary layer. It may suggest that the $O_3$ trends in the UT are further linked to in situ emissions of precursors directly in the UT (lightning, aircraft) and/or an increase in the mid-latitudinal STE $O_3$ flux as reported in other studies (e.g. Neu et al., 2014a). The latter is not inconsistent with our results in the LS, as the absence of a significant trend of $O_3$ in this layer may be due to a strong IAV. It is also consistent with the

## 5 Data availability

All ozone and carbon monoxide data [..[469] ]used in this study were obtained from the existing IAGOS database, freely available on the IAGOS website (http://www.iagos.org) and via the AERIS website (http://www.aeris-data.fr).

*Acknowledgements.* The authors acknowledge the strong support of the European Commission, Airbus, and the Airlines (Lufthansa, Air-France, Austrian, Air Namibia, Cathay Pacific, Iberia and China Airlines so far) who carry the MOZAIC or IAGOS equipment and perform the maintenance since 1994. In its last 10 years of operation, MOZAIC has been funded by INSU-CNRS (France), Météo-France, Université Paul Sabatier (Toulouse, France) and Research Center Jülich (FZJ, Jülich, Germany). IAGOS has been additionally funded by the EU projects IAGOS-DS and IAGOS-ERI. We also wish to acknowledge our colleagues from the IAGOS team in FZJ, Jülich for useful discussions. The MOZAIC–IAGOS database is supported by AERIS (CNES and INSU-CNRS). Data are also available via AERIS web site www.aeris-data.fr. The authors ought to thank the Midi-Pyrénées region and Météo-France too, for funding Yann Cohen's PhD.

increase in $O_3$ STE over the century suggested by Neu et al. (2014a), and the increase in the vertical STT mass flux in the extratropics during 1996-2011 modeled with ERA-Interim and JRA-55 reanalyses (Boothe and Homeyer, 2017), linked to the acceleration of the Brewer-Dobson circulation, in agreement with observations and chemistry-climate models (Butchart, 2014 and references therein; more recently, e.g. Garfinkel et al., 2017).

[468]removed: Finally, IAGOS data have shown significant regional and interannual variabilities that can provide helpful information for shorter and/or localized observation campaigns. The vertical accuracy makes IAGOS an available tool for realistic comparisons with CTMs and CCMs, allowing a precise evaluation of their ability to reproduce characteristics of the chemical composition of the UTLS, including climatologies and trends.

[469]removed: sets

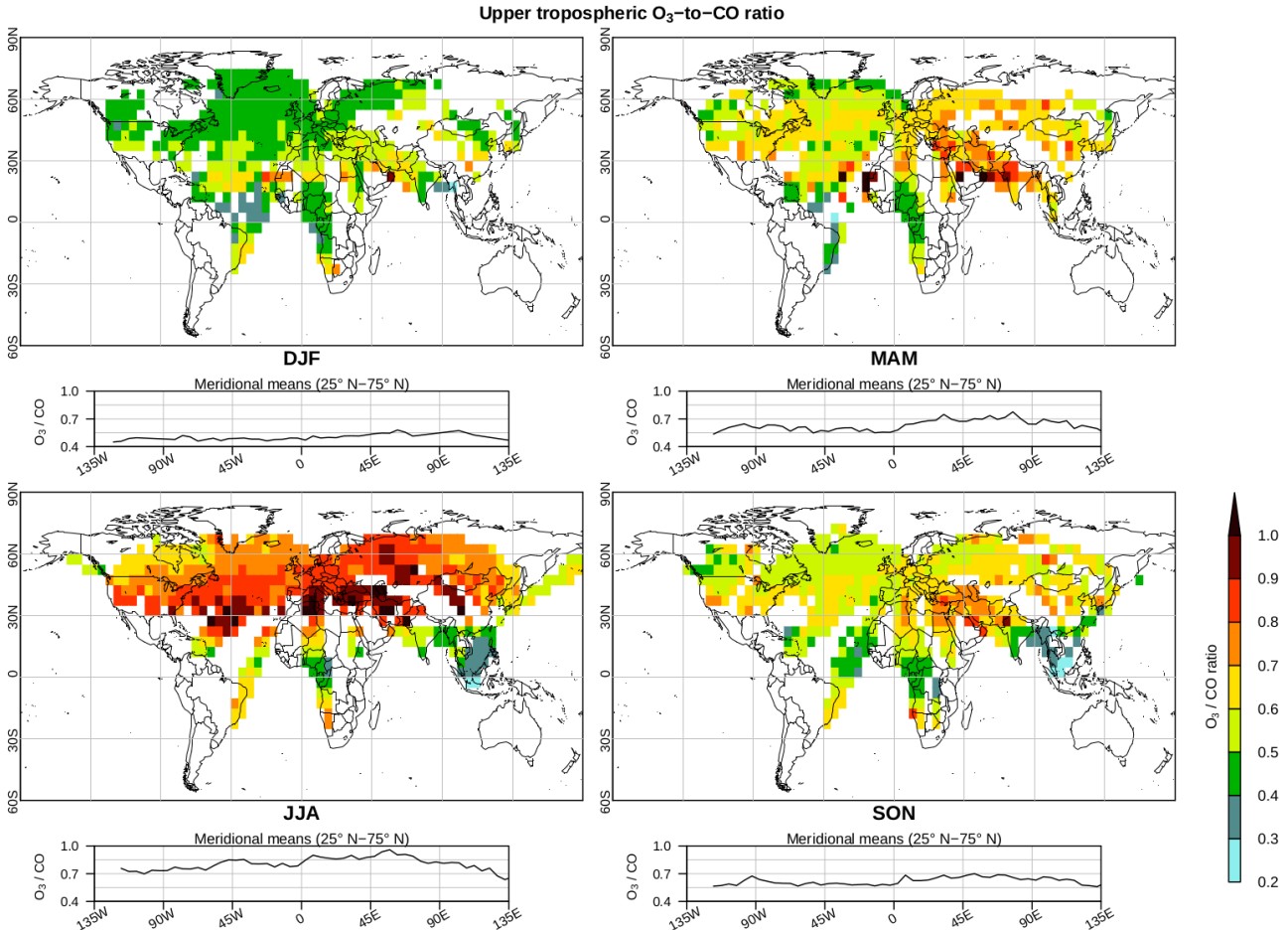

**Figure A1.** Horizontal distributions of the O₃-to-CO ratio in the UT averaged from December 2001 to November 2013, in every season. Each 2D distribution is projected on the zonal axis below, with a meridional average of the northern extratropical zonal band (from 25°N to 75°N). The 2D (respectively 1D) distributions have a 5° × 5° (respectively 5°) resolution.

# Upper tropospheric O$_3$

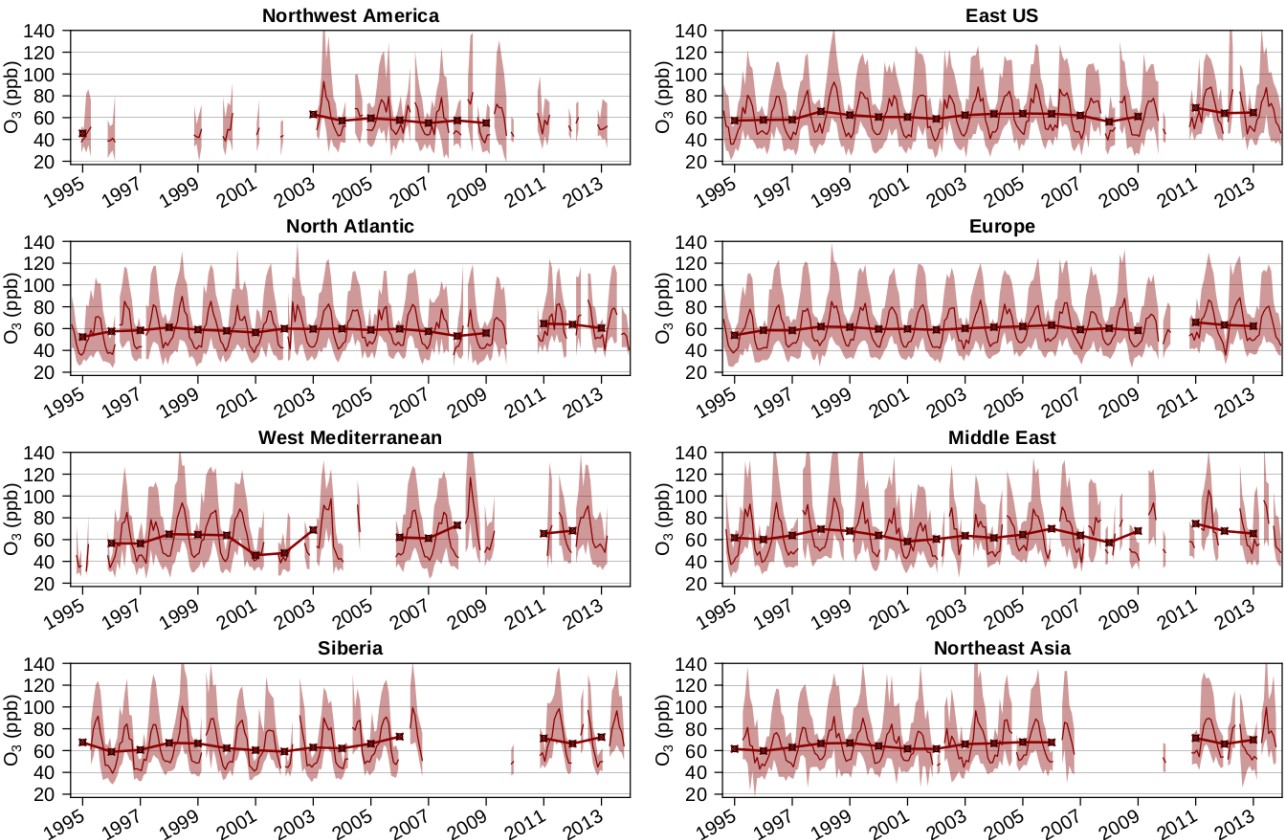

**Figure B1.** O$_3$ monthly mean values in the UT (thin curve). The interval from fifth and 95$^{th}$ percentiles is filled in red. The yearly mean values are represented by the bold curve and the black points.

# Tropopause layer O₃

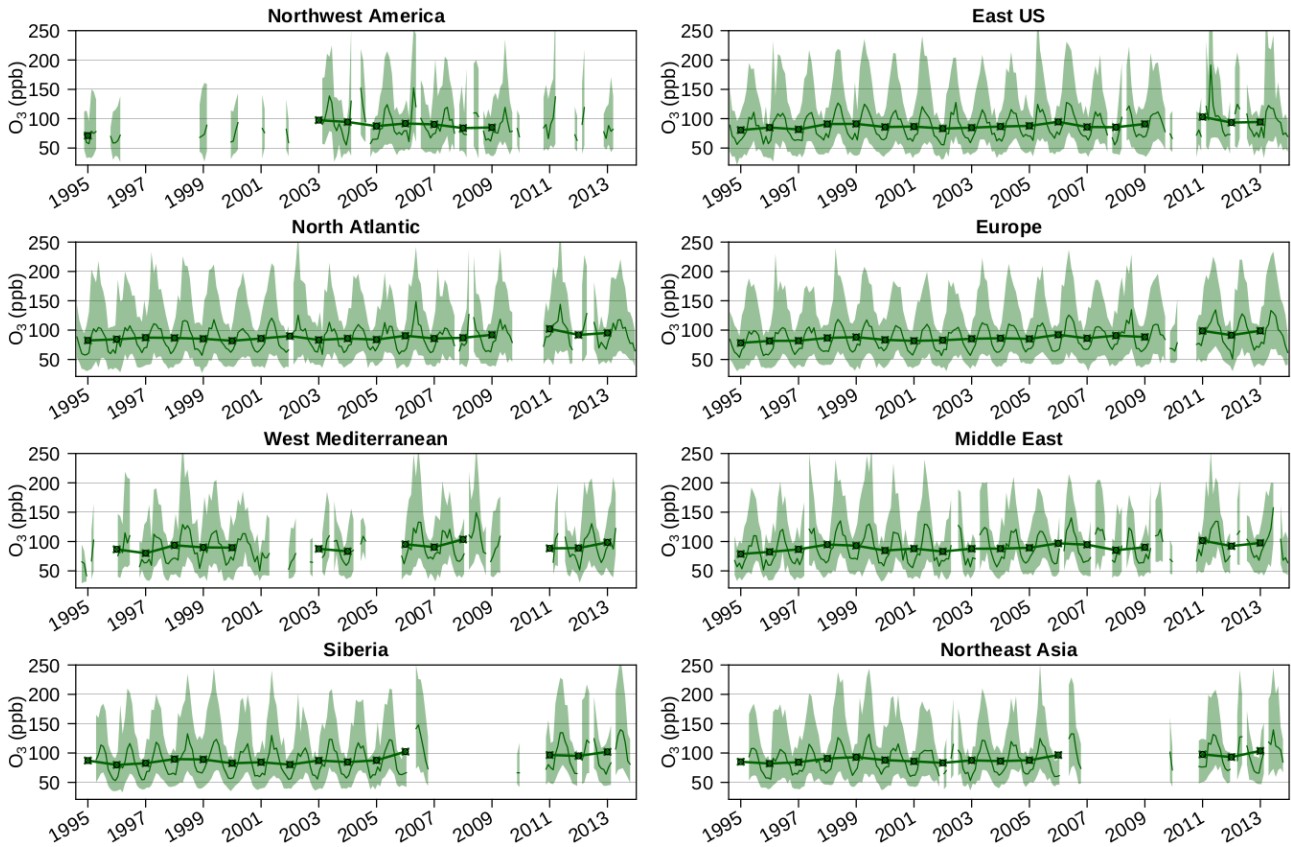

**Figure B2.** O₃ monthly mean values in the TPL (thin curve). The interval from fifth and 95ᵗʰ percentiles is filled in green. The yearly mean values are represented by the bold curve and the black points.

# Lower stratospheric O₃

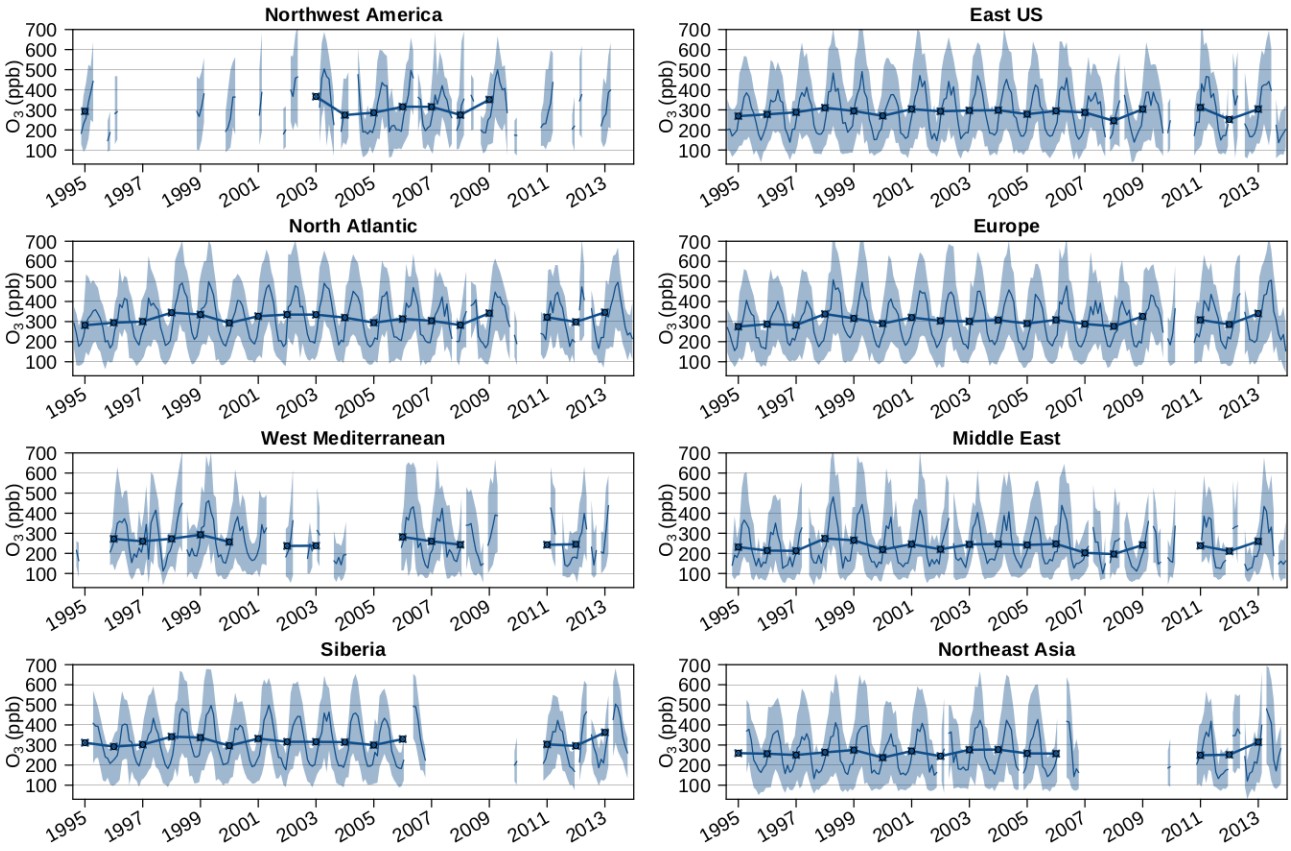

**Figure B3.** O₃ monthly mean values in the LS (thin curve). The interval from fifth and 95th percentiles is filled in blue. The yearly mean values are represented by the bold curve and the black points.

# Upper tropospheric CO

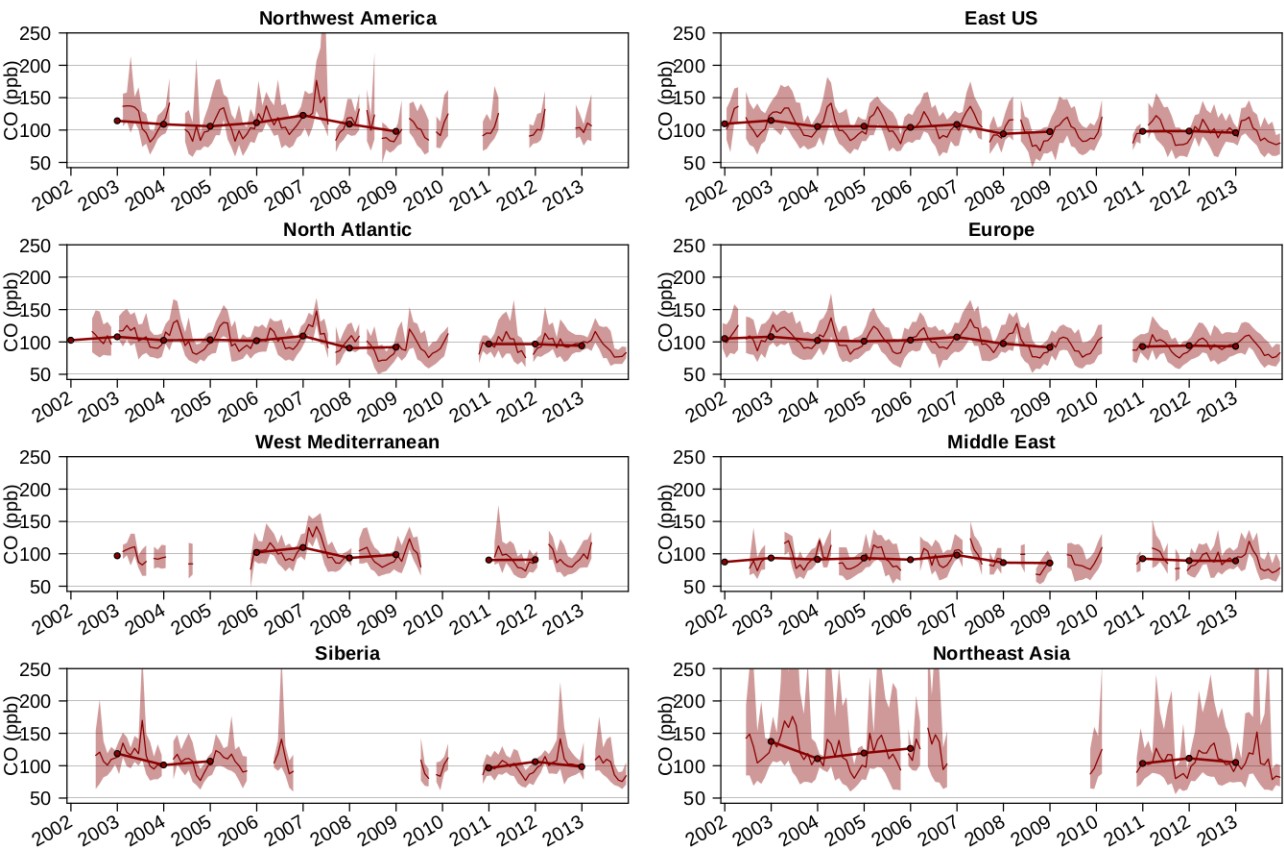

**Figure B4.** CO monthly mean values in the UT (thin curve). The interval from fifth and 95th percentiles is filled in red. The yearly mean values are represented by the bold curve and the black points.

# Tropopause layer CO

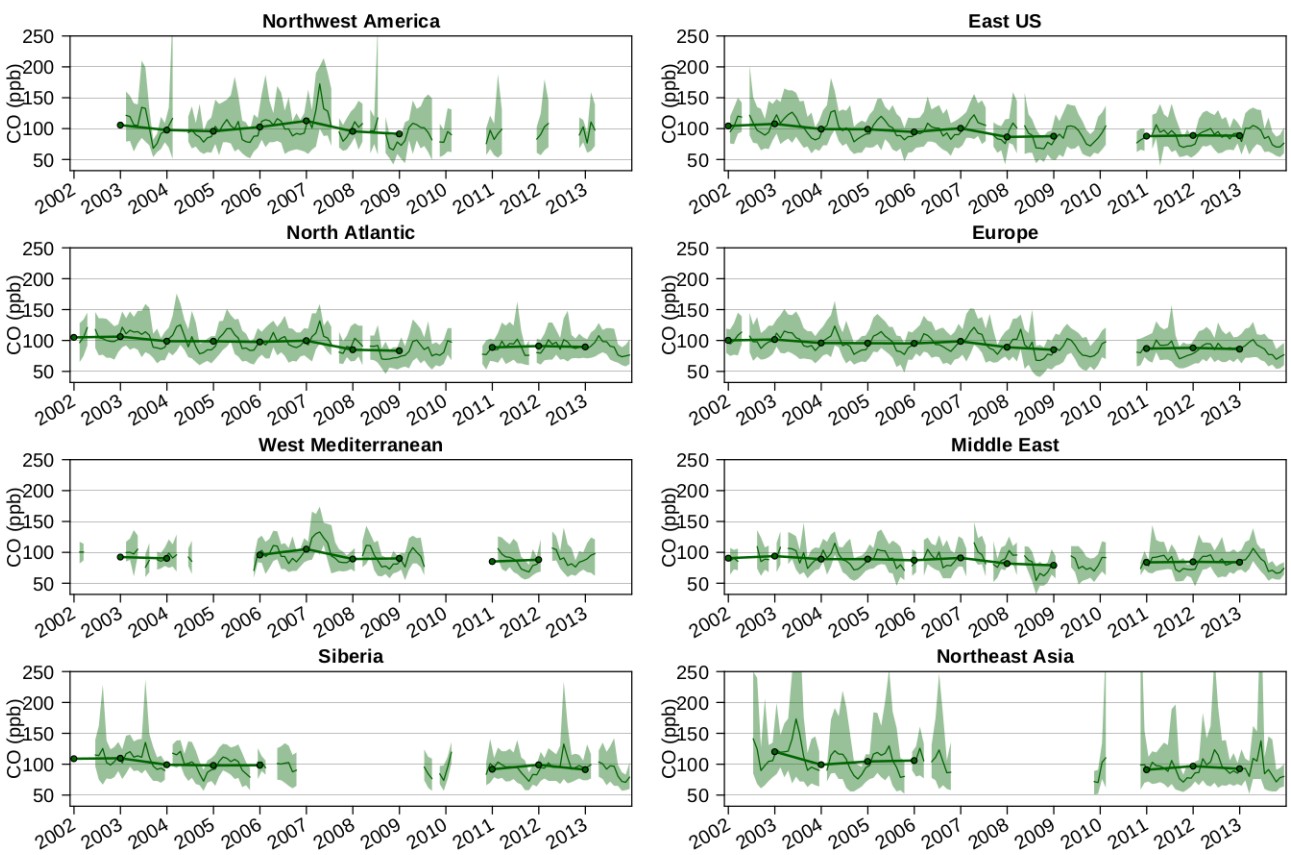

**Figure B5.** CO monthly mean values in the TPL (thin curve). The interval from fifth and 95th percentiles is filled in green. The yearly mean values are represented by the bold curve and the black points.

# Lower stratospheric CO

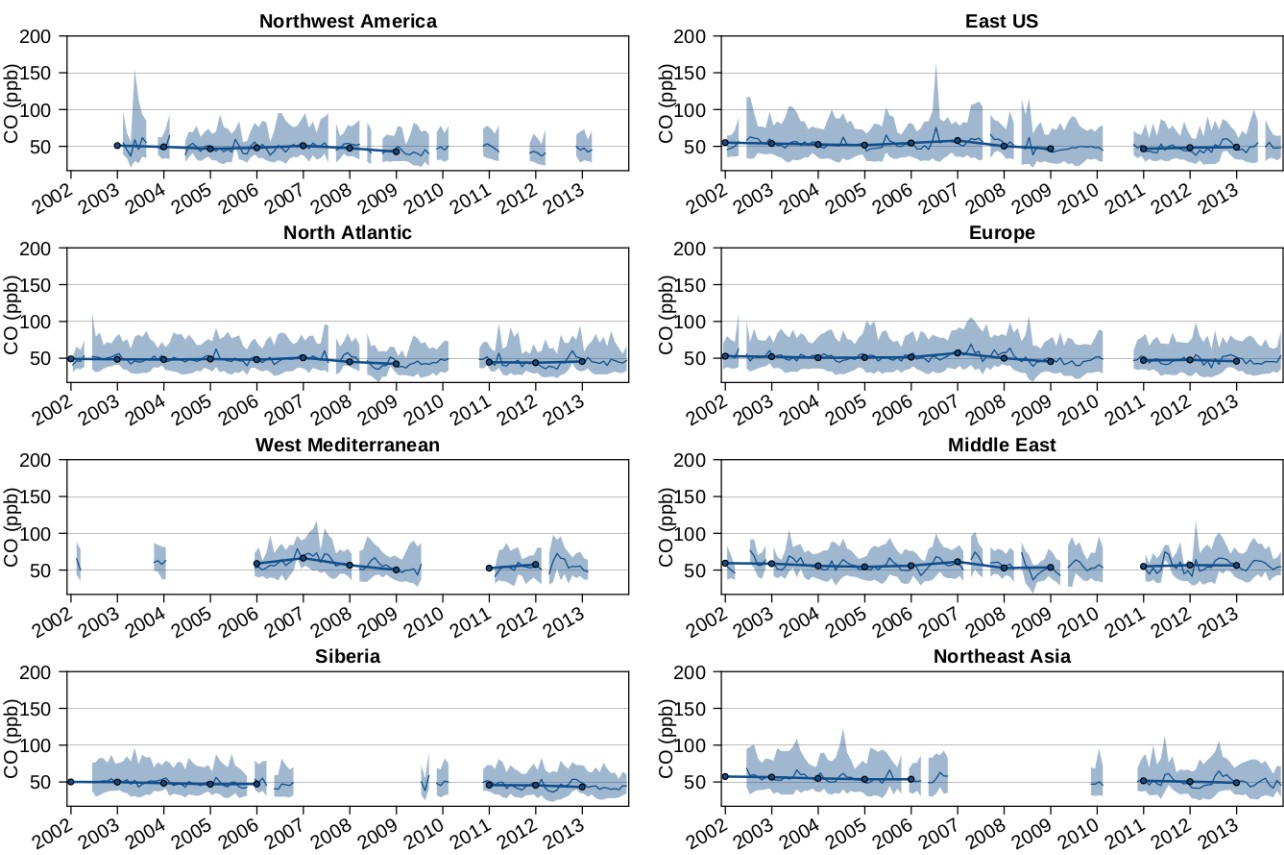

**Figure B6.** CO monthly mean values in the LS (thin curve). The interval from fifth and 95[th] percentiles is filled in blue. The yearly mean values are represented by the bold curve and the black points.

# Upper tropospheric O$_3$

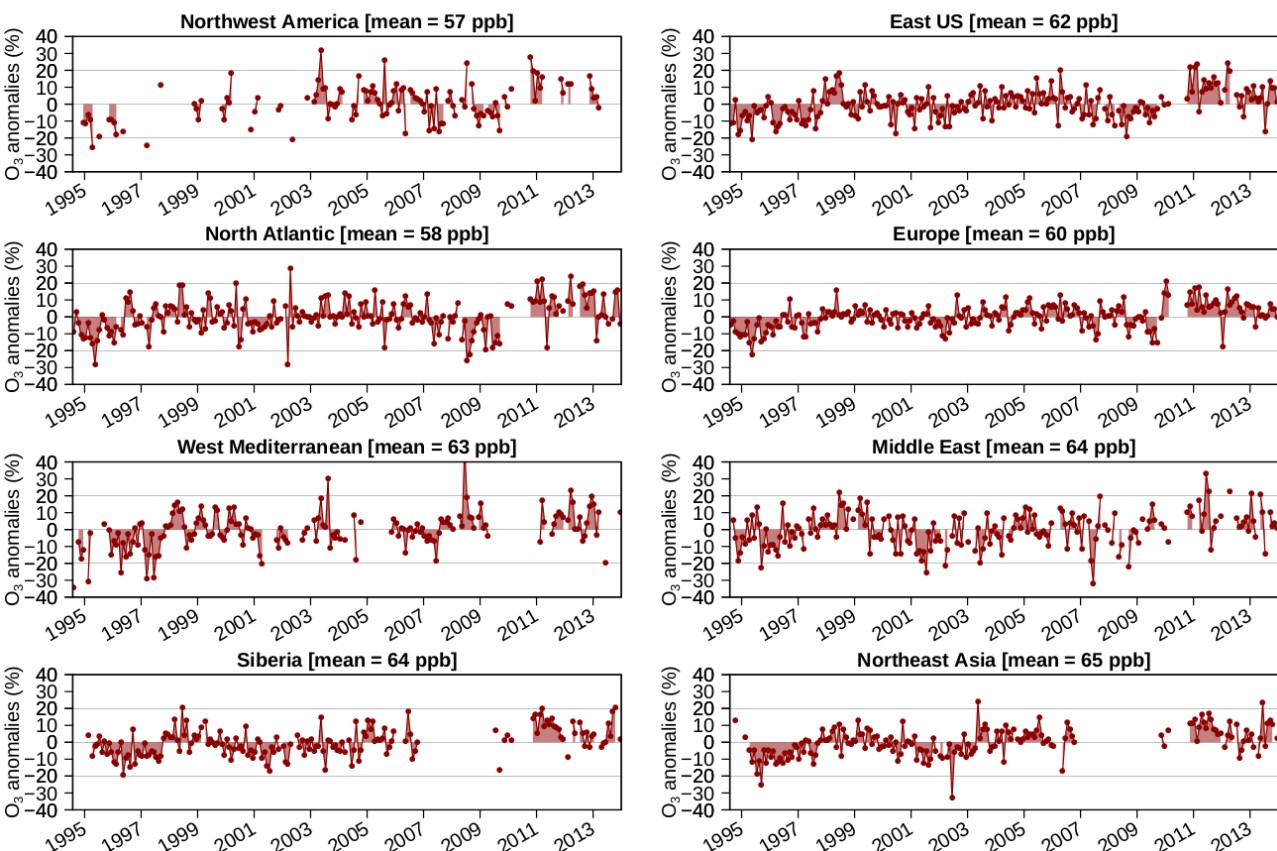

**Figure C1.** Relative monthly anomalies of O$_3$ volume mixing ratio in the upper troposphere. The units % refer to the corresponding climatological means indicated above each graphic.

# Lower stratospheric O$_3$

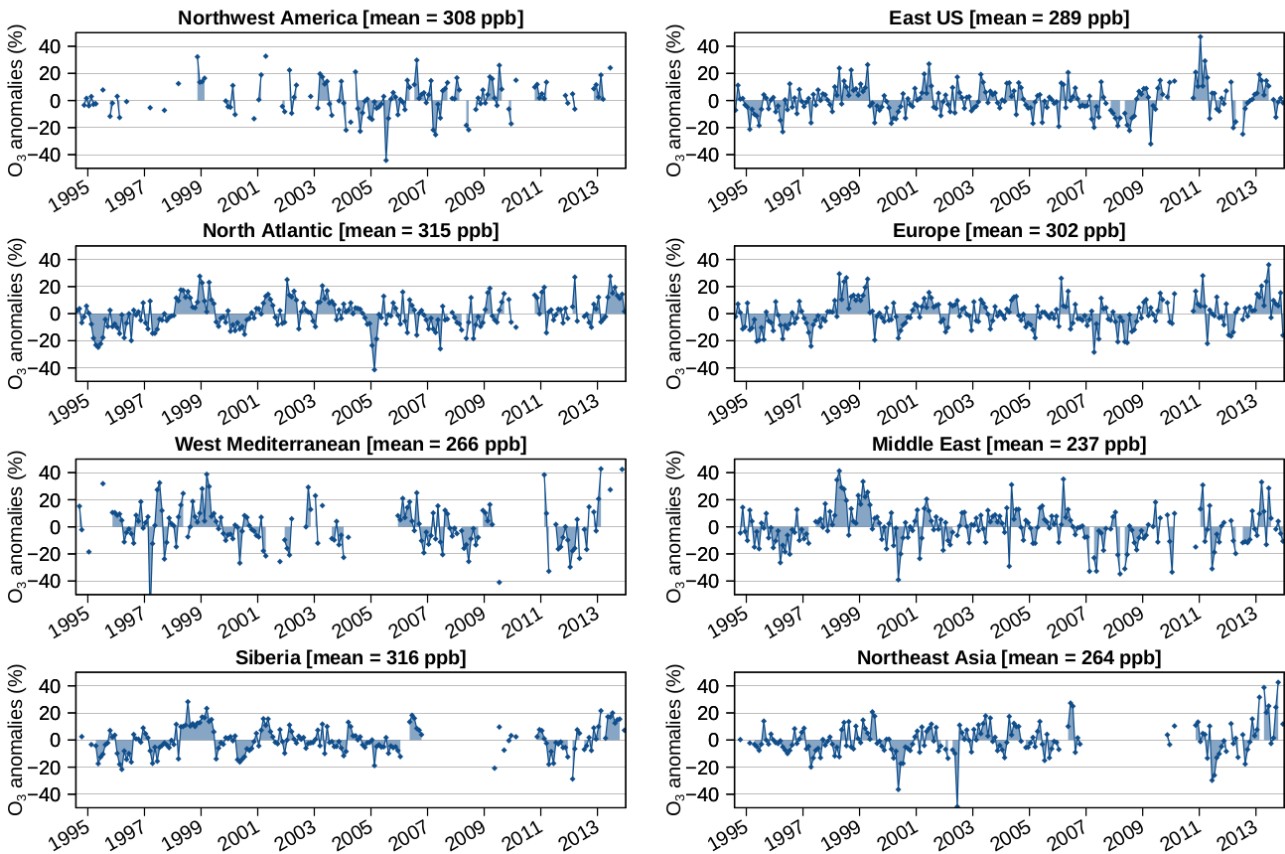

**Figure C2.** Relative monthly anomalies of O$_3$ volume mixing ratio in the lower stratosphere. The units % refer to the corresponding climatological means indicated above each graphic.

# Upper tropospheric CO

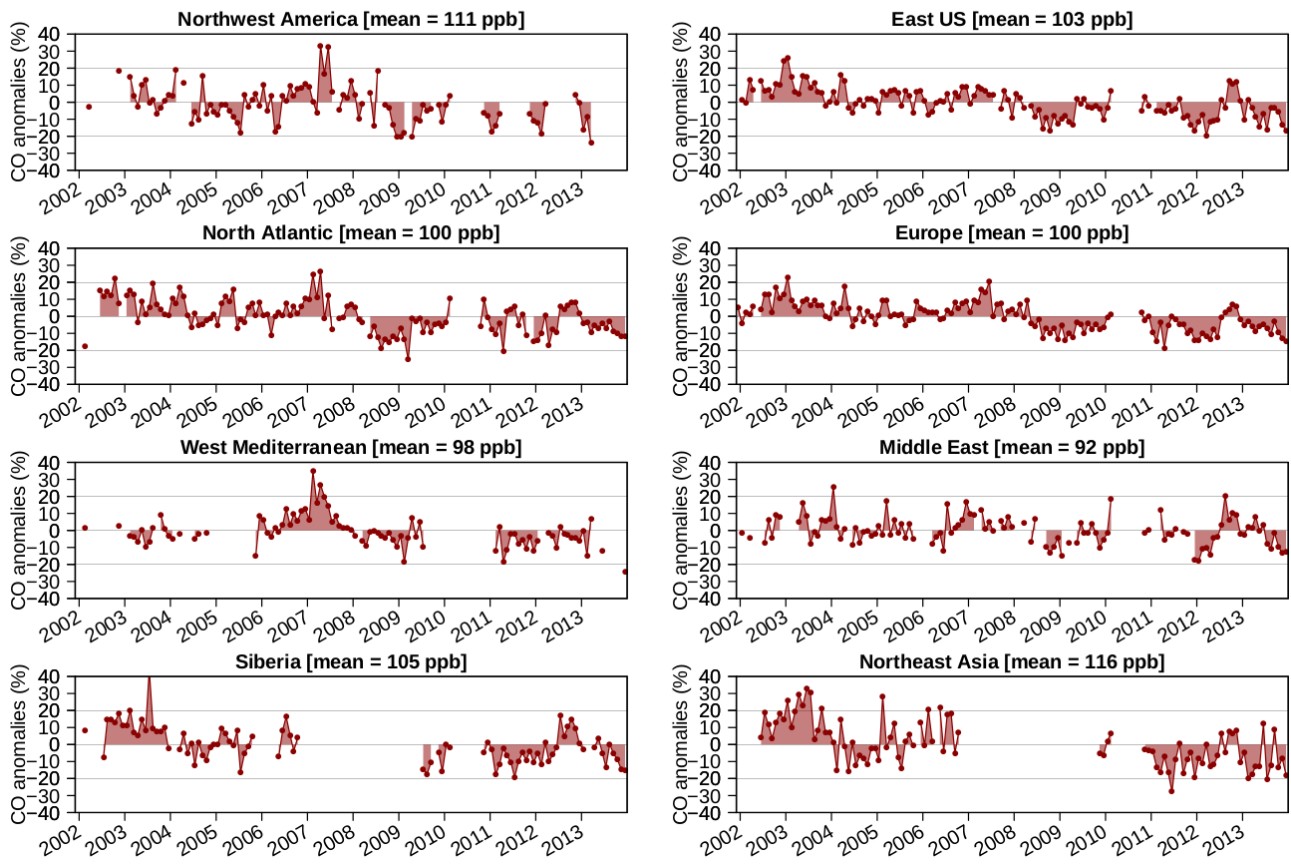

**Figure C3.** Relative monthly anomalies of CO volume mixing ratio in the upper troposphere. The units % refer to the corresponding climatological means indicated above each graphic.

# Lower stratospheric CO

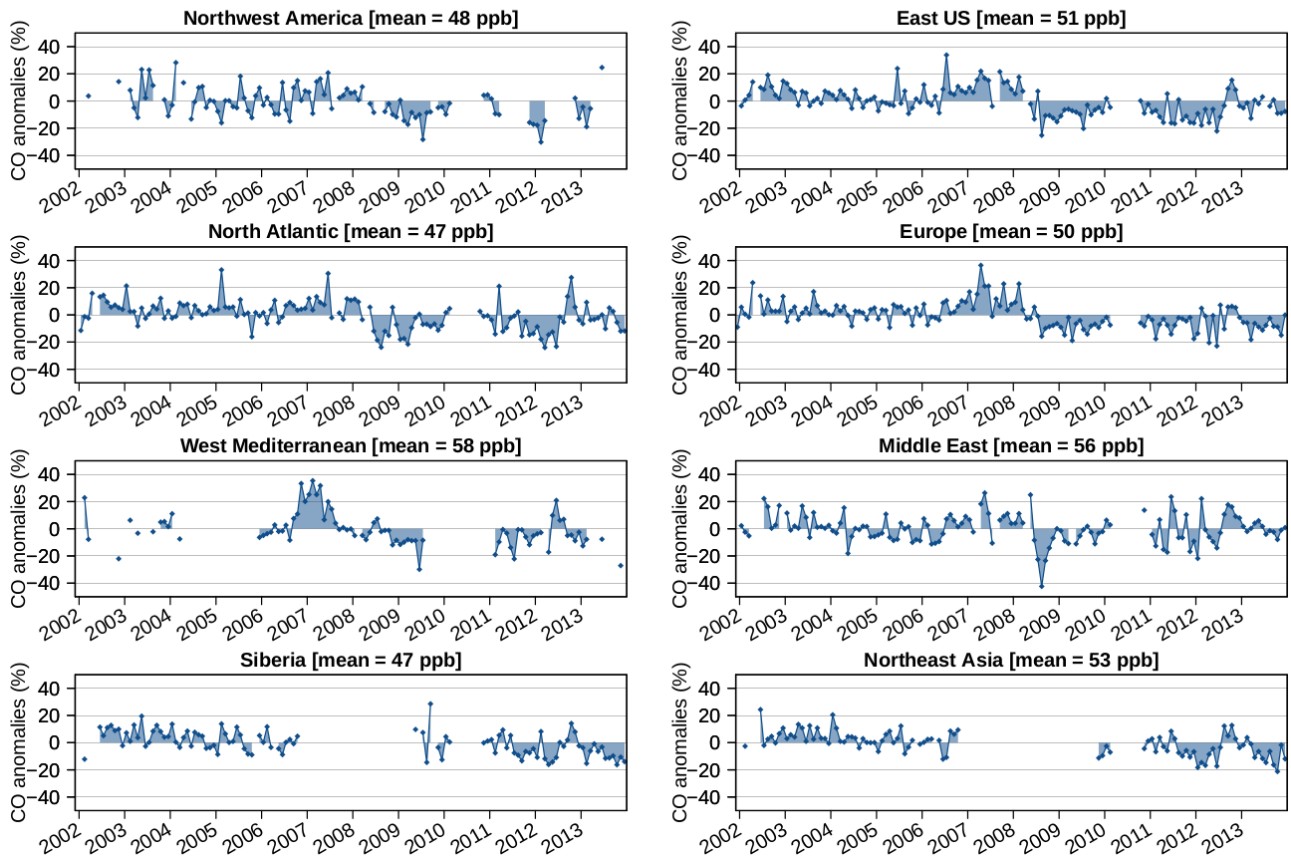

**Figure C4.** Relative monthly anomalies of CO volume mixing ratio in the lower stratosphere. The units % refer to the corresponding climatological means indicated above each graphic.

## Appendix D: Tables of trends

**Table D1.** $O_3$ absolute trends in ppb.yr$^{-1}$ during all the year in the UT. From left to right, the anomalies are taken from [..[470] ]deseasonalized P05, monthly means and P95. The insignificant trends with respect to the 95 % confidence level are indicated as NS.

| SEASON | REGION | P05 | | | MEAN | | | P95 | | |
|---|---|---|---|---|---|---|---|---|---|---|
| (ppb.yr$^{-1}$) | | Slope | Conf. int. (95%) | p-value | Slope | Conf. int. (95%) | p-value | Slope | Conf. int. (95%) | p-value |
| ANN | EUS | 0.44 | [0.19;0.64] | 0.003 | 0.34 | [0.09;0.50] | 0.013 | | NS | |
| | NAt | 0.43 | [0.18;0.62] | 0.003 | | NS | | | NS | |
| | Eur | 0.35 | [0.18;0.50] | $<10^{-3}$ | 0.31 | [0.14;0.45] | 0.003 | 0.29 | [0.06;0.47] | 0.022 |
| | WMed | 0.34 | [0.08;0.52] | 0.003 | 0.42 | [0.14;0.64] | 0.003 | 0.64 | [0.06;1.16] | 0.04 |
| | MidE | 0.30 | [0.12;0.48] | 0.007 | 0.25 | [0.01;0.45] | 0.037 | | NS | |
| | Sib | 0.50 | [0.31;0.67] | $<10^{-3}$ | 0.37 | [0.12;0.58] | 0.007 | | NS | |
| | NEAs | 0.57 | [0.35;0.80] | $<10^{-3}$ | 0.45 | [0.23;0.68] | $<10^{-3}$ | | NS | |

**Table D2.** $O_3$ absolute trends in ppb.yr$^{-1}$ during each season in the UT. From left to right, the anomalies are taken from [..[471]]deseasonalized P05, monthly means and P95. The insignificant trends with respect to the 95 % confidence level are indicated as NS.

| SEASON | REGION | P05 | | | MEAN | | | P95 | | |
|---|---|---|---|---|---|---|---|---|---|---|
| (ppb.yr$^{-1}$) | | Slope | Conf. int. (95%) | p-value | Slope | Conf. int. (95%) | p-value | Slope | Conf. int. (95%) | p-value |
| DJF | EUS | 0.50 | [0.16;0.93] | 0.007 | 0.38 | [0.01;0.61] | 0.04 | | NS | |
| | NAt | 0.54 | [0.34;0.84] | <10$^{-3}$ | 0.43 | [0.13;0.65] | 0.003 | | NS | |
| | Eur | 0.41 | [0.13;0.72] | <10$^{-3}$ | 0.37 | [0.11;0.62] | 0.003 | 0.43 | [0.10;0.88] | 0.008 |
| | WMed | | NS | | | NS | | | NS | |
| | MidE | | NS | | | NS | | | NS | |
| | Sib | 0.50 | [0.11;0.91] | 0.02 | | NS | | | NS | |
| | NEAs | 0.34 | [0.12;0.76] | 0.003 | | NS | | | NS | |
| MAM | EUS | | NS | | | NS | | | NS | |
| | NAt | 0.63 | [0.09;1.03] | 0.028 | | NS | | | NS | |
| | Eur | | NS | | | NS | | | NS | |
| | WMed | 0.53 | [0.10;1.11] | 0.037 | | NS | | | NS | |
| | MidE | 0.48 | [0.15;1.00] | 0.017 | | NS | | | NS | |
| | Sib | 0.56 | [0.24;1.00] | 0.018 | 0.50 | [0.11;0.88] | 0.023 | | NS | |
| | NEAs | 0.56 | [0.17;0.82] | 0.007 | | NS | | | NS | |
| JJA | EUS | | NS | | | NS | | | NS | |
| | NAt | | NS | | | NS | | | NS | |
| | Eur | | NS | | | NS | | | NS | |
| | WMed | | NS | | | NS | | | NS | |
| | MidE | | NS | | | NS | | | NS | |
| | Sib | 0.60 | [0.42;0.91] | <10$^{-3}$ | | NS | | | NS | |
| | NEAs | 1.07 | [0.58;1.42] | 0.003 | 0.84 | [0.23;1.36] | 0.01 | | NS | |
| SON | EUS | 0.44 | [0.05;0.76] | 0.012 | 0.47 | [0.11;0.64] | 0.013 | | NS | |
| | NAt | | NS | | | NS | | | NS | |
| | Eur | 0.40 | [0.00;0.64] | 0.045 | 0.35 | [0.04;0.64] | 0.033 | | NS | |
| | WMed | 0.38 | [0.16;0.54] | 0.003 | 0.47 | [0.11;0.67] | 0.02 | | NS | |
| | MidE | 0.26 | [0.05;0.52] | 0.023 | 0.35 | [0.09;0.69] | 0.01 | | NS | |
| | Sib | | NS | | | NS | | | NS | |
| | NEAs | | NS | | 0.38 | [0.04;0.67] | 0.043 | | NS | |

**Table D3.** $O_3$ relative trends in $\%.yr^{-1}$ during all the year in the UT. From left to right, the anomalies are taken from [..[472] ]deseasonalized P05, monthly means and P95. The insignificant trends with respect to the 95 % confidence level are indicated as NS.

| SEASON | REGION | P05 | | | MEAN | | | P95 | | |
|--------|--------|------|------|------|------|------|------|------|------|------|
| $(\%.yr^{-1})$ | | Slope | Conf. int. (95%) | p-value | Slope | Conf. int. (95%) | p-value | Slope | Conf. int. (95%) | p-value |
| ANN | EUS | 1.10 | [0.48;1.61] | 0.003 | 0.55 | [0.14;0.80] | 0.013 | | NS | |
| | NAt | 1.10 | [0.45;1.59] | 0.003 | | NS | | | NS | |
| | Eur | 0.88 | [0.46;1.24] | $<10^{-3}$ | 0.51 | [0.23;0.75] | 0.003 | 0.33 | [0.07;0.53] | 0.022 |
| | WMed | 0.82 | [0.19;1.28] | 0.003 | 0.67 | [0.23;1.02] | 0.003 | 0.70 | [0.07;1.26] | 0.04 |
| | MidE | 0.68 | [0.27;1.08] | 0.007 | 0.39 | [0.01;0.70] | 0.037 | | NS | |
| | Sib | 1.06 | [0.66;1.42] | $<10^{-3}$ | 0.58 | [0.19;0.91] | 0.007 | | NS | |
| | NEAs | 1.26 | [0.77;1.78] | $<10^{-3}$ | 0.69 | [0.35;1.04] | $<10^{-3}$ | | NS | |

**Table D4.** O$_3$ relative trends in %.yr$^{-1}$ during each season in the UT. From left to right, the anomalies are taken from [..[473] ]deseasonalized P05, monthly means and P95. The insignificant trends with respect to the 95 % confidence level are indicated as NS.

| SEASON | REGION | P05 | | | MEAN | | | P95 | | |
|---|---|---|---|---|---|---|---|---|---|---|
| (%.yr$^{-1}$) | | Slope | Conf. int. (95%) | p-value | Slope | Conf. int. (95%) | p-value | Slope | Conf. int. (95%) | p-value |
| DJF | EUS | 1.24 | [0.39;2.32] | 0.007 | 0.62 | [0.01;0.99] | 0.04 | | NS | |
| | NAt | 1.38 | [0.87;2.16] | <10$^{-3}$ | 0.74 | [0.22;1.12] | 0.003 | | NS | |
| | Eur | 1.02 | [0.33;1.79] | <10$^{-3}$ | 0.61 | [0.19;1.03] | 0.003 | 0.48 | [0.11;0.99] | 0.008 |
| | WMed | | NS | | | NS | | | NS | |
| | MidE | | NS | | | NS | | | NS | |
| | Sib | 1.07 | [0.23;1.93] | 0.02 | | NS | | | NS | |
| | NEAs | 0.75 | [0.27;1.70] | 0.003 | | NS | | | NS | |
| MAM | EUS | | NS | | | NS | | | NS | |
| | NAt | 1.61 | [0.22;2.63] | 0.028 | | NS | | | NS | |
| | Eur | | NS | | | NS | | | NS | |
| | WMed | 1.30 | [0.25;2.71] | 0.037 | | NS | | | NS | |
| | MidE | 1.10 | [0.34;2.27] | 0.017 | | NS | | | NS | |
| | Sib | 1.20 | [0.50;2.13] | 0.018 | 0.78 | [0.17;1.38] | 0.023 | | NS | |
| | NEAs | 1.24 | [0.37;1.83] | 0.007 | | NS | | | NS | |
| JJA | EUS | | NS | | | NS | | | NS | |
| | NAt | | NS | | | NS | | | NS | |
| | Eur | | NS | | | NS | | | NS | |
| | WMed | | NS | | | NS | | | NS | |
| | MidE | | NS | | | NS | | | NS | |
| | Sib | 1.28 | [0.90;1.93] | <10$^{-3}$ | | NS | | | NS | |
| | NEAs | 2.37 | [1.29;3.15] | 0.003 | 1.30 | [0.35;2.10] | 0.01 | | NS | |
| SON | EUS | 1.10 | [0.12;1.91] | 0.012 | 0.76 | [0.18;1.03] | 0.013 | | NS | |
| | NAt | | NS | | | NS | | | NS | |
| | Eur | 1.00 | [0.00;1.60] | 0.045 | 0.59 | [0.06;1.06] | 0.033 | | NS | |
| | WMed | 0.92 | [0.38;1.31] | 0.003 | 0.74 | [0.17;1.07] | 0.02 | | NS | |
| | MidE | 0.60 | [0.12;1.19] | 0.023 | 0.55 | [0.14;1.08] | 0.01 | | NS | |
| | Sib | | NS | | | NS | | | NS | |
| | NEAs | | NS | | 0.58 | [0.06;1.03] | 0.043 | | NS | |

**Table D5.** CO absolute trends in ppb.yr$^{-1}$ during all the year in the UT. From left to right, the anomalies are taken from [..[474]]deseasonalized P05, monthly means and P95. The insignificant trends with respect to the 95 % confidence level are indicated as NS.

| SEASON (ppb.yr$^{-1}$) | REGION | P05 | | | MEAN | | | P95 | | |
|---|---|---|---|---|---|---|---|---|---|---|
| | | Slope | Conf. int. (95%) | p-value | Slope | Conf. int. (95%) | p-value | Slope | Conf. int. (95%) | p-value |
| ANN | EUS | -1.37 | [-1.92;-0.75] | <10$^{-3}$ | -1.53 | [-2.08;-0.98] | <10$^{-3}$ | -2.12 | [-2.80;-1.10] | <10$^{-3}$ |
| | NAt | -0.98 | [-1.77;-0.45] | 0.003 | -1.38 | [-1.98;-0.69] | <10$^{-3}$ | -1.79 | [-2.61;-0.79] | <10$^{-3}$ |
| | Eur | -1.08 | [-1.63;-0.63] | <10$^{-3}$ | -1.41 | [-1.96;-0.90] | <10$^{-3}$ | -1.89 | [-2.47;-1.28] | <10$^{-3}$ |
| | WMed | -0.93 | [-2.02;-0.31] | 0.003 | -0.80 | [-2.06;-0.14] | 0.013 | | NS | |
| | MidE | -0.59 | [-1.17;-0.07] | 0.027 | | NS | | | NS | |
| | Sib | -0.84 | [-1.48;-0.26] | 0.007 | -1.38 | [-2.15;-0.32] | 0.01 | -1.56 | [-2.98;-0.19] | 0.02 |
| | NEAs | -1.22 | [-2.02;-0.56] | <10$^{-3}$ | -2.19 | [-3.33;-0.82] | 0.003 | -3.81 | [-6.75;-1.37] | 0.003 |

**Table D6.** CO absolute trends in ppb.yr$^{-1}$ during each season in the UT. From left to right, the anomalies are taken from [..475 ]deseasonalized P05, monthly means and P95. The insignificant trends with respect to the 95 % confidence level are indicated as NS. Note that the trends in [..476 ]Northwest America are available only in DJF.

| SEASON (ppb.yr$^{-1}$) | REGION | P05 | | | MEAN | | | P95 | | |
|---|---|---|---|---|---|---|---|---|---|---|
| | | Slope | Conf. int. (95%) | p-value | Slope | Conf. int. (95%) | p-value | Slope | Conf. int. (95%) | p-value |
| DJF | WNAm | | NS | | -2.48 | [-4.14;-1.04] | 0.02 | -3.30 | [-6.25;-1.12] | 0.003 |
| | EUS | | NS | | -1.56 | [-3.13;-0.65] | <10$^{-3}$ | -2.42 | [-3.42;-1.42] | <10$^{-3}$ |
| | NAt | | NS | | -1.65 | [-3.19;-0.75] | 0.007 | -2.68 | [-4.21;-1.46] | 0.005 |
| | Eur | -1.24 | [-2.32;-0.37] | 0.01 | -1.49 | [-2.77;-0.67] | 0.007 | -2.15 | [-3.54;-1.19] | 0.007 |
| | WMed | | NS | | | NS | | | NS | |
| | MidE | | NS | | -1.15 | [-3.67;-0.06] | 0.05 | | NS | |
| | Sib | -0.99 | [-2.24;-0.15] | 0.035 | -1.75 | [-2.56;-0.35] | 0.023 | -1.43 | [-3.24;-0.39] | 0.025 |
| | NEAs | -2.51 | [-3.32;-1.55] | <10$^{-3}$ | -2.97 | [-5.00;-0.75] | 0.01 | | NS | |
| MAM | EUS | -1.92 | [-3.03;-1.00] | <10$^{-3}$ | -2.11 | [-3.10;-1.45] | <10$^{-3}$ | -2.66 | [-3.86;-0.99] | 0.017 |
| | NAt | -1.88 | [-3.81;-0.79] | 0.013 | -1.97 | [-3.24;-1.16] | 0.003 | | NS | |
| | Eur | -1.19 | [-2.42;-0.26] | 0.017 | -1.49 | [-3.03;-1.12] | 0.003 | -2.03 | [-3.69;-1.51] | 0.003 |
| | WMed | | NS | | | NS | | | NS | |
| | MidE | | NS | | | NS | | | NS | |
| | Sib | | NS | | | NS | | | NS | |
| | NEAs | -1.54 | [-2.55;-0.02] | 0.05 | -3.42 | [-5.70;-0.53] | 0.03 | | NS | |
| JJA | EUS | -1.39 | [-2.09;-0.85] | <10$^{-3}$ | -1.36 | [-2.05;-0.58] | 0.003 | -1.88 | [-3.38;-0.34] | 0.007 |
| | NAt | | NS | | | NS | | | NS | |
| | Eur | -1.01 | [-1.99;-0.42] | 0.007 | -1.34 | [-2.41;-0.38] | 0.01 | -1.80 | [-2.99;-0.50] | 0.017 |
| | WMed | | NS | | | NS | | | NS | |
| | MidE | | NS | | | NS | | | NS | |
| | Sib | | NS | | | NS | | | NS | |
| | NEAs | | NS | | | NS | | | NS | |
| SON | EUS | | NS | | | NS | | | NS | |
| | NAt | | NS | | | NS | | | NS | |
| | Eur | | NS | | -1.14 | [-1.96;-0.07] | 0.023 | | NS | |
| | WMed | | NS | | -1.23 | [-3.85;-0.17] | 0.04 | -1.29 | [-5.70;-0.40] | 0.018 |
| | MidE | | NS | | | NS | | | NS | |
| | Sib | | NS | | | NS | | | NS | |
| | NEAs | | NS | | | NS | | | NS | |

**Table D7.** CO relative trends in $\%.\text{yr}^{-1}$ during all the year in the UT. From left to right, the anomalies are taken from [..[477]]deseasonalized P05, monthly means and P95. The insignificant trends with respect to the 95 % confidence level are indicated as NS.

| SEASON | REGION | P05 | | | MEAN | | | P95 | | |
|--------|--------|------|----------------|---------|------|----------------|---------|------|----------------|---------|
| $(\%.\text{yr}^{-1})$ | | Slope | Conf. int. (95%) | p-value | Slope | Conf. int. (95%) | p-value | Slope | Conf. int. (95%) | p-value |
| ANN | EUS | -1.74 | [-2.43;-0.95] | $<10^{-3}$ | -1.49 | [-2.02;-0.95] | $<10^{-3}$ | -1.63 | [-2.15;-0.85] | $<10^{-3}$ |
| | NAt | -1.21 | [-2.19;-0.55] | 0.003 | -1.38 | [-1.98;-0.69] | $<10^{-3}$ | -1.47 | [-2.14;-0.65] | $<10^{-3}$ |
| | Eur | -1.39 | [-2.09;-0.81] | $<10^{-3}$ | -1.41 | [-1.96;-0.90] | $<10^{-3}$ | -1.54 | [-2.01;-1.04] | $<10^{-3}$ |
| | WMed | -1.15 | [-2.50;-0.38] | 0.003 | -0.82 | [-2.10;-0.14] | 0.013 | | NS | |
| | MidE | -0.80 | [-1.58;-0.09] | 0.027 | | NS | | | NS | |
| | Sib | -0.99 | [-1.74;-0.30] | 0.007 | -1.31 | [-2.05;-0.30] | 0.01 | -1.21 | [-2.31;-0.15] | 0.02 |
| | NEAs | -1.52 | [-2.53;-0.70] | $<10^{-3}$ | -1.89 | [-2.87;-0.71] | 0.003 | -2.04 | [-3.61;-0.73] | 0.003 |

**Table D8.** CO relative trends in $\%.\text{yr}^{-1}$ during each season in the UT. From left to right, the anomalies are taken from [..[478] ]deseasonalized P05, monthly means and P95. The insignificant trends with respect to the 95 % confidence level are indicated as NS. Note that the trends in [..[479] ]Northwest America are available only in DJF.

| SEASON | REGION | P05 | | | MEAN | | | P95 | | |
|--------|--------|-------|-----------------|---------|-------|-----------------|---------|-------|-----------------|---------|
| $(\%.\text{yr}^{-1})$ | | Slope | Conf. int. (95%) | p-value | Slope | Conf. int. (95%) | p-value | Slope | Conf. int. (95%) | p-value |
| DJF | WNAm | | NS | | -2.23 | [-3.73;-0.94] | 0.02 | -2.29 | [-4.34;-0.78] | 0.003 |
| | EUS | | NS | | -1.51 | [-3.04;-0.63] | $<10^{-3}$ | -1.86 | [-2.63;-1.09] | $<10^{-3}$ |
| | NAt | | NS | | -1.65 | [-3.19;-0.75] | 0.007 | -2.20 | [-3.45;-1.20] | 0.005 |
| | Eur | -1.59 | [-2.98;-0.47] | 0.01 | -1.49 | [-2.77;-0.67] | 0.007 | -1.75 | [-2.88;-0.97] | 0.007 |
| | WMed | | NS | | | NS | | | NS | |
| | MidE | | NS | | -1.25 | [-3.99;-0.07] | 0.05 | | NS | |
| | Sib | -1.17 | [-2.64;-0.18] | 0.035 | -1.67 | [-2.44;-0.33] | 0.023 | -1.11 | [-2.51;-0.30] | 0.025 |
| | NEAs | -3.14 | [-4.15;-1.94] | $<10^{-3}$ | -2.56 | [-4.31;-0.65] | 0.01 | | NS | |
| MAM | EUS | -2.43 | [-3.84;-1.27] | $<10^{-3}$ | -2.05 | [-3.01;-1.41] | $<10^{-3}$ | -2.05 | [-2.97;-0.76] | 0.017 |
| | NAt | -2.32 | [-4.70;-0.97] | 0.013 | -1.97 | [-3.24;-1.16] | 0.003 | | NS | |
| | Eur | -1.53 | [-3.10;-0.33] | 0.017 | -1.49 | [-3.03;-1.12] | 0.003 | -1.65 | [-3.00;-1.23] | 0.003 |
| | WMed | | NS | | | NS | | | NS | |
| | MidE | | NS | | | NS | | | NS | |
| | Sib | | NS | | | NS | | | NS | |
| | NEAs | -1.93 | [-3.19;-0.02] | 0.05 | -2.95 | [-4.91;-0.46] | 0.03 | | NS | |
| JJA | EUS | -1.76 | [-2.64;-1.07] | $<10^{-3}$ | -1.32 | [-1.99;-0.56] | 0.003 | -1.45 | [-2.6;-0.26] | 0.007 |
| | NAt | | NS | | | NS | | | NS | |
| | Eur | -1.30 | [-2.55;-0.54] | 0.007 | -1.34 | [-2.41;-0.38] | 0.01 | -1.46 | [-2.43;-0.41] | 0.017 |
| | WMed | | NS | | | NS | | | NS | |
| | MidE | | NS | | | NS | | | NS | |
| | Sib | | NS | | | NS | | | NS | |
| | NEAs | | NS | | | NS | | | NS | |
| SON | EUS | | NS | | | NS | | | NS | |
| | NAt | | NS | | | NS | | | NS | |
| | Eur | | NS | | -1.14 | [-1.96;-0.07] | 0.023 | | NS | |
| | WMed | | NS | | -1.26 | [-3.93;-0.17] | 0.04 | -1.09 | [-4.83;-0.34] | 0.018 |
| | MidE | | NS | | | NS | | | NS | |
| | Sib | | NS | | | NS | | | NS | |
| | NEAs | | NS | | | NS | | | NS | |

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
