# Peer review of "Climatology and long-term evolution of ozone and carbon monoxide in the UTLS at northern mid-latitudes, as seen by IAGOS from 1995 to 2013"

_Atmospheric Chemistry and Physics, 2017_

## Referee Comment (RC1) · Anonymous Referee #1 · 22 Oct 2017

Cohen et al., 2017 ACPD

Major comments: This paper presents the more recent results of IAGOS measurements. There is a lot of data presented, with certain behavior explained. But there is little in the way of science interpretation of these the observations. I found it hard to distill down the main points of the paper. However, I think with a little additional work, this paper could be a lot stronger.

The authors present both O3 and CO data but rarely combine these observations in

their analysis. For example on Pg 16, they discuss that some increased CO in North East Asia may be due to biomass burning but never discuss how these fires would impact O3. Maybe consider the implications of the dual tracers you have here. How does the CO:O3 ratio vary across the UT, TPL and LS. There is no real testing of the definition of TPL using both tracers either, and this could be an important result. The authors compare their O3 and CO trends with other results but don't do any investigation into what is driving those trends. There are a lot of "may" and "could". I think the paper would be a lot stronger if the trends in CO were compared with estimated CO emissions from BB (using GFED?) and anthropogenic emissions. Or included a direct comparison with the MOPITT trends (rather than just discussion).

The paper could be made more concise (maybe the time series plots could go in an SI?). The summary/conclusions section is great and should be the lead in to more of the paper overall.

Technical comments (I'm sure I've missed some) Minor detail: The use of all the acronyms makes this paper really difficult to read. If you could remove some and replace things like NEA with North-eastern Asia, it would not add much length but would make it easier to get through the text.

Pg 1, 6 western Maritime continent? Not sure what this is. Pg 1, 8 You have very little southern hemisphere data and nothing over the pacific in this paper. quasi-global is an overestimate.

Pg 2, 14: clarify what you mean by NMVOCs Pg 2, 16: emitted by lightning? Do you mean produced by lightning? Pg 4, 30: I like this section. I was wondering about it. Pg 5, 30: What is the uncertainty in the PV calculation? How much would this uncertainty affect the partitioning of data between UT, LS, and tropopause? Pg 6, 30: Do you really mean data points? Or do you mean 5ox5o binned data? This also invites the question: As a rough idea, many O3 (4s) and CO (30s) data points do you fit into a 5o x 5o box? Pg 6, 29: I think the use of 7 months and 3 seasons is a nice compromise

between representativeness and data coverage. Do you get the same results if you used 6 or 9 months or 4 seasons (the statistics probably won't be robust but it could act as a sensitivity study)? Pg 6, 33: I haven't heard for the Theil-Sen slope estimate or the OpenAir package before so I can't comment on these. PG 7 12; "Except this one". Do you mean "Aside from this region, the next least sampled regions are..." Pg 9, 10: What longitudinal averaging is done in the 25-75 bands? Or is this all the data the data or all the data averaged over 5o (the grid basis)? Pg 9, 14: "eastern Asia and northern Pacific". You dont have any northern Pacific data included in the map. Stick with the regions you have identified (NEA) Pg 9, 30: Could you make the comparison with the satellites a new paragraph and extend it a little? This seems to just be tacked on at the end here. Pg 11: These graphs need axis points. You refer to gradients between e.g. 60E to 135E but I have to guess where those are on the plots. Pg 11, Fig 3: Also need lat/long labels on this figure. Pg 12, 9 Definition of STE? Pg 15, 5 How many years of CO data is included? Pg 15, 7 This should not be a new paragraph. You are discussing the same figure Pg 15, 10 "In most re- gions, there is no noticeable seasonal variation in the LS." Really? At least half of them show a similar range of seasonal changes in the UT and TPL. Which begs the question: What is the uncertainty/range of variability within each of the monthly mean values shown in Fig 5? Are any of these regional seasonal trends statistically different? Pg 16, 5 What is the uncertainty range on all these means? Are the results statistically different? Pg 16, 15 Definition of WCB? Pg 16, 21 "percentile 95" What is this? How did you calculate it? What data was used to define it? There needs to be some detail on this here (or did I miss something earlier?!) Pg 16, 22 What is the uncertainty/variability on these twin "peaks". Are the differences between April, May and June statistically significant so that you can call this a double peak? How much of this difference could be attributed to uncertainty in the calculation of the TPL height? Pg 16, 29 Can you check GFED for the timing delay between peak fires and the peak you see? The peak fire season has been shifting towards August in recent years so has the CO signal from BB also been delayed in the UT? You also don't define GFED or GFAS. Pg 17, 25 "All these

features seem to indicate less frequent springtime STE events in WNAm." You never explained what STE is so I've no idea what this is. This whole paragraph needs to be re-written in simpler language. P5 and P95 have not been explained so far aside from the abstract. Pg 25, 12 "mentioned" mis-spelled throughout paper Pg 25, 12 what years did the Thouret paper cover? 9 years is a bit vague. Pg 25, 23 "a more frequent sampling, thus smoothing the temporal variability". This doesn't make sense. Do you mean you have more data available within the greater sized grid box, allowing for a more significant statistical analysis? Pg 27, 24 "We assume that in the UT, the spatial variability is too weak to be responsible for the discrepancies between the two studies." Could you use the longitudinal trend from Fig 2/3 do quantify the expected changes? How significant is the P95 trend you see? Pg 27, 25 do you mean O3 increasing over time? Pg 27, 34 what O3 trend do you get using only 1994-2008? Pg 27, 32 "Although no trend calculation was performed in WNAm, it may confirm the upper tropospheric O3 increase as effective in the whole northern mid-latitudes, except the Pacific ocean." If there was no trend calculated due to a lack of data, then this sentence should not be included. You can't confirm or deny anything! Pg 27, 35 Fig 8 shows an increase in O3 in the UT because there is missing data! How is the mean/smoothing calculated for each region when there is a data gap? 2011 probably has one of the lowest O3 minimums with missing data before. Pg 28, 3 "mentionned" should be mentioned Pg 32, Fig 16, 17: What do the hatched areas mean? (I assume insignificant trend but you should state it). You should also explain in one of these figures what regions each of the labels represented. All info to interpret the figure should be in the caption. Specifically to Fig 16: What do you gain from including the 5%, 50% and 95% trends? How is computed? Are the 95% CI for the Pg 34, 2: extend should be extent

---

## Referee Comment (RC2) · Anonymous Referee #2 · 28 Oct 2017

Review of Atmos. Chem. Phys. Discuss., https://doi.org/10.5194/acp-2017-778

Cohen et al., Climatology and long-term evolution of ozone and carbon monoxide in the UTLS at northern mid-latitudes, as seen by IAGOS from 1995 to 2013

This paper presents a decadal scale record of ozone and carbon monoxide measurements that is used in the Tropospheric Ozone Assessment Report (TOAR) and will be critical in the evaluation of chemistry climate models that require accurate atmospheric composition for understanding emissions, chemistry, transport and radiative forcing. I

think the basic methods for climatology and trend analysis are sound. I agree with Referee #1 that more science motivation could be added to the paper and would require only minor revisions.

Regarding O3 trends and sources of variability, the following references could be included:

Lin, M., A. M. Fiore, L. W. Horowitz, A. O. Langford, S. J. Oltmans, D. Tarasick, and H. E. Rieder (2015a), Climate variability modulates western U.S. ozone air quality in spring via deep stratospheric intrusions, Nature Communications, 6 (7105), doi: 10.1038/ncomms8105.

Lin, M., W. Horowitz, R. Payton, A.M. Fiore, G. Tonnesen (2017). US surface ozone trends and extremes from 1980 to 2014: Quantifying the roles of rising Asian emissions, domestic controls, wildfires, and climate. Atmos. Chem. Phys., doi:10.5194/acp-17-2943-2017

Wespes, C., D. Hurtmans, C. Clerbaux, and P.-F. Coheur (2017), O3 variability in the troposphere as observed by IASI over 2008–2016—Contribution of atmospheric chemistry and dynamics, J. Geophys. Res. Atmos., 122, 2429–2451, doi:10.1002/2016JD025875.

Specific comments:

P1, Line 15: Since trends were only computed for the northern mid-latitudes, perhaps this should say: ". . .to derive trends in the northern mid-latitude UTLS"

P2, Line 20: "It is also important for enhancing the knowledge about the role of O3 in the increasing atmospheric radiative forcing." CO emissions also contribute to increasing RF. Maybe say: "Trends in both gases are also important for assessing changes in atmospheric radiative forcing."

P5, Line 15: Define TOAR acronym

P5, Line 25: Do you ever find double tropopause cases? If so, are these included or filtered out? These can be very common (50-70%) in winter in the N. Midlatitudes [Randel et al., JGR, 2007: doi:10.1029/2006JD007904], but they might be hard to detect with 1°x1° PV.

P7, Line 1: It would be helpful to say that a 95% confidence level corresponds to trends that are > 2-sigma.

P9, Line 19: warm conveyor belts = WCBs?

P12, Line 10: "The discrepancies with the IAGOS climatologies can be due to uncertainties involving the stratospheric signal, i.e. the ozone stratospheric column, the height of the tropopause, and the total ozone column" I don't understand why "total column ozone" is in this list. Don't you mean the differences between the full tropospheric column and only the UT column?

P12, Line 25: "Except for India where few summertime IAGOS data do not allow the comparison" Are there less flights in summer? Should be changed to "Except for India where summertime IAGOS data is limited due to [explanation]

P29,32 Figs 14,15,16,17 caption should state that bars with faded colors (and no error bars) did not have significant trends.

---

## Author Comment (AC1) · 24 Jan 2018

We are thankful to the reviewers for their positive and accurate feedbacks on our study, and for the improvements they allowed us to perform. Please find each referee's comment with the corresponding author's response in the following text. Regards.

Text from the reviewers is in blue. Our answers are in black, and the changes proposed for the revised manuscript are in italic (black for modified sentences, grey for unchanged sentences that have been pasted here in order to remind the context).

Referee #1:

**Major comments: This paper presents the more recent results of IAGOS measurements. There is a lot of data presented, with certain behavior explained. But there is little in the way of science interpretation of these the observations. I found it hard to distill down the main points of the paper. However, I think with a little additional work, this paper could be a lot stronger.**

**The authors present both $O_3$ and CO data but rarely combine these observations in their analysis. For example on Pg 16, they discuss that some increased CO in North East Asia may be due to biomass burning but never discuss how these fires would impact $O_3$. Maybe consider the implications of the dual tracers you have here. How does the CO:$O_3$ ratio vary across the UT, TPL and LS.**

The revised manuscript will propose a deeper scientific interpretation by including a further analysis of the $O_3$/CO ratio in particular. Discussing the origin of positive CO anomalies in Northeast Asia is relatively straightforward in terms of biomass burning but discussing how these fires would affect ozone is not. Ozone production in the UT, as everywhere else, is not linear and depends on many factors that only a global CTM can integrate. Quantifying such impact is thus beyond the scope of this paper. With this article we aim to provide a useful data set describing the main characteristics of regional and interannual variabilities of ozone and CO in the UT, TPL and LS. A natural perspective of this paper is to check how CTMs reproduce such characteristics.

Additional results on the $O_3$/CO ratio are now available in the revised manuscript and bring a great piece of information into the study. The new results and their analysis have been added to the article. The following changes have been done:

- The horizontal distributions of the $O_3$/CO ratio have been added in Appendix as the new Fig. A1. The comments added in the paper are shown below:

[revised manuscript text omitted]

Concerning the seasonal cycles of the $O_3$/CO ratio, the changes are shown in the following text, from the revised paper:

*3.2 Regional seasonal cycles*

*In order to further assess the regional variability of $O_3$ and CO mixing ratios, we analyzed the time series of the eight regions displayed in Fig. 1 and defined in Tab. 1. We first compare the mean seasonal cycles, before characterizing and analyzing the anomalies and then, derive the trends. In order to make a first estimate of the inter-regional variability of the two trace gases, the mean seasonal cycles are displayed in Fig. 6. Similarly, we show the seasonal cycles of the $O_3$-to-CO ratio in order to provide a synthesis between the two data sets. First of all, as a support to our analysis, we present the seasonal cycles for the mean pressure at the 2 pvu altitude in Fig. 5.*

*[...]*
*Complementary information is shown in Figs. 7, 8 and 9 representing the seasonal cycles of $O_3$, CO and $O_3$/CO respectively, for the monthly fifth percentile (P5), mean value and 95th percentile (P95). The interannual variability is illustrated by the error bars.We first present the main characteristics in 
[revised manuscript text omitted]
 upper tropospheric seasonal cycles of the $O_3$/CO exhibit lower values (0.7) in these two Pacific regions compared to elsewhere (at least 0.1 below). It is mainly characterized by a lower fifth percentile (0.4), consistent with a higher frequency of high-CO and poor-$O_3$ air masses. However, in contrast with ozone, the seasonal cycles of the $O_3$/CO ratio do have a summer maximum. The interruption observed in the $O_3$ seasonal cycles thus**

*remains characterized by the summer maximum in photochemical activity, despite the fast change in the monthly air composition.*

[...]

*West Mediterranean basin and Middle East*
*As expected in Fig. 7 (Fig. 8), the mean $O_3$ (CO) concentrations in the LS are lower (higher) in West Mediterranean basin, Middle East and Northeast Asia, which are the southernmost regions of this study. Indeed, as the dynamical tropopause is generally higher there (Fig. 5), although the flights are classified LS, they are likely to sample air masses closer to the tropopause: the substantial difference in $O_3$ mixing ratios is thus explained by its strong vertical gradient in the stratosphere.*

*During July-August, three categories can be established depending on the width of the monthly distribution of the $O_3$/CO ratio. Referring to Fig. 9, the northernmost regions (Northwest America, North Atlantic, west Europe, Siberia) exhibit a higher $O_3$/CO ratio for all the three metrics (mean value, fifth and 95th percentiles). On the opposite, the southernmost regions (West Mediterranean basin, Middle East and Northeast Asia) show a lower ratio, for all the three metrics again. The remaining region (East United States) shows a strong intra-monthly variability with a low fifth percentile, an intermediate mean value and a high 95[th] percentile. These categories thus correspond respectively to regions mostly impacted by extratropical air masses, to regions strongly impacted by subtropical air masses and to the region influenced by both extratropical and subtropical air masses. In June, Northeast Asia belongs to this intermediate category with its low 5[th] and high 95[th] percentiles, before reaching the southernmost regions category in July.*

*In summer, CO mixing ratios are similar over West Mediterranean basin and Middle East, but Middle East $O_3$ is significantly lower. This feature is consistent with the mixing ratios derived from OMI-MLS observations at 150 hPa presented in Park et al. (2007) and at 100 hPa in Park et al. (2009). In southern Asia, polluted surface air masses are uplifted by deep convection during the Asian Monsoon, up to the tropical UT. The western part of the anticyclonic circulation then transports poor-$O_3$ air masses northward and horizontally (Barret et al., 2016). This may impact the LS in Middle East (Park et al., 2007), consistent with CH4 measurements from IASI and AIRS, coupled with modelling using the CNRM-AOCCM, LMDz-OR-INCA CCMs and the MOCAGE CTM (Ricaud et al., 2014). The summertime lower stratospheric CO mixing ratio is comparable between West Mediterranean basin, Middle East and Northeast Asia. The last one is impacted by frequent warm conveyor belts coupled with a strong fire activity (Madonna et al., 2014; Jiang et al., 2017).*
*In the UT, there is more $O_3$ and less CO in the West Mediterranean basin and Middle East.* **Figure 9 better illustrates the distinction of these two regions from the others, with a high $O_3$/CO ratio. Its mean value reaches 0.95 in July in the two regions in the UT, and is both characterized by higher fifth and 95[th] percentiles. Note that $O_3$/CO is also higher in the TPL (~ 1.5). All these features show the impact of the stronger summertime subsidence at these latitudes on the whole monthly distribution in the UT.** *In Middle East particularly, Etesian winds interact with the Asian Monsoon Anticyclone (AMA), enhancing the subsidence of high-level air masses (e.g. Tyrlis et al., 2013) thus allowing a recurrent summertime $O_3$ pool down to the mid-troposphere (Zanis et al., 2014).*

**There is no real testing of the definition of TPL using both tracers either, and this could be an important result.**

The objective of this study is to use the most commonly accepted definition of the dynamical tropopause to extract distinct distributions with distinct characteristics (UT, TPL, and LS) from the IAGOS data recorded between 9 and 12 km at cruise altitude. Thus we aim at characterizing these distinct reservoirs in terms of ozone and CO regional/interannual/decadal variabilities. TPL is indeed defined as a 30 hPa thick layer on purpose to take into account the

variability or errors associated to the definition of the tropopause. Thouret et al. (2006) have originally described this methodology. In particular, the sensitivity to the value of the PV iso-surface (between 1.5 and 4 pvu) has been evaluated. The monthly mean ozone seasonal cycle was the criteria used to test the consistency of this TPL definition : the selected definition is the one allowing (i) the UT to highlight a clear summer maximum, (ii) the LS to highlight a clear spring maximum, and (ii) the TPL to exhibit an intermediate behavior.

This coherence with the seasonal cycles of ozone is also verified now with the seasonal cycles of the $O_3$/CO ratio (new Figs. 6 and 9 in the revised manuscript).

**The authors compare their $O_3$ and CO trends with other results but don't do any investigation into what is driving those trends. There are a lot of "may" and "could". I think the paper would be a lot stronger if the trends in CO were compared with estimated CO emissions from BB (using GFED?) and anthropogenic emissions.**

We agree with the reviewer and the use of "may" and "could" has been reduced in the revised manuscript. Comparing estimated CO emissions to concentrations in the UTLS in order to understand such trends is not straightforward as we miss information on the history of transport, the secondary production source of CO (that represents about a half of its budget) and its oxidation by OH. Such deeper analysis is beyond the scope of the paper as this is the subject for another publication in preparation. This incoming paper takes advantage on the new developed application within the IAGOS data base (SOFT-IO, Sauvage et al., ACP 2017) which links all the observed CO anomalies to a source contribution via the FLEXPART lagrangian model coupled to the ECCAD data base for emissions inventories.

**Or included a direct comparison with the MOPITT trends (rather than just discussion).**

A direct comparison with MOPITT is beyond the scope of this paper because (i) the CO trends from MOPITT have already been extensively investigated (e.g., Worden et al., 2013), and (ii) the direct comparison between MOPITT (passive remote sensing with a coarse vertical resolution) and IAGOS (in situ measurements in the UTLS) is not straightforward. The most important message here is to highlight that the trends estimated in the UTLS based on the IAGOS dataset are consistent with those obtained in the troposphere with MOPITT and other satellites (i.e. differences of trends remain statistically insignificant).

**The paper could be made more concise (maybe the time series plots could go in an SI?).**

We have revised the entire manuscript with the objective to make it more concise. Following your recommendation, the monthly time series have been moved in the appendix. The associated comments remain in the text, because we find important to highlight some advantages and weaknesses in the IAGOS time series. We just added the following sentence as introduction of their description:

*" For a more complete description of the dataset, the corresponding time series for both species are given for the three layers in Figs. B1 – B6 in Appendix."*

**The summary/conclusions section is great and should be the lead in to more of the paper overall.**

We thank the reviewer for this comment. We have revised the manuscript with this in mind, to make it clearer/more focused. The summary/conclusion has also been reshaped. The whole new conclusion is shown below.

[revised manuscript text omitted]

**Technical comments (I'm sure I've missed some) Minor detail:**

**The use of all the acronyms makes this paper really difficult to read. If you could remove some and replace things like NEA with North-eastern Asia, it would not add much length but would make it easier to get through the text.**

Reviewer is right, this will certainly add some clarity to the article. The acronyms that are not frequently used are now replaced by full names, as well as the names of the regions. The acronyms used for the layers (UT, TPL and LS) have also been replaced by their full names in the introduction section.

**Pg 1, 6 western Maritime continent? Not sure what this is.**

By "western Maritime Continent", we only meant "the western half of Maritime Continent". Roughly, it corresponds to an area extending from Borneo to Sumatra and to southeastern Asia.

This has been made clearer in the revised manuscript.

**Pg 1, 8 You have very little southern hemisphere data and nothing over the pacific in this paper. quasi-global is an overestimate.**

Reviewer is right. This adjective has been replaced by "semi-global" in the revised manuscript.

**Pg 2, 14: clarify what you mean by NMVOCs**

Non-methane volatile organic compounds. Every acronym is now written in full letters at least once in the text (at first occurrence).

**Pg 2, 16: emitted by lightning? Do you mean produced by lightning?**

Yes. The new sentence is:

*"Third, nitrogen oxides (NOx) implicated in the production of $O_3$ are not only emitted at the surface by combustion processes, but also produced in the free troposphere by lightning."*

**Pg 5, 30: What is the uncertainty in the PV calculation? How much would this uncertainty affect the partitioning of data between UT, LS, and tropopause?**

Uncertainty in the PV calculation must be understood as uncertainties in the ECMWF analysis themselves due to the resolution of the model and interpolation on the IAGOS flight tracks. Given the strong vertical gradients of PV values around the tropopause, we have decide to apply a 30 hPa (about 700 m at this altitude) thickness around the 2pvu iso-surface to consider this uncertainty in PV calculation that we cannot assess. Besides, defining such a thick tropopause prevents any impact on the partitioning of data between UT, LS and TPL.
It is worth noting that we do focus neither on the composition of the TPL, nor on its structure. Since its composition is strongly impacted by both tropospheric and stratospheric air masses, the TPL itself is defined in a way it represents a transition layer or an "uncertainty region". Again, the objective of this method is to analyse the composition in the two distinct reservoirs UT and LS.

The following text has been added in order to clarify the aim of the methodology.
*"The choice of a transition layer rather than a surface for the tropopause definition is motivated by the need to lower the impact of the vertical uncertainties on the PV calculations. With a thickness as great as 30 hPa, the TPL is then defined as a transition layer allowing a partitioning clearly characteristic of the UT and of the LS. Thus, the analysis focuses on the composition and trends in these two distinct layers. Note also that a comparison between the dynamical tropopause estimate based on the PV fields, and the chemical tropopause estimate based on $O_3$ vertical profiles considering all IAGOS measurements near Frankfurt (where IAGOS data are the most numerous), have shown some errors but a negligible systematic bias (Petetin et al., 2016)."*

**Pg 6, 30: Do you really mean data points? Or do you mean 5ox5o binned data?**

Monthly means are calculated by averaging all data points available in the region, and not the 5°x5° binned data. We added this precision in the text. In order to be more concise, we also simplified the first part of this section as follows :

*"2.2.2 IAGOS data analysis*

*Horizontal distributions of $O_3$ and CO are computed by gathering all IAGOS measurements recorded into $5° \times 5°$ cells, with respect to the season and the layer. Averages are calculated only in cells where the amount of observations exceeds 2,000 (1,000) for $O_3$ (CO) over the period 1995-2013 (2002-2013). The horizontal distribution of $O_3$/CO ratios are also computed by averaging the instantaneous $O_3$/CO ratios in each cell. The selected size of the cells and thresholds in the amount of data ensure a good representativity of the time period, without filtering a too high proportion of data.*

*In eight regions of interest (see Fig. 1), monthly time series are calculated by averaging for each month all individual observations available in the region (not the 5° * 5° binned data). These monthly averages are calculated only when (i) the amount of available data points in a month exceeds 300, and (ii) the first and last measurements are separated by a 7-day period at least. These selected thresholds limit the influence of short-term events which are not representative of the whole month. The boundaries of each region are chosen as a compromise between a high level of sampling and a good representativity of the expected impact of local surface emissions and meteorological conditions. Based on these monthly time series, mean seasonal cycles are computed over the period 1995-2013. In order to avoid inter-seasonal biases due to the interannual variability, we retained only the years with data available during at least 7 months distributed over at least 3 seasons. Requiring available data for all 4 seasons slightly reduces the amount of complete years (2 years or less removed in each region's seasonal cycle), yielding quasi-identical results. The seasonal cycles have also been shown to be poorly sensitive to a 2-month change in the amount of required months per year.*

*These mean seasonal cycles in all regions are used to extract deseasonalized monthly anomalies from the original monthly time series. Besides the mean mixing ratios, other useful metrics like the $5^{th}$ and $95^{th}$ percentiles of the $O_3$ and CO mixing ratios are also calculated, based on the same population of data points than used for calculating the mean.*

*Based on these monthly anomalies, the trends are computed using the non-parametric Mann-Kendall analysis combined with the Theil-Sen slope estimate (Sen, 1968). […]"*

**This also invites the question: As a rough idea, many $O_3$ (4s) and CO (30s) data points do you fit into a 5o x 5o box?**

Given the speed of the aircraft at cruise altitudes, 4 seconds correspond to 1 km horizontally. Depending on the flight track into the 5°x5° cell and the latitude of the cell, we can fit up to several hundred ozone and CO single measurements per flight per box, at mid-latitudes.

**Pg 6, 29: I think the use of 7 months and 3 seasons is a nice compromise between representativeness and data coverage. Do you get the same results if you used 6 or 9 months or 4 seasons (the statistics probably won't be robust but it could act as a sensitivity study)?**

The sensitivity of the seasonal cycles to these filtering parameters is very low. We tested the choice of 2 and 4 seasons as yearly thresholds, the results are quasi-unchanged. We also tested the influence of a 2-month change in the threshold. Only the 9-month threshold caused a 25 ppb decrease in ozone during the April peak in the $5^{th}$ percentile in Northwest America. Still, the ozone P5 remains high during spring in this region.

**Pg 6, 33: I haven't heard for the Theil-Sen slope estimate or the OpenAir package before so I can't comment on these.**

This non-parametric approach is commonly used in the literature for calculating trends, also in atmospheric sciences.

**PG 7 12; "Except this one". Do you mean "Aside from this region, the next least sampled regions are…"**

Exactly. Thank you for this clearer formulation. The manuscript has been modified accordingly.

**Pg 9, 10: What longitudinal averaging is done in the 25-75 bands? Or is this all the data the data or all the data averaged over 5o (the grid basis)?**

It means that for each 5° longitude interval, the 5°*5° mean values are averaged meridionally, from the cell at 25-30N up to the cell at 70-75N.

We reformulated the explanation in the manuscript:

*"By averaging over the latitude range 25 - 75°N all the cells for each 5° interval in longitude, we obtain a longitudinal distribution that is projected below each map. It allows to examine the longitudinal gradients of ozone or CO, and to highlight an intercontinental differences. The averaging interval has been chosen in order to include all upper tropospheric IAGOS measurements in the area considered here as the extratropics."*

**Pg 9, 14: "eastern Asia and northern Pacific". You dont have any northern Pacific data included in the map. Stick with the regions you have identified (NEA)**

To be more precise, we replaced "northern Pacific" by "Bering Sea" (where there are some IAGOS data in summer along the flight corridor linking Asia to northern America). Prior to investigating in more detail the $O_3$ and CO variability and trends in the eight regions, we think that it is important and valuable to give an overview of the climatological horizontal distribution at the global scale.

**Pg 9, 30: Could you make the comparison with the satellites a new paragraph and extend it a little? This seems to just be tacked on at the end here.**

As suggested by Referee #1, this comparison is now in a new paragraph, in a more complete description. As follows:

*"The IAGOS CO data set is complementary to several satellite-based instruments which have different vertical sensitivity profiles, for providing vertical information. Laat et al. (2010) present the horizontal distribution of yearly averaged CO column from SCIAMACHY during 2004 - 2005 (see their Fig. 7, top left panel), with an almost vertically uniform sensitivity profile. The maxima observed by IAGOS over central Africa and over Southeast Asia, the latter during summer, are well represented. Both reach more than 2.75 $10^{18}$ molecules $cm^{-2}$. On the contrary, the fall maximum observed by IAGOS over the Brazilian coast is not visible in the SCIAMACHY columns. All those*

*three (sub)tropical maxima seen by IAGOS are visible in the seasonal climatologies of MOPITT total columns from 2001 to 2012 (Osman et al., 2016) and MLS mixing ratios at 215 hPa (Huang et al., 2016). In the northern extratropics, the main difference with the MLS climatology by Huang et al. (2016) lies in the springtime and summertime maxima over eastern Siberia and Manchuria observed by IAGOS only. One possible reason is that the altitude level at 215 hPa in the extratropics is generally included in the LS according to our tropopause definition, despite the associated 5 km vertical resolution of MLS in the UTLS which allows upper tropospheric CO to impact the measurements at 215 hPa."*

**Pg 11: These graphs need axis points. You refer to gradients between e.g. 60E to 135E but I have to guess where those are on the plots. Pg 11, Fig 3: Also need lat/long labels on this figure.**

The longitudes shown below the longitudinal profiles correspond to the longitudes in the maps. This is now clearly indicated in the caption. Also, meridional coordinates have been added on the maps.

**Pg 12, 9 Definition of STE?**

STE means stratosphere-troposphere exchange. It is written in full words in the revised manuscript.

**Pg 15, 5 How many years of CO data is included?**

Figure 5 was modified and splitted in different plots; the number of years for both $O_3$ and CO are indicated in the new version. There are up to 11 years of CO data that are included in the cycles.

**Pg 15, 7 This should not be a new paragraph. You are discussing the same figure.**

The correction has been done.

**Pg 15, 10 "In most regions, there is no noticeable seasonal variation in the LS." Really? At least half of them show a similar range of seasonal changes in the UT and TPL. Which begs the question: What is the uncertainty/range of variability within each of the monthly mean values shown in Fig 5? Are any of these regional seasonal trends statistically different? Pg 16, 5 What is the uncertainty range on all these means? Are the results statistically different?**

The manuscript has been modified as shown below, in order to better explain how most regions have a low magnitude in their seasonal cycles, in the LS. The text below refers to the new Fig. 6.

*"[...] The seasonal cycles of the mean CO mixing ratio (central column) show a broader range of variability between the regions. The five western regions (Northwest America, East United-States, North Atlantic, Europe and the Western Mediterranean basin) exhibit a spring maximum and a late*

*summer - fall minimum in the UT and in the TPL. Asian regions (Siberia, Northeast Asia) present a different behaviour in the UT with a broad spring - summer maximum. Northwest America is noticeable for the highest mixing ratios recorded from November to April. Siberia shows a significant maximum in July. The relative seasonal amplitude – here defined as the (max-min)/mean ratio – ranges from 33% in the Western Mediterranean basin up to 46% in Northwest America. In comparison, the seasonal variations of CO in the LS are substantially lower in most regions. In this layer, the relative seasonal amplitude reaches its maximum at 27% in Middle East, followed by substantially lower amplitudes at 22% in Northeast Asia and East United-States, and within the 16 - 18% range in the other regions."*

Regarding the range of variability that characterize the monthly means shown in previous Fig. 5, we have added new figures (from 7 to 9) in the revised manuscript. They are meant to further inform on the interannual standard deviations of these climatological mean values. For ozone, the accompanying paragraph has been changed as follow:

*Complementary information is shown in Figs. 7, 8 and 9 representing the seasonal cycles of $O_3$, CO and $O_3$/CO respectively, for the monthly fifth percentile (P5), mean value and 95$^{th}$ percentile (P95). The interannual variability (IAV) is illustrated by the error bars, and defined as an interannual standard deviation of the monthly mean value. We first present the main characteristics in each of these three figures and their corresponding columns in Fig. 6, before analyzing and discussing the regional behaviours.*
*In Fig. 6, the mean $O_3$ mixing ratio (left column) shows a similar seasonal cycle for all regions. On average, the upper tropospheric $O_3$ mixing ratio ranges from 46 +/- 4 ppb in December - January up to 81 +/- 7 ppb in June - July, while the lower stratospheric $O_3$ mixing ratio ranges from 180 +/- 20 ppb in October - November up to 410 +/- 43 ppb in April - May, according to the following notation: mean +/- IAV. The IAV indicated here (not shown in this figure) is averaged over the regions.*

**Pg 16, 15 Definition of WCB?**

Warm Conveyor Belt . This has been corrected in the revised manuscript.

**Pg 16, 21 "percentile 95" What is this? How did you calculate it? What data was used to define it? There needs to be some detail on this here (or did I miss something earlier?!)**

A percentile (or a centile) is a metric used in statistics indicating the value below which a given percentage of observations in a group of observations fall. Percentile 95 is the value such that 95% of the measured values in the entire distribution are below and 5% are above. Here we use percentile 95 and percentile 5 to give the range of ozone or CO observations. The section 2.2.2 describes all the treatments applied to the IAGOS data.

**Pg 16, 22 What is the uncertainty/variability on these twin "peaks". Are the differences between April, May and June statistically significant so that you can call this a double peak?**

In Northeast Asia, the P95 is highly variable from one year to another, which is a common characteristic linked to the variability of biomass burning emissions. It takes place from

March until October (with an IAV >= 60 ppb). We do not try to interpret the interruption in May, since it can also be a sampling artifact. So the reviewer is right, we cannot assume it to be a double peak and the text has been changed. To summarize in the revised manuscript, we have replaced "peaks" by "high values". As shown in the following paragraph:

*"Among all the regions, Northeast Asia shows the highest mixing ratios during May-September. Its CO maximum ranges up to 140 ppb and lasts from April to June, in contrast to the April climatological mean from most regions around 120 ppb. **It is characterized by the seasonal maximum in the percentile 95 during the same months. The high values** in April are likely due to a strong wintertime CO accumulation in the lower troposphere (Zbinden et al., 2013), strong springtime agricultural fire emissions (Terezschuk et al., 2013) and boreal biomass burning emissions (Andela et al., 2013). **The high values** in June can be associated with anthropogenic and biomass burning emissions coupled with geographically more frequent warm conveyor belts (e.g. Madonna et al., 2014; Nédélec et al., 2005) and summertime Asian convection (Huang et al., 2016). [...]"*

New Fig. 8 provides information on the June peak in Northwest America (blue line), the Siberian July peak (brown line) and the Northeast Asian high extreme CO values in June too, and are consistent with the high interannual variability in the biomass burning emissions.

**How much of this difference could be attributed to uncertainty in the calculation of the TPL height?**

Indeed, a part of the difference between April, May and June may be due to uncertainties in the tropopause height. These uncertainties are difficult to quantify, notably because of the lack of additional observations to validate the PV fields in ECMWF simulations. IAGOS profiles can be used to investigate this point, but they represent only a short part of the IAGOS dataset used in our study (most data come from the cruise phase). As explained in another answer (and added into the manuscript), comparing the dynamical tropopause to the chemical tropopause (based on $O_3$ mixing ratios), Petetin et al. (2016) found a negligible systematic bias considering all profiles at Frankfurt. However, this result may not be valid in other regions.

**Pg 16, 29 Can you check GFED for the timing delay between peak fires and the peak you see? The peak fire season has been shifting towards August in recent years so has the CO signal from BB also been delayed in the UT? You also don't define GFED or GFAS.**

GFED and GFAS are now defined, and the associated references are indicated (respectively: Van der Werf et al., 2010 and Kaiser et al., 2012). We have checked GFAS for the timing delay in the months of intense fire emissions for CO. First, there is no clear indication that the peak fire season has been shifted towards August. Actually, the seasonal cycle of GFAS emissions over Boreal Asia varies a lot from one year to the other. For instance, if we focus on the years of most intense fires, the months that maximize GFAS emissions are May in 2003, April in 2008, and June-July in 2012. Second, we did not notice systematic coincidence between these emissions maxima and the maxima observed with IAGOS in the UT: in Siberia, the highest anomaly in 2012 effectively occurs during July (+18 ppb), but in 2003,

July shows the highest anomaly by far (43 ppb, more than twice for every other month in this region). This feature contrasts to GFAS emissions and thus shows that such source-receptor link has to be investigated with specific tools, based on Lagrangian models (see Sauvage et al., 2017 for application of a coupled system between FLEXPART and GFAS to assess the origin of CO anomaly observed by IAGOS. Note that there is no IAGOS data in Siberia during 2008.

**Pg 17, 25 "All these features seem to indicate less frequent springtime STE events in WNAm." You never explained what STE is so I've no idea what this is. This whole paragraph needs to be re-written in simpler language. P5 and P95 have not been explained so far aside from the abstract.**

Corrected in the revised manuscript.

**Pg 25, 12 "mentioned" mis-spelled throughout paper**

Corrected in the revised manuscript.

**Pg 25, 12 what years did the Thouret paper cover? 9 years is a bit vague.**

1994 - 2003. Corrected in the revised manuscript.

**Pg 25, 23 "a more frequent sampling, thus smoothing the temporal variability". This doesn't make sense. Do you mean you have more data available within the greater sized grid box, allowing for a more significant statistical analysis?**

Exactly. We propose to change the formulation for the following one:

*"Since we compute the trends with the same methodology, the discrepancies are due to the greater size of our region Europe, which allows a higher amount of data and thus a more significant statistical analysis."*

**Pg 27, 24 "We assume that in the UT, the spatial variability is too weak to be responsible for the discrepancies between the two studies." Could you use the longitudinal trend from Fig 2/3 do quantify the expected changes? How significant is the P95 trend you see?**

This sentence on the spatial variability was not necessary and has been removed in the revised manuscript. It simplifies the text.

The P95 is 3-sigma significant (the p-value equals 0.008). It has been added in the text:

*"However, they did not observe the 99%-significant positive trend of P95 that we derived (+0.4 ppb.yr-1, p = 8.10-3)."*

**Pg 27, 25 do you mean $O_3$ increasing over time?**

Yes. It has been corrected.

A new figure has been added for this purpose (Fig. 12) in the revised manuscript, as shown below.

[Figure]

*Figure 12. O₃ trends (ppb.yr⁻¹) in the UT, over the whole monitoring period (1994 - 2013: top panels), over 1994 - 2008 (central panels) and over the monitoring period for CO (2002 - 2013: bottom panels), for the monthly fifth percentile (left column), mean value (central column) and 95ᵗʰ percentile (right column). For each significant trend, the error bars represent the 95% confidence interval. The insignificant trends are represented by the hatched areas. The coloured labels correspond to the regions defined in Fig. 1.*

In this figure, the trends in ozone in the UT are computed on the period 1994 - 2008 (medium level panels). Compared to the trends computed for the whole period (shown again in the top panels, in order to make the comparison easier), the statistical significance of the increase in upper tropospheric ozone has been lost in four of the seven regions. Only Europe and Northeast Asia have a significant increase in ozone, respectively at +0.3 [0.05; 0.55] and +0.5 [0.1; 0.9], when removing the last five years. It also does when removing the last three years only (see the bottom panels). Thus, it is probably not a consequence of a lower amount of data. It rather comes from a high sensitivity of the 95%-confidence intervals to the higher ozone values during the period 2011 - 2013.

 As written in the revised manuscript:

*The upper tropospheric $O_3$ trends are also computed over several periods, in order to test the sensitivity of our results to the monitoring period. These trends are shown in Fig. 12 for the three metrics (mean value, fifth and 95th percentiles). The trends over the whole period are compared to the ones computed over 1994 - 2008, thus excluding the last three years' positive anomalies, and to the ones computed over 2002 - 2013, i.e. the monitoring period for CO. Removing the last years positive anomaly leads to the loss of the 2-sigma statistical significance for most significant trends, in three regions for the fifth percentile (North Atlantic, Western Mediterranean basin, Middle East), four regions for the mean value (East United States, Western Mediterranean basin, Middle East, Siberia), and the only two regions for the 95th percentile (Europe, Western Mediterranean basin). Consequently, the significant increase of the mean $O_3$ mixing ratio shown in Fig. 10 is robust for Europe and Northeast Asia, and the significant increase in the fifth percentile is robust also in East US and Siberia. In the other cases, the 2-sigma confidence interval is strongly sensitive to the higher ozone values during 2001 - 2013.*
*On the 2001 – 2013 period (bottom central panel in Fig. 12), the statistical significance for the mean values is stronger in Middle East and Siberia, and weaker elsewhere compared to the full period. The regions where it decreases also show a weakening in the significance of the fifth percentile. The trends in 95th percentiles (right panels) appear to be limited by the first years. Thus, it shows a transition between the first part and the last part of the monitoring period, highlighted by a change in the trends in the monthly distribution. The upper tropospheric $O_3$ trends are higher during 2002 - 2013 because of the positive anomaly at the end of this period. One possible explanation for this anomaly lies in an enhanced transport across the tropopause caused by the 2009 - 2010 strong El Nino event, as shown in Lin et al. (2015). Responses in tropospheric ozone at the midlatitudes to the El Nino Southern Oscillation (ENSO) have been identified in Wespes et al. (2017) with a 4-month or 6-month time lag. Further modeling studies are needed to assess the link between the chemical composition of the upper troposphere and the ENSO.*

 **Pg 27, 32 "Although no trend calculation was performed in WNAm, it may confirm the upper tropospheric $O_3$ increase as effective in the whole northern mid-latitudes, except the Pacific ocean." If there was no trend calculated due to a lack of data, then this sentence should not be included. You can't confirm or deny anything!**

This is true, the sentence has been removed in the revised manuscript.

**Pg 27, 35 Fig 8 shows an increase in $O_3$ in the UT because there is missing data! How is the mean/smoothing calculated for each region when there is a data gap? 2011 probably has one of the lowest $O_3$ minimums with missing data before.**

There may be a misunderstanding. By "missing data", do you mean the 1 or 2 month(s) in late 2011/early 2012 ? 2010 has almost no data, but 2011 is almost a fully-monitored year. There is no smoothing, the trends are computed directly from the monthly anomalies, as explained in the methodology section.

We do not clearly understand the comment/question of the reviewer. The data treatment is explained in Sect. 2.2.2: no interpolation is used to fill data gaps, all trends are computed directly from the monthly anomalies (with or without data gaps). During the year 2011, there are no data gap, except for one month. Strong positive monthly anomalies of $O_3$ are observed over almost all the year 2011, the only exception being one strong negative anomaly observed during one specific month in the 2011/2012 winter (as clearly shown for Europe in Fig. A1 in the Appendix A). Besides, this negative anomaly is most probably due to a low sampling during this month, because the monthly distribution is very sharp (anomalously low difference between the P95 and the P5). Thus, the year 2011 remains characterized by relatively strong $O_3$ mixing ratios.

**Pg 28, 3 "mentionned" should be mentioned**

Corrected in the revised manuscript.

**Pg 32, Fig 16, 17: What do the hatched areas mean? (I assume insignificant trend but you should state it). You should also explain in one of these figures what regions each of the labels represented. All info to interpret the figure should be in the caption.**

It is now indicated explicitly in the caption. For the acronyms, the caption refers to Fig. 1 (the definition of the regions).

**Specifically to Fig 16: What do you gain from including the 5%, 50% and 95% trends? How is computed? Are the 95% CI for the**

The mean value is represented in those figures, but not the median.

Focusing on the trend analysis of mean mixing ratios is limited, as it gives no additional information on the trends of the other parts of the distribution of mixing ratios. In particular, it is common to investigate the trends of the 5[th] and 95[th] percentiles of mixing ratios in order to see how are changing the low and high tails of the distribution of mixing ratios. The new version of Sect. 2.2.2 explains how they are calculated. About what we can infer about the last question of the reviewer (in which the end is missing), all trends in the paper are given at a 95% confidence level.

**Pg 34, 2: extend should be extent**

Corrected.

Referee #2

We thank reviewer#2 for his/her comments that definitely helped improving the manuscript.

Note that following reviewer#1's recommendations, previous Figs. 8 to 13 have been moved in the Appendix.

**This paper presents a decadal scale record of ozone and carbon monoxide measurements that is used in the Tropospheric Ozone Assessment Report (TOAR) and will be critical in the evaluation of chemistry climate models that require accurate atmospheric composition for understanding emissions, chemistry, transport and radiative forcing. I think the basic methods for climatology and trend analysis are sound. I agree with Referee #1 that more science motivation could be added to the paper and would require only minor revisions.**

Some changes have been done, involving an additional study of the $O_3/CO$ ratio (see our answers to Referee#1).

**Regarding $O_3$ trends and sources of variability, the following references could be included:**

**Lin, M., W. Horowitz, R. Payton, A.M. Fiore, G. Tonnesen (2017). US surface ozone trends and extremes from 1980 to 2014: Quantifying the roles of rising Asian emissions, domestic controls, wildfires, and climate. Atmos. Chem. Phys., doi:10.5194/acp-17-2943-2017**

Although very interesting, this article deals with surface ozone/air quality, and when free tropospheric ozone is mentioned, it corresponds to the lower part of the free troposphere (700 hPa). As our study focuses on the UTLS region, including this reference in our study may not be relevant.

**Lin, M., A. M. Fiore, L. W. Horowitz, A. O. Langford, S. J. Oltmans, D. Tarasick, and H. E. Rieder (2015a), Climate variability modulates western U.S. ozone air quality in spring via deep stratospheric intrusions, Nature Communications, 6(7105), doi:10.1038/ncomms8105.**

**Wespes, C., D. Hurtmans, C. Clerbaux, and P.-F. Coheur (2017), $O_3$ variability in the troposphere as observed by IASI over 2008–2016 Contribution of atmospheric chemistry and dynamics, J. Geophys. Res. Atmos., 122, 2429–2451, doi:10.1002/2016JD025875.**

We thank the reviewer for these suggestions. They are helpful to understand the positive anomalies at the end of our monitoring period. They have been added within the new paragraph that comments new Fig. 12. This paragraph is pasted below, and those references are underlined in this new text.

*"The upper tropospheric $O_3$ trends are also computed over several periods, in order to test the sensitivity of our results to the monitoring period. These trends are shown in Fig. 12 for the three metrics (mean value, fifth and $95^{th}$ percentiles). The trends over the whole period are compared to the ones computed over 1994 - 2008, thus excluding the last three years positive anomalies, and to the ones computed over 2002 - 2013, i.e. the monitoring period for CO. Removing the final positive anomalies leads to the loss of the 2-sigma statistical significance for most significant trends, in three regions for the fifth percentile (North Atlantic, West Mediterranean basin, Middle East), four regions for the mean value (East United States, West Mediterranean basin, Middle East, Siberia), and the only two regions for the $95^{th}$ percentile (Europe, West Mediterranean basin). Consequently, the significant*

*increase in the mean O₃ mixing ratio shown in Fig. 10 is robust for Europe and Northeast Asia, and the significant increase in the fifth percentile is robust also in East US and Siberia. In the other cases, the 2-sigma confidence interval is strongly sensitive to the higher ozone values during 2001 - 2013. On the central panel at the bottom, the statistical significance for the mean values is increased in Middle East and Siberia, and decreased elsewhere by removing the start of the monitoring period, until 2001. The regions where it decreases also show a weakening in the significance of the fifth percentile. The trends in 95th percentiles (right panels) appear to be limited by the first years. Thus, it shows a transition between the first part and the last part of the monitoring period, highlighted by a change in the trends in the monthly distribution. The upper tropospheric O₃ trends are higher during 2002 - 2013 because of the positive anomaly at the end of this period.* **One possible explanation for this anomaly lies in an enhanced transport across the tropopause caused by the 2009 - 2010 strong El Nino event, as shown in Lin et al. (2015). Responses in tropospheric ozone at the midlatitudes to the El Nino Southern Oscillation (ENSO) have been identified in Wespes et al. (2017) with a 4-month or 6-month time lag. Further modeling studies are needed to assess the link between the chemical composition of the upper troposphere and the ENSO."**

Specific comments:

**P1, Line 15: Since trends were only computed for the northern mid-latitudes, perhaps this should say: "... to derive trends in the northern mid-latitude UTLS"**
Corrected.

**P2, Line 20: "It is also important for enhancing the knowledge about the role of O₃ in the increasing atmospheric radiative forcing." CO emissions also contribute to increasing RF. Maybe say: "Trends in both gases are also important for assessing changes in atmospheric radiative forcing."**
Corrected.

**P5, Line 15: Define TOAR acronym**
In the revised manuscript, the TOAR acronym has been replaced by its full name: Tropospheric Ozone Assessment Report.

**P5, Line 25: Do you ever find double tropopause cases? If so, are these included or filtered out? These can be very common (50-70%) in winter in the N. Midlatitudes [Randel et al., JGR, 2007: doi:10.1029/2006JD007904], but they might be hard to detect with 1∘x1∘ PV.**
The following paragraph has been added in the 2.2.1 subsection, in the methodology, as shown below:
*"The tropopause pressure is systematically defined as the highest level of the 2 pvu isosurface. As a consequence, the double tropopause events commonly encountered at mid-latitudes, more frequently in winter (Randel et al., 2007), are not filtered out. It implies that air masses attributed to the UT are not purely tropospheric in case of double tropopause but may include some stratospheric intrusions."*

There has also been a change in the 3.2 section (underlined text in the paragraph below), when we comment the figure on ozone seasonal cycles:
*"On the left panels in Fig. 6, the mean ozone mixing ratio shows a similar seasonal cycle for all regions. The seasonal maximum generally takes place in June-July in the UT, May-June in the TPL and April-May in the LS. This feature highlights a seasonal cycle in the TPL halfway between upper tropospheric and lower stratospheric cycles, and confirms again the tropopause definition used here as a realistic transition between the troposphere and the stratosphere.* Also, although the present methodology does not filter out the double tropopause events, the seasonal cycles in the UT show systematically their minima during winter. Even the percentile 95 does not show high wintertime*

*ozone values. Hence the impact of the unaccounted intrusions of stratospheric air masses is not visible in the upper tropospheric ozone seasonal cycles."*

**P7, Line 1: It would be helpful to say that a 95% confidence level corresponds to trends that are > 2-sigma.**
Thank you for this suggestion, it makes the text clearer. The changes are shown in the reformulation below:
*"In this paper, all trend uncertainties are given at a 95% confidence level (i.e. above 2 sigma)."*

**P9, Line 19: warm conveyor belts = WCBs?**
True. It is written in full words now.

**P12, Line 10: "The discrepancies with the IAGOS climatologies can be due to uncertainties involving the stratospheric signal, i.e. the ozone stratospheric column, the height of the tropopause, and the total ozone column" I don't understand why "total column ozone" is in this list. Don't you mean the differences between the full tropospheric column and only the UT column?**
Reviewer is right. It has been removed in the revised manuscript.

**P12, Line 25: "Except for India where few summertime IAGOS data do not allow the comparison" Are there less flights in summer? Should be changed to "Except for India where summertime IAGOS data is limited due to [explanation]**
The text has been changed, as shown below.

*"In the tropics, Livesey et al. (2013) used Aura-MLS observations at 215 hPa since 2004 until 2011. They highlighted CO maxima over India and southeast Asia during July - August, northern equatorial Africa in February-April, southern equatorial Africa in September-November, and equatorial Brazil in October-November. According to their study and references therein, the two maxima over Asian regions are linked with anthropogenic emissions uplifted to the tropical UT by strong convection (Jiang et al., 2007), whereas the other maxima originate from biomass burning. Most of these maxima are consistent between the MLS and IAGOS data sets. One exception may be northern India (25 - 30°N cells) where IAGOS does not clearly highlight a summertime maximum. However, available IAGOS data in this region and season are too scarce to conclude, likely because IAGOS aircraft often fly below the UT lower boundary ($P_{2\,pvu}$ + 75 hPa) in the 25 - 30°N zonal band that becomes subtropical in summer, with a tropopause typically reaching altitudes above 150 hPa."*

**P29,32 Figs 14,15,16,17 caption should state that bars with faded colors (and no error bars) did not have significant trends**
It is true. It is now stated in all the captions in the revised manuscript.